# In-silico tool based on Boolean networks and meshless simulations for prediction of reaction and transport mechanisms in the systemic administration of chemotherapeutic drugs

**Fabián Mauricio Vélez Salazar**⬤*, **Iván David Patiño**

Grupo de Investigación e Innovación Ambiental (GIIAM), Institución Universitaria Pascual Bravo, Medellín, Colombia

\* mauricio.velez@pascualbravo.edu.co

## Abstract

Using in-house computational tools, this work focuses on investigating how the combination of the electric field magnitude ($E$), bloodstream velocity ($\lambda_{inl}$) and pharmaco-kinetic profile ($PK$) impacts the reaction and transport mechanisms of drug ($RTMs$) arising in electro-chemotherapeutic treatments. The first step implies retrieving the ratios between extracellular, free intracellular, and bound intracellular concentrations from numerical simulations, employing a meshless code developed, calibrated and validated in a previous work. Subsequently, a Boolean model is developed to determine the presence, interaction and rates of $RTMs$ based on the comparison of the spatio-temporal evolution of the drug concentration ratios, being this the main contribution of the present work to the comprehension of the phenomena involved in the systemic administration of chemotherapeutic drugs in cancer tumors. Different combinations of $E$ (0 kV/m, 46 kV/m, 70 kV/m), $\lambda_{inl}$ ($1x10^{-4}m/s$, $1x10^{-3}m/s$, $1x10^{-2}m/s$) and $PK$ (One-short tri-exponential, mono-exponential) are examined. In general, results show that both the presence and relative importance of $RTMs$ can differ between both $PKs$ for a given combination of $E$ and $\lambda_{inl}$. Additionally, for a given $PK$, radial uniformity of transmembrane transport rate is aversively affected by the increase of $E$ and $\lambda_{inl}$, whereas radial homogeneity of association/dissociation rate is monotonously affected only by $E$. Regarding the axial uniformity of transmembrane transport rate, this is benefited by the increase of $\lambda_{inl}$ and, in a lower extent, by the reduction of $E$.

## 1. Introduction

The systemic administration of chemotherapeutic medicaments highly depends on the pharmacokinetic profile and blood perfusion rate. Despite this administration method can pose some disadvantages due to the potential occurrence of side effects on healthy tissues, it is still the most common method to deliver drug to cancerous tissues since it facilitates drug

**Funding:** The author(s) received no specific funding for this work.

**Abbreviations:** Symbol, Meaning; $C_1$, Extracellular drug concentration; $C_2$, Free intracellular drug concentration; $C_3$, Bound intracellular drug concentration; $C_v$, Drug concentration along the bloodstream; $AS$, Association rate; $D$, Interstitial diffusion coefficient of tissue; $D_0$, Interstitial diffusion coefficient of tissue (non-electroporated); $DIE$, Degree of irreversible electroporation; $DIS$, Dissociation rate; $DOE$, Degree of electroporation; $DOE_R$, Transient degree of electroporation; $d_{pulses}$, Pulse spacing; $E$, Electric field magnitude; $ECM$, Extracellular matrix; $ECT+$, Positive extracellular transport; $ECT-$, Negative extracellular transport; $EP$, Electroporation; $E_{rev}$, Reversible threshold of electroporation; $E_{irrev}$, Irreversible threshold of electroporation; $EV$, Extravasation rate; $EX$, Externalization rate; $f_{pulse}$, Pulse frequency; $FEM$, Finite element method; $FVM$, Finite volume method; $GMAPS$, Global Method of Approximate Particular Solutions; $ICT$, Intracellular transport; $IFP$, Interstitial fluid pressure; $IN$, Internalization rate; $IRE$, Irreversible electroporation; $k_1$, Cell membrane permeability; $k_{1,max}$, Maximum value of cell membrane permeability; $k_{1,min}$, Minimum value of cell membrane permeability; $k_2$, Drug association rate (Reaction property); $k_{-2}$, Drug dissociation rate (Reaction property); $k_{v,max}$, Maximum value of vessel wall effective permeability; $k_{v,min}$, Minimum value of vessel wall effective permeability; $k_v$, Effective vessel wall permeability; $k_{vc}$, Vessel wall permeability coefficient; $L_{cord}$, Length of cord domain; $LD$, Lymphatic drainage; $m_r$, Vasoconstriction coefficient; $N_{ep}$, Number of electroporation treatments; $N_{points}$, Number of collocation points; $N_{pulses}$, Number of consecutive pulses; $p_{inl}$, Inlet blood pressure; $PK$, Pharmacokinetic profiles; $p_p$, Permeate pressure; $Q_{leak}$, Leakage flow rate through the vessel wall; $Q_{inl}$, Inlet flow rate; $r$, Radial coordinate; $r_v$, Vessel radius; $\hat{r}$, Normalized vessel radius; $\hat{r}_{min}$, Minimum normalized vessel radius; $R_{cord}$, Radius of cord domain; $RE$, Reversible electroporation; $RUC$, Representative unitary cell; $RTMs$, Reaction and transport mechanisms; $S_C$, Concentration scale; $S_L$, Length scale; $S_T$, Time scale; $TBV$, Transport along the blood vessels; $t$, Time; $T_{chemo}$, Simulation time of electro-chemotherapy; $T_{ep}$, Time of each electroporation treatment; $\tau_i$, Decay time coefficient for the permeability (vessel wall or cell membrane); $TME$, Tumor microenvironment; $TMT$, Transmembrane transport; $t_p$, Time elapsed from

distribution through the tumor microvasculature [1]. Electroporation has arisen as a suitable technique that enables to reach the target concentration of chemotherapeutic agents within specific regions of the tissue, thereby localizing the cytotoxic effects of the treatment. The appropriate choice of parameters such as the voltage level, pulse form, spacing, length, and frequency, as well as number and duration of electroporation treatments, and electrodes location is crucial for achieving the primary objective of reversible electroporation ($RE$), namely, to augment the cell membrane permeability and facilitate the drug translocation towards the intracellular space while maintaining the cell viability [2–4]. Additionally, the utilization of high-voltage, short-duration electric pulses can induce modification of vessel diameter and endothelial permeability, which in turns leads to the alteration of convective drug transport within the vessel, being this particularly critical in fenestrated and/or distal tumor vessels [2, 5, 6]. As previously studied in Vélez et al. [7], the type of pharmacokinetic profile and the bloodstream velocity are very influential factors in the spatio-temporal evolution of the extracellular, free intracellular and bound intracellular concentrations of drug in electroporated and vasoconstricted tissues. This implies that the reaction and transport mechanisms ($RTMs$) involved in the drug passage from the circulatory system to the cell can be also modified by these parameters. These mechanisms are represented in Fig 1, where they can be identified: transport along the blood vessels ($TBV$), transvascular transport ($TVT$), extracellular transport ($ECT$), transmembrane transport ($TMT$), and intracellular transport ($ICT$).

The $TBV$ mechanism is modelled here using an advection/convection equation, which is based on an established model for fluid flow through porous circular tubes and incorporates a previously published model [7] to address the vessel vasoconstriction and subsequent recovery. The transvascular transport is denoted as $TVT+$ or $EV$ when it occurs from the vessel to the interstitial space (extravasation), and as $TVT-$ or $LD$ when it occurs in the opposite direction (lymphatic drainage). In the present work, it is employed a Fickian diffusion model that incorporates an effective vessel wall permeability and is valid when the transvascular transport is mainly driven by the species concentration gradients rather than by the pressure gradient across the vessel wall [8–10].

On the other hand, the term $ECT+$ is used to denote a positive net extracellular transport in the Representative Unitary Cell ($RUC$), indicating an increase in the extracellular concentration ($C_1$), while $ECT-$ refers to a negative net extracellular transport, signifying a decrease in $C_1$. In the original tumor cord tumor model, it is assumed that the dominant mode of drug transport in the extracellular space is diffusion [8–10]. A modified tumor cord model that incorporates a convective term to account for the volume change of the tissue when vasoconstriction occurs was used in Vélez et al. [7] and is taken up here again.

Besides, given that cellular uptake and efflux of drug can occur simultaneously inside the same $RUC$, the net rate of transmembrane transport ($TMT$) is considered here, in such a way that the terms $TMT+$ or $INT$ denote a net internalization of drug, whereas $TMT-$ or $EX$ indicate a net externalization. In the tumor cord approach used here the transmembrane transport is incorporated by source/sink terms into the species transport equations for extracellular concentration ($C_1$) and free intracellular concentration ($C_2$), taking into account the effective permeability of the cell membrane ($K_1$) modified by a function known as the degree of reversible electroporation ($DOE_R$) proposed by Boyd and Becker [11].

Regarding the reaction mechanisms in the intracellular environment, the processes of drug association ($AS$) and dissociation ($DIS$) with proteins play a significant role in determining the free intracellular ($C_2$) and bound intracellular concentrations ($C_3$). More details about the $RTMs$ are presented in the Supplementary material. In the tumor cord model employed in this study, the aforementioned reaction mechanisms are incorporated into the species transport

application of last pulse; $\tau_r$, Decay time coefficient of vasoconstriction; *TVT*, Transvascular transport; *V*, Voltage; $\alpha$, Ratio of total surface area of cell membrane; $\alpha^*$, Sigmoidal function parameter; $\beta^*$, Sigmoidal function parameter; $\delta_1$, Extracellular volume content; $\delta_2$, Intracellular volume content; $\sigma$, Effective electrical conductivity; $\sigma_{max}$, Maximum effective electrical conductivity; $\sigma_{min}$, Minimum effective electrical conductivity; $\sigma_{irrev}$, Irreversible effective electrical conductivity; $\lambda$, Blood velocity along the vessel; $\lambda_{inl}$, Inlet blood velocity; $\mu$, Plasma viscosity; $\omega$, Relaxation factor; $\Delta t$, Time step.

equations as source/sink factors, depending on the drug association ($K_2$) and dissociation rates ($K_{-2}$), which in turns are considered unaltered with the electric pulse application.

As observed, the transport and reaction processes associated with the drug passage from blood vessels to tumor cells exhibit a high level of complexity, which in turns can be increased

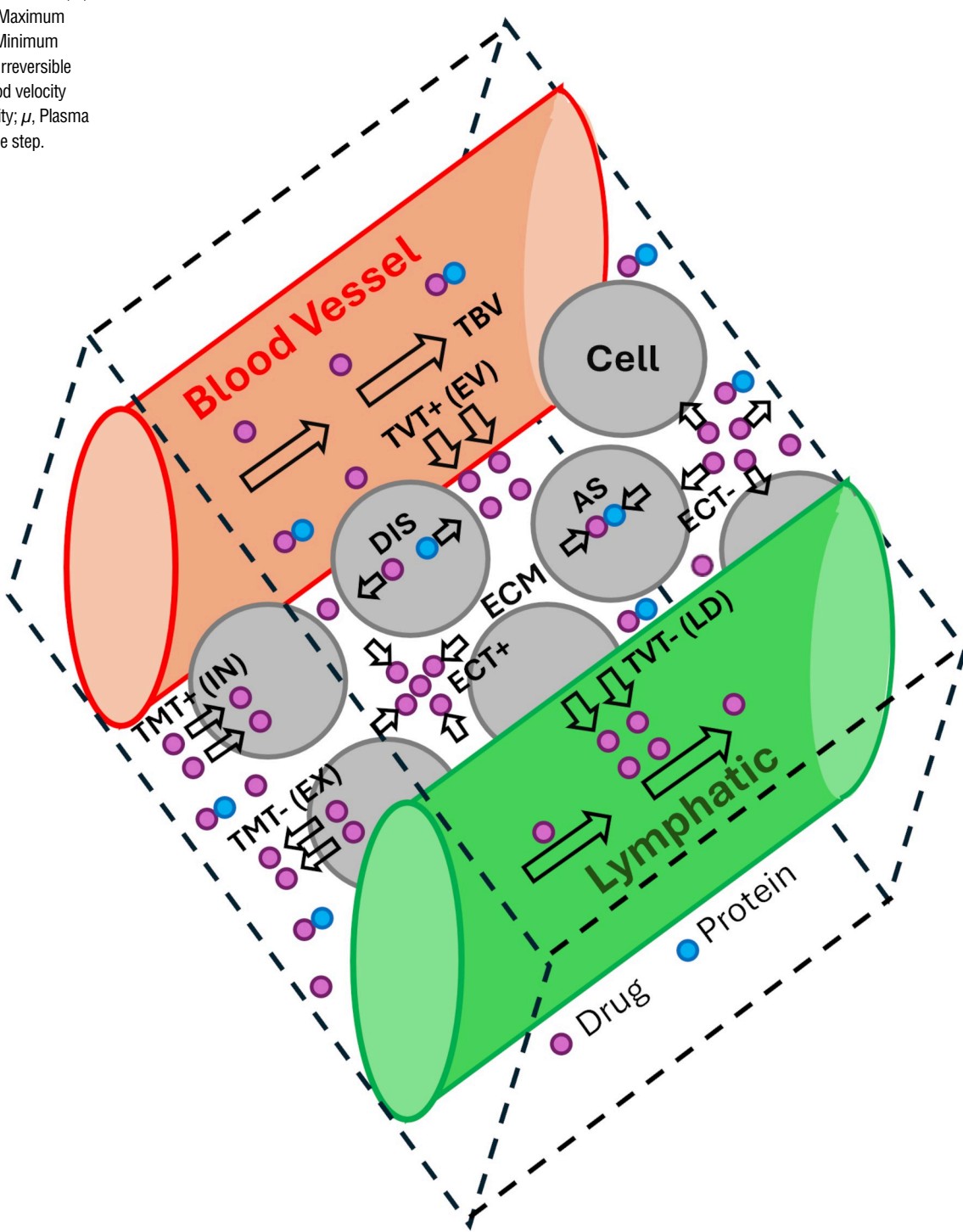

**Fig 1. Reaction and transport mechanisms of electro-chemotherapeutic treatments.**

by the electric pulse application. As a result, there has been an increasing demand of computational tools to better comprehend the intricate interaction between these mechanisms, and how this interaction influences the spatial and temporal changes of drug concentration within the tissue domain [12–18]. For instance, Mohammad et. al. [18] developed a multi-scale numerical model of drug delivery into solid tumors that included transport along blood vessel, transvascular and extravascular transport. This model was used to analyze how the microvascular network and vascular normalization from anti-angiogenic therapy affect the convective and diffusive transport of drugs. In the present study, the influence of two parameters of the systemic drug administration on the transport and reaction mechanisms in electroporated and vasoconstricted tissues is investigated, namely, the inlet bloodstream velocity ($\lambda_{inl}$) and pharmacokinetic profile (*PK*), considering several electric field magnitudes (*E*). Firstly, the spatio-temporal evolution of the ratios $C_2/C_1$, $C_3/C_1$, and $C_3/C_2$ is obtained using a formerly developed, calibrated and validated in-silico tool that employs the Global Method of Approximate Particular Solutions (*GMAPS*) to solve the governing Eq [7]. This code incorporates the effects of applying electric pulses on some parameters, including vessel diameter, vessel wall permeability, cell membrane permeability, and interstitial porosity. Then, logical inference rules are applied here to determine relationships between the previously mentioned mechanisms (*TBV*, *TVT+ or EV*, *TVT− or LD*, *ECT+*, *ECT−*, *TMT+ or IN*, *TMT− or EX*, *AS*, *DIS*). The simulation campaign conducted in the present study comprises three different magnitudes of electric field: $E = 0\ kV/m$ (no electroporation), $E = 46\ kV/m$ (reversible threshold), and $E = 70\ kV/m$ (irreversible threshold). Additionally, two different pharmacokinetic profiles, *PK* (one-short tri-exponential and mono-exponential), and three inlet blood velocities, $\lambda_{inl}$ ($1x10^{-4}m/s$, $1x10^{-3}m/s$ and $1x10^{-2}m/s$), are deemed.

The pharmacokinetic profiles stand for the time evolution of drug concentration ($C_v$) along the bloodstream considering processes of absorption, distribution, metabolism and elimination with little physiological details [19]. These profiles depend on the drug type, loading and maintenance doses, infusion time, perfusion rate, tissue affinity, drug lipophilicity or hydrophilicity, plasma protein binding, among other factors [19, 20]. Both mono-exponential (*MPK*) and tri-exponential profiles (*TPK*) has been considered for intravenous (IV) injection of doxorubicin in previous works [9, 10, 21–26]. The *MPK* is a one-compartment, bolus IV injection profile, where total drug dose is rapidly injected (order of seconds) and the time decay rate of $C_v$ is constant since the drug distribution and elimination rates are assumed to be uniform for all tissue groups [19, 20]. On the other hand, the *TPK* is a multi-compartment, one-short IV injection profile, where the total drug injection does not occur immediately, but in short period of time (order of minutes), to then decrease tri-exponentially due to the consideration of different distribution and elimination rates according to the tissue group [19, 20]. In general, a mono-exponential decay is typically evidenced when the drug is prone to be distributed to only richly perfused tissues and organs, such as liver, kidney and brain [27]. The parameters of both pharmacokinetic profiles (*MPK* and *TPK*) for doxorubicin were obtained from Groh et al. [9], where the total drug exposure, defined as the Area Under the Curve (*AUC*) of $C_v$ *vs* t, namely, $AUC = \int_0^\infty C_v.dt$, which is a more useful target parameter at clinical practice, is the same for both profiles ($AUC \approx 2.78\ \mu Mh$).

The bloodstream velocity ($\lambda_{inl}$), which is the other parameter considered here, is directly related with the blood perfusion rate, which in turns plays a major role for the systemic administration of drugs. At clinical level, the low blood perfusion in some tumors due to the vessels compression and leakiness can lead to hypoperfusion and hypoxia. While hypoperfusion reduces the medication delivery, hypoxia can foster the tumor growth through immunosuppression, increase of metastatic potential and drug resistance, and generation of disordered

angiogenesis [28]. Therefore, several therapeutic strategies have been developed to improve the blood perfusion and therapy outcomes, such as [28, 29]: vascular normalization with anti-angiogenic agents, stroma normalization with mechano-therapeutics, tumor microenvironment (*TME*) normalization with metronomic chemotherapy or nanomedicine, ultrasound sono-permeabilization, among others. These strategies are focused on the regularization of the bloodstream velocity in the vessels by reducing the endothelial gaps associated to the leakage flow rate, and/or alleviating the mechanical stresses exerted by the tumor microenvironment (*TME*) surrounding these vessels. Accordingly, the influence of the bloodstream velocity ($\lambda_{inl}$) on the reaction and transport mechanisms (*RTMs*) of chemotherapeutic drugs can have clinical importance.

The principal purpose of the present research is to investigate the influence of the combined input parameters ($E$, $\lambda_{inl}$, *PK*) on the interaction between the aforementioned reaction and transport mechanisms in the context of electro-chemotherapeutic treatments, aiming to propose potential clinical implications resulting from this combination. As detailed in the Supplementary Material, the transport and reaction phenomena involved in the drug passage from the circulatory system to the intracellular space have been studied at both macroscopic and microscopic scales by experiments and numerical simulations. The cited in-silico studies have used mainly meshed techniques (Finite Element Methods, Finite Volume Method, Finite Difference Method, etc.). Particularly, this research group have developed several in-silico tools using meshless and mesh-boundary techniques to predict the extracellular ($C_1$), free intracellular ($C_2$) and bound intracellular concentration ($C_3$) of drug in electroporated and vasoconstricted tissues at both microscopic [7, 17, 30] and macroscopic scale [31, 32]. To the best of our knowledge, the evolution of the reaction and transport mechanisms (RTMs) considering the spatio-temporal behavior of these concentration fields ($C_1$, $C_2$ and $C_3$) has been tackled using merely descriptive analyses, but not a rigorous logical inference analysis. Accordingly, one of the main contributions of this work lies on the development of a Boolean model to accurately establish the presence, interaction and rates of the *RTMs* in electroporated and vaso-constricted tissues applying logical inference laws for all possible combinations of the ratios $C_2/C_3$, $C_3/C_1$ and $C_3/C_2$, which in turns are retrieved from numerical simulations using a *GMAPS* code previously developed, calibrated and validated [7].

## 2. Mathematical modeling and numerical methods

The current study uses an axisymmetric, continuum, tumor cord model for the species transport (see Fig 2). Usually, this model has been used considering stationary domains [8, 9]. In a recent work [7], the governing equation for the interstitial space was modified by incorporating a divergence term that accounts for the advective transport resulting from the tissue expansion and contraction due to the vessel vasoconstriction. A summary of the deduction of the modified model is presented in the Appendix section of Vélez et al. [7]. The modified tumor cord equations read:

$$\delta_1 \frac{\partial C_1}{\partial t} = D\nabla^2 C_1 - \nabla C_1 \cdot \vec{u} + \alpha.k_1(E,t).(C_2 - C_1) \tag{1}$$

$$\delta_2 \frac{\partial C_2}{\partial t} = \alpha.k_1(E,t).(C_1 - C_2) - \delta_2 k_2 C_2 (C_o - C_3) + \delta_2 k_{-2} C_3 \tag{2}$$

$$\delta_2 \frac{\partial C_3}{\partial t} = \delta_2 k_2 C_2 (C_o - C_3) - \delta_2 k_{-2} C_3 \tag{3}$$

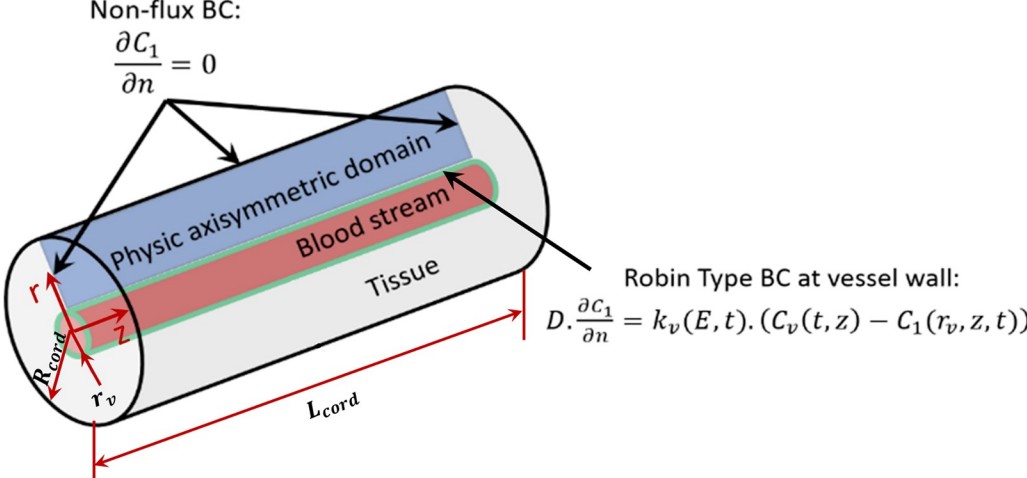

**Fig 2. Boundary conditions of the physical domain.**

where the field variables consist of the extracellular drug concentration ($C1$), free intracellular drug concentration ($C_2$) and bound intracellular drug concentration ($C_3$). The geometric-dependent properties $\delta_1$, $\delta_2$ and $\alpha$ stand for the extracellular volume content, intracellular volume content, and ratio of total surface area of cell membrane to the cord domain volume, respectively. The transport properties of the tissue are the interstitial diffusion coefficient ($D$) and cell membrane permeability ($k_1$), which are dependent on the magnitude of the electric field and time. On the other hand, the drug association rate ($k_2$) and dissociation rate ($k_{-2}$) are the reaction properties. It is crucial to emphasize that the effective diffusivity, $D$, is dependent on the porosity of the extracellular space, $\delta_1$, and can be updated as $D = D_0 \cdot (\delta_1/\delta_{1,0})$ when the time change of porosity is small, with $D_0$ and $\delta_{1,0}$ as the diffusion coefficient and porosity corresponding to the original cord domain (without electroporation). The velocity field of the tissue induced by the vasoconstriction and subsequent recovery of vessel radius is assumed to have only radial components, namely, $\overrightarrow{u} = (u_r, 0)$.

The boundary conditions depicted in Fig 2 include a Robin-type boundary condition at the tissue-vessel wall interface, as shown in the following equation:

$$D.\frac{\partial C_1}{\partial n} = k_v.(C_v(t,z) - C_1(r_v,z,t)) \tag{4}$$

where $r_v$ is the vessel radius that is considered changeable due to vasoconstriction, $k_v(E,t)$ is the vessel wall permeability, $C_v(t,z)$ is the time and space-dependent drug concentration along the bloodstream, $C_1(r_v,z,t)$ is the extracellular drug concentration at the tissue-vessel wall interface, and $\partial C_1/\partial n$ is the normal gradient of $C_1$. At the remaining boundaries, a non-flux condition is considered.

In a previous study [7], it was provided a detailed description about how the application of electric pulses affects the vessel wall permeability ($k_v$), vessel radius ($r_v$), cell membrane permeability ($k_1$), interstitial diffusion coefficient ($D$), and geometric-dependent properties of the tissue ($\delta_1$, $\delta_2$ and $\alpha$). In summary, the following was implemented:

- For the electro-permeabilization of the vessel wall and cell membrane, a function for the transient degree of electroporation ($DOE_R$) is considered [11, 33]:

$$DOE_R\left(E, t_p\right) = DOE(E).\left[(1 - DIE(E)).e^{-\frac{t_p}{\tau_i}} + DIE(E)\right] \tag{5}$$

With the electric field strength-dependent functions $DOE(E)$ and $DIE(E)$ as given by:

$$DOE(E) = \frac{\sigma(E) - \sigma_{min}}{\sigma_{max} - \sigma_{min}} \tag{6}$$

$$DIE(E) = \begin{cases} 0 & when\ E < E_{irrev} \\ \dfrac{\sigma(E) - \sigma_{irrev}}{\sigma_{max} - \sigma_{irrev}} & when\ E \geq E_{irrev} \end{cases} \tag{7}$$

where $E$, $E_{irrev}$, $t_p$ and $\tau_i$ stand for electric field magnitude, irreversible threshold of the field magnitude, time elapsed from last pulse application, and decay time coefficient for the permeability, whereas $\sigma(E)$, $\sigma_{min}$, $\sigma_{max}$ and $\sigma_{irrev}$ are the effective, minimum, maximum and irreversible electrical conductivity, respectively. For a given value of $E$, the effective electrical conductivity can be computed using a sigmoidal-type function [34]:

$$\sigma(E) = \frac{\sigma_{max} - \sigma_{min}}{1 + \alpha^*.e^{-\frac{E-a^*}{b^*}}} + \sigma_{min} \tag{8}$$

where the fitting parameters $a^*$ and $b^*$ are computed in terms of the reversible threshold ($E_{rev}$) and irreversible threshold ($E_{irrev}$) of the electric field magnitude:

$$a^* = (E_{rev} + E_{irrev})/2 \tag{9}$$

$$b^* = (E_{irrev} - E_{rev})/\beta^* \tag{10}$$

The permeabilization state of the cell membrane and vessel wall is given by:

$$k_i(E, t_p) = DOE_R(E, t_p).(k_{i,max} - k_{i,min}) + k_{i,min} \tag{11}$$

where $k_{i,max}$ and $k_{i,min}$ are the maximum and minimum values of the permeability, with $i$ = [1, v] in such way that subindex "1" stands for the cell membrane, whereas subindex "v" represents the vessel wall.

- For the vessel vasoconstriction, the mathematical model proposed in [7] is also considered, where the normalized vessel radius, $\hat{r}$, as defined by the quotient between the current and the original vessel radius, i.e., $\hat{r} = r_v/r_{v0}$, is estimated using a double-exponential function:

$$\hat{r}(E, t_p) = [(1 - \hat{r}_{min}).e^{-m_r.E.e^{-t_p/\tau_r}} + \hat{r}_{min}] \tag{12}$$

where $\hat{r}$, $\hat{r}_{min}$, $m_r$ and $\tau_r$ stand for the normalized ratio, minimum normalized ratio, coefficient of vasoconstriction, and decay time coefficient of vasoconstriction, respectively.

- Assuming that the motion of the tissue-vessel wall interface occurs in the radial direction and is uniform, that the opposite boundary remains fixed, and the function of the interface velocity does not depend on the space, the radial velocity ($u_r$) at any point along the tissue can be calculated following the approach used by [35, 36] for cell migration in changing tissue domains, obtaining the following expression:

$$u_r(r, E, t) = \left(\frac{r - R_{cord}}{r_v - R_{cord}}\right) \cdot \frac{dr_v(E, t)}{dt}, \text{for } r_v \leq r \leq R_{cord} \tag{13}$$

where $r$, $r_v$ and $R_{cord}$ stand for the radial coordinate, current vessel radius and cord radius, respectively, whereas $dr_v/dt$ is the interface velocity, which in turns comes from the time derivative of Eq (12).

The iterative *GMAPS* scheme developed in [7] is implemented here. The principal characteristics of this scheme are:

- A fully implicit formulation is used for the time discretization.

- Scale factors $S_L$, $S_T$ and $S_C$ are used for the spatial domain, time and concentration to improve the conditioning of the final system.

- A Gauss-Seidel iterative scheme is used for solving the concentration fields ($C_1$, $C_2$ and $C_3$) at each time instant considering a relative error of $1x10^{-6}$.

- Under-relaxation is applied to update the concentration fields during the iterative process.

- An advection/convection model is used to calculate the concentration profile of drug along the bloodstream, $C_v$, which in turns depends on the pharmacokinetic profile of drug (*PK*), flow velocity in vessel ($\lambda$), inlet blood pressure ($p_{inl}$), plasma viscosity ($\mu$), permeability coefficient of vessel wall ($k_{vc}$), and radius of blood vessel ($r_v$). The equation is given by:

$$\frac{dC_v}{dt} = -\lambda(z)\frac{dC_v(z,t)}{dz} + k_v \cdot (C_1(r_v, z, t) - C_v(z, t)) \tag{14}$$

Non-dimensionalization, time implicit scheme, and backward scheme for the spatial gradients can be applied in Eq (14) to get a recurrence relationship for the drug concentration profile, $C_v$. This is shown in [7].

- To reduce the computational cost, vasoconstriction effect is not considered when the relative reduction of the vessel radius is above 90%. Otherwise, this effect is deemed by recalculating the mesh of the physical domain, the geometry-dependent matrices of the *GMAPS* scheme, and the geometry-dependent properties of the tissue ($\delta_1$, $\delta_2$ and $\alpha$).

- The flow absorption through the vessel wall, which in turns leads to the change of the blood velocity along the vessel ($\lambda$), is captured using a model for flow through porous circular tubes [37], which is given in terms of the plasma viscosity ($\mu$), permeability coefficient of vessel wall ($k_{vc}$), inlet blood velocity ($\lambda_{inl}$), inlet flow rate ($Q_{inl}$), inlet blood pressure ($p_{inl}$) and permeate pressure ($p_p$). Accordingly, the blood velocity, $\lambda(z)$, can be computed from the following formulae:

$$\lambda(z) = \lambda_{inl} - \frac{Q_{leak}(z)}{\pi r_v^2} \tag{15}$$

where $\lambda_{inl}$ is the inlet blood velocity, and $Q_{leak}(z)$ is the leakage flow rate through the vessel

wall as computed by:

$$Q_{leak} = 2\pi r_v k_{vc.} \left[ (p_{inl} - p_p)z - \frac{A^*}{\vartheta^*} \left( e^{\vartheta^* z} + e^{-\vartheta^* z} - 2 \right) - (p_{inl} - p_p) \left( z - \frac{e^{\vartheta^* z} - e^{-\vartheta^* z}}{2\vartheta^*} \right) \right] \quad (16)$$

With coefficients $\vartheta^*$ and $A^*$ given by the following equations:

$$\vartheta^* = \sqrt{\frac{16\mu k_{vc}}{r_v^3}} \quad (17)$$

$$A^* = \frac{1}{2} \frac{8\mu Q_{inl}}{\pi r_v^4 \vartheta^*} \quad (18)$$

- It is important to emphasize that an accuracy and convergence analysis of the *GMAPS* code employed here was carried out in [7]. The number of collocation points ($N_{points}$), time step size ($\Delta t$) and under-relaxation factor ($\omega$) selected there are used here as well.

Fig 3 depicts a simplified flowchart of the *GMAPS* scheme employed in the present study. A more detailed explanation of this flowchart can be found in [7].

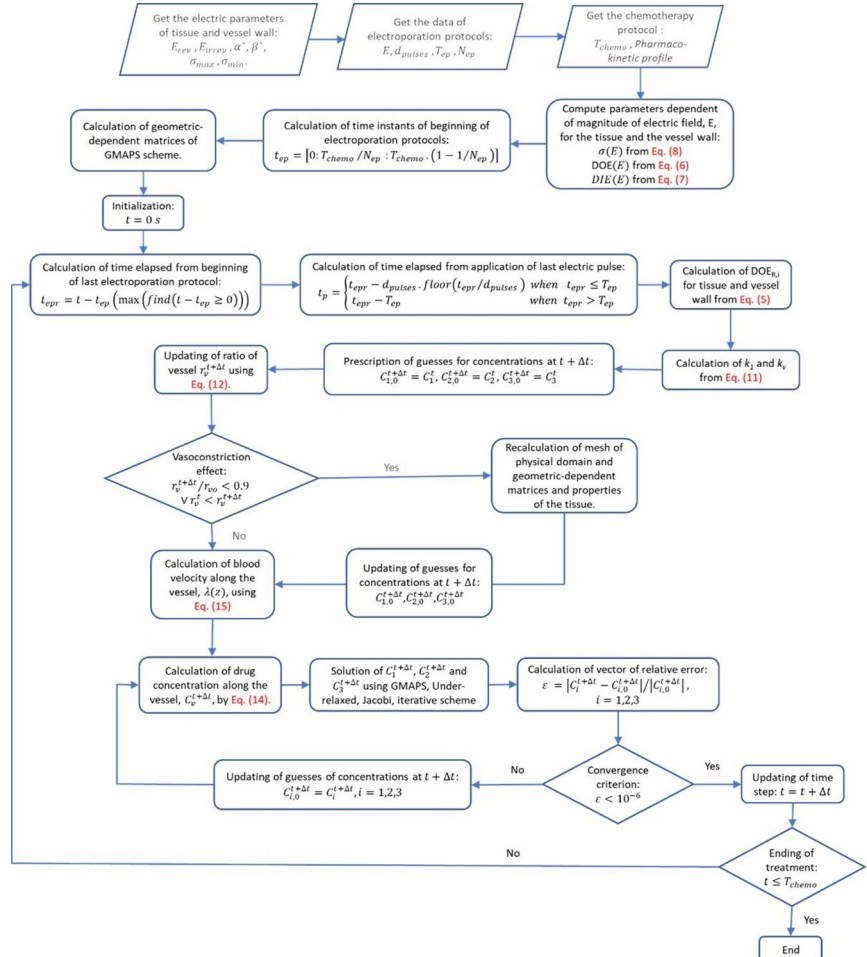

**Fig 3. Simplified flowchart of the GMAPS scheme.**

It is worth mentioning that when simulations are conducted at the macroscopic scale (tissue domain of order of millimeters), the electric field magnitude, $E$, shall be considered as variable and its distribution highly depends on the electrode positioning [11, 31–33]. On the other hand, for simulations at the microscopic scale (Representative Unitary Cell), as the ones considered in the present work, bearing in mind that the characteristic length of the tumor cord domain is usually from one to three orders of magnitude smaller than such of the tissue domain, a constant electric field magnitude, $E$, can be supposed, and electrode positioning is not relevant in the simulations [7, 9, 10, 17, 30].

## 3. Boolean modeling

Boolean modeling has proved to be a valuable instrument for the examination of the intricate dynamic involved in biological systems [38–41]. In cancer treatments, this tool can be used to elucidate the interactions between the mechanisms implicated in the systemic drug administration and drug response of cancer cells [42–45]. As aforementioned, multiple transport and reaction mechanisms play an important role in the drug motion from the bloodstream to the cells, and vice versa. The interaction between these mechanisms governs the spatial and temporal change of the extracellular concentration ($C_1$), free intracellular concentration ($C_2$) and bound intracellular concentration ($C_3$). Hence, the examination of the spatio-temporal behavior of the ratios $C_2/C_1$, $C_3/C_1$ and $C_3/C_2$ allows determining how is the interaction between the mechanisms considered here (*TBV*, *TVT+ or EV*, *TVT– or LD*, *ECT+*, *ECT–*, *TMT+ or IN*, *TMT– or EX*, *AS*, *DIS*) as the electro-chemotherapeutic treatment evolves, and how this is modified with the electric field magnitude ($E$), pharmacokinetic profile (*PK*) and inlet blood velocity ($\lambda_{inl}$).

### 3.1 Physical significance of ratios $C_2/C_1$ and $C_3/C_1$ at a given time instant

In the context of diffusion-dominated transmembrane transport (*TMT*), the drug transport direction across the cellular membrane is principally determined by the concentration gradient between $C_1$ and $C_2$, as shown in Eqs 1 and 2. It is worth noticing that the bound drug cannot escape into the extracellular space. Considering this, the net rate of *TMT* at a given time instant and *RUC* of the tissue domain is related with the ratio $C_2/C_1$, in such a way that when $C_2/C_1<1$ the drug concentration gradient is inward the cell, leading to the drug internalization, whereas $C_2/C_1>1$ implies a reverse concentration gradient, occurring drug externalization. In the particular case of $C_2/C_1 = 1$, partial species equilibrium is reached. Regarding the drug association in the intracellular space, it is worth-mentioning that this is constrained by the binding site concentration, $C_0$, in such a way that when $C_3$ reaches the value of $C_0$, further drug association is not possible. The logical propositions and implications for the ratios $C_2/C_1$ and $C_3/C_1$, as well as their physical meaning, are shown in Table 1.

**Table 1. Logical propositions and implications for $C_2/C_1$ and $C_3/C_1$ at each time instant and for any point along the tissue domain.**

| Logical proposition | | Logical implication | |
|---|---|---|---|
| **Statement** | **Physical Meaning** | **Statement** | **Logical meaning** |
| $P_1$: $C_2/C_1<1$ | Concentration gradient is inward the cell | $P_1 \leftrightarrow \exists IN$ | $P_1$ is sufficient and necessary condition for existence of drug internalization. |
| $P_2$: $C_2/C_1>1$ | Concentration gradient is outward the cell | $P_2 \leftrightarrow \exists EX$ | $P_2$ is sufficient and necessary condition for existence of drug externalization. |
| $P_3$: $C_2/C_1 = 1$ | Partial equilibrium of extracellular and free intracellular concentrations. | $P_3 \leftrightarrow \nexists EX \land \nexists IN$ | $P_3$ is sufficient and necessary condition for the partial equilibrium of species concentration between $C_1$ and $C_2$. |
| $P_4$: $C_3/C_1 = C_0/C_1$ | The binding sites concentration, $C_0$, has been reached by $C_3$. | $P_4 \rightarrow \nexists AS$ | $P_4$ is sufficient condition to inhibit the association of $C_2$ into $C_3$. |

## 3.2 Influence of transport and reaction mechanisms on the evolution of ratios $C_2/C_1$, $C_3/C_1$ and $C_3/C_2$

The transport and reaction mechanisms discussed above have varying effects on the spatio-temporal changes of $C_1$, $C_2$ and $C_3$. Accordingly, the extracellular concentration ($C_1$) rises at the tissue-vessel wall interface when positive transvascular transport ($TVT+$) or extravasation ($EV$) occurs. Additionally, $C_1$ increases at any location of the interstitial space as a result of negative transmembrane transport ($TMT-$) or cellular externalization ($EX$), and/or of positive extracellular transport ($ECT+$). As reasonable, $C_1$ exhibits a reduction in response to opposing mechanisms, namely, lymphatic drainage ($LD$), cellular internalization ($IN$) and negative extracellular transport ($ECT-$). On the other hand, the process of cellular internalization ($IN$ or $TMT+$) and drug dissociation ($DIS$) results in an increment of the free intracellular concentration ($C_2$). Conversely, the cellular externalization ($EX$ or $TMT-$) and drug association ($AS$) brings about an opposing influence on $C_2$. With respect to the bound intracellular concentration ($C_3$), it increases solely through drug association ($AS$) and decreases through drug dissociation ($DIS$). Table 2 presents a summary of the logical propositions and implications that describe the impact of the transport and reaction mechanisms on the concentrations changes ($C_1$, $C_2$ and $C_3$) and their corresponding ratios ($C_2/C_1$, $C_3/C_1$ and $C_3/C_2$). In general, an increase in $C_2$ and a decrease in $C_1$ are sufficient but not necessary conditions for an increase in the $C_2/C_1$ ratio. Similar premises can be applied to the ratios $C_3/C_1$ and $C_3/C_2$. Some logical propositions of Table 2 involve the comparison between $C_1$, $C_2$, $C_3$, and their respective ratios at two times instants identified by $t_i$ and $t_{i+1}$. It is important to emphasize that in a given Representative Unitary Cell ($RUC$), extravasation and drainage, positive and negative extracellular transport, cellular internalization and externalization, and drug association and dissociation, can be present simultaneously, but the predominant mechanisms inside each $RUC$ are the ones considered in the present work. Accordingly, from a Boolean viewpoint, $EV$ (extravasation) and $LD$ (lymphatic drainage), $ECT+$ (positive extracellular transport) and $ECT-$ (negative extracellular transport), $IN$ (internalization) and $EX$ (externalization), and $AS$ (association) and $DIS$ (dissociation) are considered mutually exclusive each other in the $RUC$.

## 3.3 Boolean model for the interaction of transport and reaction mechanisms

In this subsection, the logical inference rules are applied to deduce how the time behavior of $C_2/C_1$, $C_3/C_1$ and $C_3/C_2$ allows revealing the interaction of the reaction and transport mechanisms ($RTMs$) involved in the systemic drug delivery. For instance, let us consider the logical statement $P_7 \bigwedge \neg P_5 \to P_{15}$ presented in Table 2, which means that the increment of $C_3$ and the decrease of $C_1$ between time instants $t_i$ and $t_{i+1}$ is sufficient but not necessary condition for the increase of the ratio $C_3/C_1$. Applying inference rules, this statement can be transformed as follows:

$$P_7 \wedge \neg P_5 \to P_{15} \tag{19}$$

$$\therefore \neg P_{15} \to \neg(P_7 \wedge \neg P_5), \tag{20}$$

by transposition law

$$\therefore \neg P_{15} \to (\neg P_7 \vee P_5), \tag{21}$$

**Table 2. Logical propositions and implications for the influence of the transport and reaction mechanisms on the concentration fields.**

| Logical propositions | | Logical implication | |
|---|---|---|---|
| Statement | Physical meaning | Statement | Logical meaning |
| $P_5$: $C_1(t_i) \leq C_1(t_{i+1})$ | Extracellular concentration increases or remains equal between time instants $t_i$ and $t_{i+1}$ | $P_{10}\Delta P_{11} \leftrightarrow P_5$ | $P_{10}$ or $P_{11}$ (not both) is sufficient and necessary condition for $P_5$. |
| $P_6$: $C_2(t_i) \leq C_2(t_{i+1})$ | Free intracellular concentration increases or remains equal between time instants $t_i$ and $t_{i+1}$ | $P_8\Delta P_9 \leftrightarrow P_6$ | $P_8$ or $P_9$ (not both) is sufficient and necessary condition for $P_6$. |
| $P_7$: $C_3(t_i) \leq C_3(t_{i+1})$ | Bound intracellular concentration increases or remains equal between time instants $t_i$ and $t_{i+1}$ | $P_{12} \leftrightarrow P_7$ | $P_{12}$ is sufficient and necessary condition for $P_7$. |
| $P_8$: $IN \geq AS$ | Internalization rate imposes over or equates the association rate. | $P_6 \bigwedge \neg P_5 \rightarrow P_{14}$ | The increment of $C_2$ and the decrease of $C_1$ between $t_i$ and $t_{i+1}$ is sufficient but not necessary condition for the increase of the ratio $C_2/C_1$. |
| $P_9$: $EX \leq DIS$ | Dissociation rate imposes over or equates the externalization rate | $P_7 \bigwedge \neg P_5 \rightarrow P_{15}$ | The increment of $C_3$ and the decrease of $C_1$ between $t_i$ and $t_{i+1}$ is sufficient but not necessary condition for the increase of the ratio $C_3/C_1$. |
| $P_{10}$: $EX \geq ECT-$ | Externalization rate imposes over or equates the rate of negative extracellular transport. | $P_7 \bigwedge \neg P_6 \rightarrow P_{16}$ | The increment of $C_3$ and the decrease of $C_2$ between $t_i$ and $t_{i+1}$ is sufficient but not necessary condition for the increase of the ratio $C_3/C_2$. |
| $P_{11}$: $IN \leq ECT+$ | Rate of positive extracellular transport imposes over or equates the internalization rate. | | |
| $P_{12}$: $\exists AS$ | There exists association of $C_2$ into $C_3$ | | |
| $P_{13}$: $\exists DIS$ | There exists dissociation of $C_3$ into $C_2$ | | |
| $P_{14}$: $\frac{C_2}{C_1}(t_i) \leq \frac{C_2}{C_1}(t_{i+1})$ | The ratio $C_2/C_1$ increases or remains equal between time instants $t_i$ and $t_{i+1}$ | | |
| $P_{15}$: $\frac{C_3}{C_1}(t_i) \leq \frac{C_3}{C_1}(t_{i+1})$ | The ratio $C_3/C_1$ increases or remains equal between time instants $t_i$ and $t_{i+1}$ | | |
| $P_{16}$: $\frac{C_3}{C_2}(t_i) \leq \frac{C_3}{C_2}(t_{i+1})$ | The ratio $C_3/C_2$ increases or remains equal between time instants $t_i$ and $t_{i+1}$ | | |

by De Morgan and double-negation laws

$$\therefore \neg P_{15} \rightarrow [\neg P_{12} \vee (P_{10}\Delta P_{11})], \tag{22}$$

by second law of constructive dilemma, biconditional and transposition law, considering the statements $P_{12} \leftrightarrow P_7$ and $P_{10}\Delta P_{11} \leftrightarrow P_5$ from Table 2.

Similar expressions can be deduced for the time behavior of $C_2/C_1$ and $C_3/C_2$. For instance, for the time increase of $C_2/C_1$ and time decrease of $C_3/C_2$, the same inference analysis for the logical statements $\neg P_6 \bigwedge P_5 \rightarrow \neg P_{14}$ and $P_7 \bigwedge \neg P_6 \rightarrow P_{16}$ (see Table 2) results in the following expressions:

$$P_{14} \rightarrow [(P_8\Delta P_9) \vee \neg(P_{10}\Delta P_{11})] \tag{23}$$

$$\neg P_{16} \rightarrow [\neg P_{12} \vee (P_8\Delta P_9)] \tag{24}$$

The simultaneous occurrence of $P_{14}$, $\neg P_{15}$ and $\neg P_{16}$, which corresponds to the increment of the ratio $C_2/C_1$, and decrease of the ratios $C_3/C_1$ and $C_3/C_2$, between the time instants $t_i$ and $t_{i+1}$, can be represented as:

$$P_{14} \wedge \neg P_{15} \wedge \neg P_{16} \tag{25}$$

Using inference rules, expression (25) can be developed as follows:

$$\therefore [(P_8\Delta P_9) \vee \neg(P_{10}\Delta P_{11})] \wedge [\neg P_{12} \vee (P_{10}\Delta P_{11})] \wedge [\neg P_{12} \vee (P_8\Delta P_9)], \tag{26}$$

by Modus Ponendo Ponens considering the statements (22) to (24).

$$\therefore \; [(P_8 \Delta P_9) \vee \neg(P_{10}\Delta P_{11})] \wedge \{\neg P_{12} \vee [(P_8\Delta P_9) \wedge (P_{10}\Delta P_{11})]\}, \qquad (27)$$

by distributivity law.

By replacing the logical propositions $P_8$ to $P_{12}$ defined in Table 2 on the expression (27), they are obtained all possible combinations of the transport and reaction mechanisms considered here that bring about the increase of the ratio $C_2/C_1$, and the decrease of the ratios $C_3/C_1$ and $C_3/C_2$, between two time instants ($t_i$ and $t_{i+1}$). However, it is possible to considerably simplify the expression (27) based on the time behavior of the intracellular bound concentration ($C_3$) and extracellular concentration ($C_1$), as well as on the value of the ratio $C_2/C_1$ at each time instant. In this point, it is important to remind that *IN* and *EX*, *AS* and *DIS*, *ECT+* and *ECT* − are considered here as mutually exclusive mechanisms. Accordingly, net drug association *(AS)* and dissociation (*DIS*) cannot take place at the same time at a given point of the tissue domain in the framework of the present analysis. Considering this and the fact that in the present model the drug association (*AS*) is the only mechanism that brings about a growth of the bound intracellular concentration ($C_3$) between two time instants ($t_i$ and $t_{i+1}$), when $C_3(t_i) < C_3(t_{i+1})$ it can be inferred that association (*AS*) imposes over dissociation (*DIS*) between $t_i$ and $t_{i+1}$, signifying that $P_{12} = 1$ (true), and $P_8\Delta P_9 = P_8$ by definition of exclusive disjunction and setting $P_9 = 0$ (false) since net dissociation does not exist. Under such condition, expressions (27) can be written as:

$$\therefore \; [P_8 \vee \neg(P_{10}\Delta P_{11})] \wedge \{0 \vee [P_8 \wedge (P_{10}\Delta P_{11})]\} \qquad (28)$$

$$\therefore \; [P_8 \vee \neg(P_{10}\Delta P_{11})] \wedge [P_8 \wedge (P_{10}\Delta P_{11})], \qquad (29)$$

by disjunctive identity

$$\therefore \; \{[P_8 \vee \neg(P_{10}\Delta P_{11})] \wedge P_8\} \wedge (P_{10}\Delta P_{11}), \qquad (30)$$

by association law

$$\therefore \; \{P_8 \vee [\neg(P_{10}\Delta P_{11}) \wedge 0]\} \wedge (P_{10}\Delta P_{11}), \qquad (31)$$

by distributivity law

$$\therefore \; P_8 \wedge (P_{10}\Delta P_{11}), \qquad (32)$$

by disjunction and conjunction definition.

Now, the statement (32) can still be simplified considering the ratio $C_2/C_1$. As formerly shown in Table 1, the direction of the net transmembrane transport (*TMT*) at a given time instant is related with the ratio $C_2/C_1$, in such a way that when $C_2/C_1 < 1$, net internalization occurs ($\exists IN$, $\nexists EX$), leading to $P_{10} = 0$ (false). By definition of exclusive disjunction, expression (32) can then be written as:

$$\therefore \; P_8 \wedge P_{11} \qquad (33)$$

By replacing the logical propositions $P_8$ and $P_{11}$ defined in Table 2 in the conditional statement (33), the following is obtained:

$$\therefore \; (IN \geq AS) \wedge (IN \leq ECT+) \qquad (34)$$

Assuming a non-equilibrium condition (*IN≠AS*, *IN≠ECT+*), this means that when net internalization *(IN)* and association *(AS)* occurs between two time instants ($t_i$ and $t_{i+1}$), the increment of the ratio $C_2/C_1$, and decrease of $C_3/C_1$ and $C_3/C_2$, indicates that the

internalization rate ($IN$) imposes over the association rate ($AS$) but it is smaller than the extra-cellular transport rate ($ECT+$). Following the procedure exposed above, the logical statements for the remaining scenarios of the time behavior of ratios $C_2/C_1$, $C_3/C_1$ and $C_3/C_2$ can be obtained. The statements for all possible combinations of $C_2/C_1$, $C_3/C_1$ and $C_3/C_2$ are deduced and summarized in Table 3, where *"I"* stands for the time increase of the concentration ratio and *"D"* for the time decrease. The Boolean scheme that summarizes the results obtained from the Boolean analysis is shown in Figs 4 and 5. In both Table 3 and Figs 4 and 5, *"Tautology"* means that the scenario resulting from the time change of $C_2/C_1$, $C_3/C_1$ and $C_3/C_2$, and the presence of drug association/dissociation and internalization/externalization is always true, and do not allow concluding anything about the interaction between the transport and reaction mechanisms considered here. For instance, when $C_2/C_1$ increases, $C_3/C_1$ decreases, $C_3/C_2$ decreases, and drug dissociation and internalization take place, the existing reaction and transport mechanisms ($RTMs$) can interact in any way. On the other hand, *"Contradiction"* means that the corresponding scenario is not physically consistent. For example, when the time behavior of the ratios $C_2/C_1$, $C_3/C_1$ and $C_3/C_2$ remains as in the last example, but drug association takes place, the cellular externalization is not physically possible.

## 4. Data and simulation campaign

As formerly mentioned, this work is aimed to study the influence of the electric field magnitude ($E$), type of pharmacokinetic profile ($PK$) and inlet blood velocity ($\lambda_{inl}$) on the interaction between the $RTMs$ during the electro-chemotherapeutic treatment using the Boolean model previously presented. Firstly, the concentration fields ($C_1$, $C_2$, $C_3$) are obtained from the solution of the tumor cord model defined by Eqs (1)–(3) under the boundary conditions shown in Fig 2. The influence of the pulses' characteristics on the vessel wall permeability ($k_v$), vessel radius ($r_v$), cell membrane permeability ($k_1$), interstitial diffusion coefficient ($D$), and geometric-dependent properties of the tissue ($\delta_1$, $\delta_2$ and $\alpha$) are considered from Eqs (5)–(13). On the other hand, the spatio-temporal evolution of the drug concentration along the bloodstream ($C_v$) is obtained by solving Eq (14), whereas the change of the blood velocity ($\lambda_{inl}$) along the vessel is achieved from Eq (15), where the leakage flow rate through the vessel wall ($Q_{leak}$) is computed from Eq (16). The $GMAPS$ iterative scheme represented in Fig 3 is used to solve these coupled equations. More details about the numerical development, calibration and validation of this numerical scheme can be found in [7]. Once the concentration fields are obtained at several points and time instants, ratios $C_2/C_1$, $C_3/C_1$ and $C_3/C_2$ can be computed. The Boolean model presented in Section 3 allows determining the presence and interaction of the different reaction and transport mechanisms ($RTMs$) considered here by defining logical propositions and implications related with the spatio-temporal behavior of the concentration ratios ($C_2/C_1$, $C_3/C_1$, $C_3/C_2$), Tables 1 and 2, and applying Boolean algebra to deduct equivalent logical statements for all possible combinations of these ratios, Table 3. Software Matlab$^{TM}$ was used for the algorithm development and plots generation.

The computational experiments consider 3 levels for $E$ ($0\ kV/m$, $46\ kV/m$, $70\ kV/m$), 2 levels for $PK$ (One-short Tri-exponential, Mono-exponential) and 3 levels for $\lambda_{inl}$ ($1x10^{-4}m/s$, $1x10^{-3}m/s$, $1x10^{-2}m/s$). The other parameters are fixed. Both variable and fixed parameters are shown in Table 4. In this table, $T_{chemo}$ represents the simulation time of electro-chemotherapy.

The time evolution of drug concentration in the bloodstream ($C_v$) depends on the pharmacokinetic profile considered for the drug supply. For the profiles considered here, the Equations are [9, 22, 58]:

**Table 3. Equivalent logical statements for possible scenarios of $C_2/C_1$, $C_3/C_1$ and $C_3/C_2$.**

| Supposed scenario for time change of ratios | | | Equivalent logical statement | |
|---|---|---|---|---|
| $C_2/C_1$ | $C_3/C_1$ | $C_3/C_2$ | | |
| I | I | I | ***General statement and inference analysis:*** $P_{14} \wedge P_{15} \wedge P_{16}$ <br> $\therefore [(P_8 \Delta P_9) \vee \neg(P_{10} \Delta P_{11})] \wedge [P_{12} \vee \neg(P_{10} \Delta P_{11})] \wedge [P_{12} \vee \neg(P_8 \Delta P_9)]$, by Modus Ponendo Ponens <br> $\therefore [(P_8 \Delta P_9) \vee \neg(P_{10} \Delta P_{11})] \wedge \{P_{12} \vee [\neg(P_{10} \Delta P_{11}) \wedge \neg(P_8 \Delta P_9)]\}$, by distributivity law | |
| | | | ***Drug association occurs*** ($\exists$***AS***): $P_{12} = 1$ (True) <br> $P_8 \Delta P_9 = P_8$, by exclusive disjunction <br> $\therefore [P_8 \vee \neg(P_{10} \Delta P_{11})] \wedge \{1 \vee [\neg(P_{10} \Delta P_{11}) \wedge \neg P_8]\}$ <br> $\therefore [P_8 \vee \neg(P_{10} \Delta P_{11})] \wedge 1$, by disjunction definition <br> $\therefore P_8 \vee \neg(P_{10} \Delta P_{11})$, by identity law of conjunction | ***Internalization occurs*** ($\exists$***IN***): <br> $(IN \geq AS) \vee (IN \geq ECT+)$ <br> *If $C_1$ increases, it leads to $(IN \geq AS)$* <br> ***Externalization occurs*** ($\exists$***EX***): $EX \leq ECT-$ <br> ***Partial equilibrium condition*** ($\not\exists$ ***EX*** $\wedge \not\exists$ ***IN***): <br> Tautology |
| | | | ***Drug dissociation occurs*** ($\exists$***DIS***): $P_{12} = 0$ (False) <br> $P_8 \Delta P_9 = P_8$, by exclusive disjunction <br> $\therefore [P_9 \vee \neg(P_{10} \Delta P_{11})] \wedge \{0 \vee [\neg(P_{10} \Delta P_{11}) \wedge \neg P_9]\}$ <br> $\therefore [P_9 \vee \neg(P_{10} \Delta P_{11})] \wedge [\neg(P_{10} \Delta P_{11}) \wedge \neg P_9]$, by identity law of disjunction <br> $\therefore \{[P_9 \vee \neg(P_{10} \Delta P_{11})] \wedge \neg(P_{10} \Delta P_{11})\} \wedge \neg P_9$, by association law <br> $\therefore \{\neg(P_{10} \Delta P_{11}) \vee [P_9 \wedge 0]\} \wedge \neg P_9$, by distributive law <br> $\therefore \{\neg(P_{10} \Delta P_{11}) \vee 0\} \wedge \neg P_9$, by conjunction definition <br> $\therefore \neg(P_{10} \Delta P_{11}) \wedge \neg P_9$, by disjunction definition | ***Internalization occurs*** ($\exists$***IN***): <br> *Contradiction* <br> ***Externalization occurs*** ($\exists$***EX***): <br> $(EX \leq ECT-) \wedge (EX \geq DIS)$ <br> ***Partial equilibrium condition*** ($\not\exists$ ***EX*** $\wedge \not\exists$ ***IN***): <br> *Contradiction* |
| I | I | D | ***General statement and inference analysis***: <br> $P_{14} \wedge P_{15} \wedge \neg P_{16}$ <br> $\therefore [(P_8 \Delta P_9) \vee \neg(P_{10} \Delta P_{11})] \wedge [P_{12} \vee \neg(P_{10} \Delta P_{11})] \wedge [\neg P_{12} \vee (P_8 \Delta P_9)]$, by Modus Ponendo Ponens <br> $\therefore \{\neg(P_{10} \Delta P_{11}) \vee [(P_8 \Delta P_9) \wedge P_{12}]\} \wedge [\neg P_{12} \vee (P_8 \Delta P_9)]$, by distributive and association law | |
| | | | ***Drug association occurs*** ($\exists$***AS***): $P_{12} = 1$ (True) <br> $P_8 \Delta P_9 = P_8$, by exclusive disjunction <br> $\therefore \{\neg(P_{10} \Delta P_{11}) \vee [P_8 \wedge 1]\} \wedge [0 \vee P_8]$ <br> $\therefore \{\neg(P_{10} \Delta P_{11}) \vee P_8\} \wedge P_8$, by identity law of conjunction and disjunction <br> $\therefore P_8 \vee [\neg(P_{10} \Delta P_{11}) \wedge 0]$, by distributivity law <br> $\therefore P_8$, by disjunction and conjunction definition | ***Internalization occurs*** ($\exists$***IN***): <br> $IN \geq AS$ <br> ***Externalization occurs*** ($\exists$***EX***): <br> *Contradiction* <br> ***Partial equilibrium condition*** ($\not\exists$ ***EX*** $\wedge \not\exists$ ***IN***): <br> *Contradiction* |
| | | | ***Drug dissociation occurs*** ($\exists$***DIS***): $P_{12} = 0$ (False) <br> $P_8 \Delta P_9 = P_8$, by exclusive disjunction <br> $\therefore \{\neg(P_{10} \Delta P_{11}) \vee [P_9 \wedge 0]\} \wedge [1 \vee P_9]$ <br> $\therefore \{\neg(P_{10} \Delta P_{11}) \vee 0\} \wedge 1$, by conjunction and disjunction definition <br> $\therefore \neg(P_{10} \Delta P_{11})$, identity law of conjunction and disjunction | ***Internalization occurs*** ($\exists$***IN***): <br> $IN \geq ECT+$ <br> ***Externalization occurs*** ($\exists$***EX***): <br> $EX \leq ECT-$ <br> ***Partial equilibrium condition*** ($\not\exists$ ***EX*** $\wedge \not\exists$ ***IN***): <br> *Tautology* |
| I | D | I | ***General statement and inference analysis:*** $P_{14} \wedge \neg P_{15} \wedge P_{16}$ <br> $\therefore [(P_8 \Delta P_9) \vee \neg(P_{10} \Delta P_{11})] \wedge \{[\neg P_{12} \vee (P_{10} \Delta P_{11})] \wedge [P_{12} \vee \neg(P_8 \Delta P_9)]\}$, by Modus Ponendo Ponens and association law <br> $\therefore [(P_8 \Delta P_9) \vee \neg(P_{10} \Delta P_{11})] \wedge [(P_{10} \Delta P_{11}) \vee \neg(P_8 \Delta P_9)]$, by resolution law <br> $\therefore [(P_{10} \Delta P_{11}) \rightarrow (P_8 \Delta P_9)] \wedge [(P_8 \Delta P_9) \rightarrow (P_{10} \Delta P_{11})]$, by disjunctive conditional. <br> $\therefore (P_8 \Delta P_9) \leftrightarrow (P_{10} \Delta P_{11})$, by biconditional definition | |
| | | | ***Drug association occurs*** ($\exists$***AS***): <br> $P_8 \Delta P_9 = P_8$, by exclusive disjunction <br> $\therefore P_8 \leftrightarrow (P_{10} \Delta P_{11})$ | ***Internalization occurs*** ($\exists$***IN***): <br> $(IN \geq AS) \leftrightarrow (IN \leq ECT+)$ <br> *If $C_1$ increases, it leads to $(IN \geq AS)$* <br> *If $C_1$ decreases, it leads to $(IN \leq AS)$* <br> ***Externalization occurs*** ($\exists$***EX***): $EX \leq ECT-$ <br> ***Partial equilibrium condition*** ($\not\exists$ ***EX*** $\wedge \not\exists$ ***IN***): <br> *Tautology* |
| | | | ***Drug dissociation occurs*** ($\exists$***DIS***): <br> $P_8 \Delta P_9 = P_9$, by exclusive disjunction <br> $\therefore P_9 \leftrightarrow (P_{10} \Delta P_{11})$ | ***Internalization occurs*** ($\exists$***IN***): $IN \leq ECT+$ <br> ***Externalization occurs*** ($\exists$***EX***): <br> $(EX \leq DIS) \leftrightarrow (EX \geq ECT-)$ <br> *If $C_1$ increases, it leads to $(EX \leq DIS)$* <br> *If $C_1$ decreases, it leads to $(EX \geq DIS)$* <br> ***Partial equilibrium condition*** ($\not\exists$ ***EX*** $\wedge \not\exists$ ***IN***): <br> *Contradiction* |

*(Continued)*

**Table 3.** (*Continued*)

| I | D | D | | |
|---|---|---|---|---|
| | | | ***General statement and inference analysis:***<br>$P_{14} \wedge \neg P_{15} \wedge \neg P_{16}$<br>$\therefore [(P_8 \Delta P_9) \vee \neg (P_{10} \Delta P_{11})] \wedge \{[\neg P_{12} \vee (P_{10} \Delta P_{11})] \wedge [\neg P_{12} \vee (P_8 \Delta P_9)]\}$, by Modus Ponendo Ponens and association law<br>$\therefore [(P_8 \Delta P_9) \vee \neg (P_{10} \Delta P_{11})] \wedge \{\neg P_{12} \vee [(P_8 \Delta P_9) \wedge (P_{10} \Delta P_{11})]\}$, by distributivity law | | |
| | | | ***Drug association occurs*** ($\exists AS$): $P_{12} = 1$ (True)<br>$P_8 \Delta P_9 = P_8$, by exclusive disjunction<br>$\therefore [P_8 \vee \neg (P_{10} \Delta P_{11})] \wedge \{0 \vee [P_8 \wedge (P_{10} \Delta P_{11})]\}$<br>$[P_8 \vee \neg (P_{10} \Delta P_{11})] \wedge [P_8 \wedge (P_{10} \Delta P_{11})]$, by disjunctive identity<br>$\{[P_8 \vee \neg (P_{10} \Delta P_{11})] \wedge P_8\} \wedge (P_{10} \Delta P_{11})$, by association law<br>$\{P_8 \vee [\neg (P_{10} \Delta P_{11}) \wedge 0]\} \wedge (P_{10} \Delta P_{11})$, by distributivity law<br>$P_8 \wedge (P_{10} \Delta P_{11})$, by disjunction and conjunction definition | | ***Internalization occurs*** ($\exists IN$):<br>$(IN \geq AS) \wedge (IN \leq ECT+)$<br>***Externalization occurs*** ($\exists EX$):<br>*Contradiction*<br>***Partial equilibrium condition*** ($\nexists EX \wedge \nexists IN$):<br>*Contradiction* |
| | | | ***Drug dissociation occurs*** ($\exists DIS$):<br>$P_{12} = 0$ (False)<br>$P_8 \Delta P_9 = P_9$, by exclusive disjunction<br>$\therefore [P_9 \vee \neg (P_{10} \Delta P_{11})] \wedge \{1 \vee [P_9 \wedge (P_{10} \Delta P_{11})]\}$<br>$\therefore P_9 \vee \neg (P_{10} \Delta P_{11})$, by disjunction and conjunction definition | | ***Internalization occurs*** ($\exists IN$): *Tautology*<br>***Externalization occurs*** ($\exists EX$):<br>$(EX \leq DIS) \vee (EX \leq ECT-)$<br>*If $C_1$ increases, it leads to* $(EX \leq DIS)$<br>***Partial equilibrium condition*** ($\nexists EX \wedge \nexists IN$):<br>*Tautology* |
| D | I | I | | |
| | | | ***General statement and inference analysis:*** $\neg P_{14} \wedge P_{15} \wedge P_{16}$<br>$\therefore [\neg (P_8 \Delta P_9) \vee (P_{10} \Delta P_{11})] \wedge \{[P_{12} \vee \neg (P_{10} \Delta P_{11})] \wedge [P_{12} \vee \neg (P_8 \Delta P_9)]\}$, by Modus Ponendo Ponens and association law<br>$\therefore [\neg (P_8 \Delta P_9) \vee (P_{10} \Delta P_{11})] \wedge \{P_{12} \vee [\neg (P_{10} \Delta P_{11}) \wedge \neg (P_8 \Delta P_9)]\}$, by distributivity law | | |
| | | | ***Drug association occurs*** ($\exists AS$): $P_{12} = 1$ (True)<br>$P_8 \Delta P_9 = P_8$, by exclusive disjunction<br>$\therefore [\neg P_8 \vee (P_{10} \Delta P_{11})] \wedge \{1 \vee [\neg (P_{10} \Delta P_{11}) \wedge \neg P_8]\}$<br>$\therefore [\neg P_8 \vee (P_{10} \Delta P_{11})] \wedge 1$, by disjunction definition<br>$\therefore \neg P_8 \vee (P_{10} \Delta P_{11})$, by identity law of conjunction | | ***Internalization occurs*** ($\exists IN$):<br>$(IN \leq AS) \vee (IN \leq ECT+)$<br>*If $C_1$ decreases, it leads to* $(IN \leq AS)$<br>***Externalization occurs*** ($\exists EX$): *Tautology*<br>***Partial equilibrium condition*** ($\nexists EX \wedge \nexists IN$):<br>*Tautology* |
| | | | ***Drug dissociation occurs*** ($\exists DIS$): $P_{12} = 0$ (False)<br>$P_8 \Delta P_9 = P_9$, by exclusive disjunction<br>$\therefore [\neg P_9 \vee (P_{10} \Delta P_{11})] \wedge \{0 \vee [\neg (P_{10} \Delta P_{11}) \wedge \neg P_9]\}$<br>$\therefore [\neg P_9 \vee (P_{10} \Delta P_{11})] \wedge [\neg (P_{10} \Delta P_{11}) \wedge \neg P_9]$, by identity law of disjunction<br>$\therefore \{[\neg P_9 \vee (P_{10} \Delta P_{11})] \wedge \neg P_9\} \wedge \neg (P_{10} \Delta P_{11})$, by association law<br>$\therefore \{\neg P_9 \vee [(P_{10} \Delta P_{11}) \wedge 0]\} \wedge \neg (P_{10} \Delta P_{11})$, by distributive law<br>$\therefore \{\neg P_9 \vee 0\} \wedge \neg (P_{10} \Delta P_{11})$, by conjunction definition<br>$\therefore \neg P_9 \wedge \neg (P_{10} \Delta P_{11})$, by disjunction definition | | ***Internalization occurs*** ($\exists IN$):<br>*Contradiction*<br>***Externalization occurs*** ($\exists EX$):<br>$(EX \geq DIS) \wedge (EX \leq ECT-)$<br>***Partial equilibrium condition*** ($\nexists EX \wedge \nexists IN$):<br>*Contradiction* |
| D | I | D | | |
| | | | ***General statement and inference analysis:*** $\neg P_{14} \wedge P_{15} \wedge \neg P_{16}$<br>$\therefore [\neg (P_8 \Delta P_9) \vee (P_{10} \Delta P_{11})] \wedge \{[P_{12} \vee \neg (P_{10} \Delta P_{11})] \wedge [\neg P_{12} \vee (P_8 \Delta P_9)]\}$, by Modus Ponendo Ponens and association law<br>$\therefore [\neg (P_8 \Delta P_9) \vee (P_{10} \Delta P_{11})] \wedge [(P_8 \Delta P_9) \vee \neg (P_{10} \Delta P_{11})]$, by resolution<br>$\therefore [(P_8 \Delta P_9) \rightarrow (P_{10} \Delta P_{11})] \wedge [(P_{10} \Delta P_{11}) \rightarrow (P_8 \Delta P_9)]$, by disjunctive conditional<br>$\therefore (P_8 \Delta P_9) \leftrightarrow (P_{10} \Delta P_{11})$, by biconditional definition | | |
| | | | ***Drug association occurs*** ($\exists AS$):<br>$P_8 \Delta P_9 = P_8$, by exclusive disjunction<br>$\therefore P_8 \leftrightarrow (P_{10} \Delta P_{11})$ | | ***Internalization occurs*** ($\exists IN$):<br>$(IN \geq AS) \leftrightarrow (IN \leq ECT+)$<br>*If $C_1$ increases, it leads to* $(IN \geq AS)$<br>*If $C_1$ decreases, it leads to* $(IN \leq AS)$<br>***Externalization occurs*** ($\exists EX$): $EX \leq ECT-$<br>***Partial equilibrium condition*** ($\nexists EX \wedge \nexists IN$):<br>*Tautology* |
| | | | ***Drug dissociation occurs*** ($\exists DIS$):<br>$P_8 \Delta P_9 = P_9$, by exclusive disjunction<br>$\therefore P_9 \leftrightarrow (P_{10} \Delta P_{11})$ | | ***Internalization occurs*** ($\exists IN$): $IN \leq ECT+$<br>***Externalization occurs*** ($\exists EX$):<br>$(EX \leq DIS) \leftrightarrow (EX \geq ECT-)$<br>*If $C_1$ increases, it leads to* $(EX \leq DIS)$<br>*If $C_1$ decreases, it leads to* $(EX \geq DIS)$<br>***Partial equilibrium condition*** ($\nexists EX \wedge \nexists IN$):<br>*Contradiction* |

(*Continued*)

**Table 3.** (Continued)

| D | D | I | **General statement and inference analysis:** $\neg P_{14} \wedge \neg P_{15} \wedge P_{16}$ |
|---|---|---|---|
| | | | $\therefore [\neg(P_8 \Delta P_9) \vee (P_{10} \Delta P_{11})] \wedge \{[\neg P_{12} \vee (P_{10} \Delta P_{11})] \wedge [P_{12} \vee \neg(P_8 \Delta P_9)]\}$, by Modus Ponendo Ponens and association law |
| | | | $\therefore [\neg(P_8 \Delta P_9) \vee (P_{10} \Delta P_{11})] \wedge [(P_{10} \Delta P_{11}) \vee \neg(P_8 \Delta P_9)]$, by resolution law |
| | | | $\therefore \neg(P_8 \Delta P_9) \vee (P_{10} \Delta P_{11})$, by idempotence law |

<table>
<tr>
<td colspan="2">

**Drug association occurs** ($\exists \mathbf{AS}$):
$P_8 \Delta P_9 = P_8$, by exclusive disjunction
$\therefore \neg P_8 \vee (P_{10} \Delta P_{11})$
</td>
<td colspan="2">

**Internalization occurs** ($\exists \mathbf{IN}$):
$(IN \leq AS) \vee (IN \leq ECT+)$
*If $C_1$ decreases, it leads to* $(IN \leq AS)$
**Externalization occurs** ($\exists \mathbf{EX}$): *Tautology*
**Partial equilibrium condition** ($\nexists \mathbf{EX} \wedge \nexists \mathbf{IN}$):
*Tautology*
</td>
</tr>
<tr>
<td colspan="2">

**Drug dissociation occurs** ($\exists \mathbf{DIS}$):
$P_8 \Delta P_9 = P_9$, by exclusive disjunction
$\therefore \neg P_9 \vee (P_{10} \Delta P_{11})$
</td>
<td colspan="2">

**Internalization occurs** ($\exists \mathbf{IN}$):
$IN \leq ECT+$
**Externalization occurs** ($\exists \mathbf{EX}$):
$(EX \geq DIS) \vee (EX \geq ECT-)$
*If $C_1$ decreases, it leads to* $(EX \geq DIS)$
**Partial equilibrium condition** ($\nexists \mathbf{EX} \wedge \nexists \mathbf{IN}$):
*Contradiction*
</td>
</tr>
</table>

| D | D | D | **General statement and inference analysis:** $\neg P_{14} \wedge \neg P_{15} \wedge \neg P_{16}$ |
|---|---|---|---|
| | | | $\therefore [\neg(P_8 \Delta P_9) \vee (P_{10} \Delta P_{11})] \wedge \{[\neg P_{12} \vee (P_{10} \Delta P_{11})] \wedge [\neg P_{12} \vee (P_8 \Delta P_9)]\}$, by Modus Ponendo Ponens and association law |
| | | | $\therefore [\neg(P_8 \Delta P_9) \vee (P_{10} \Delta P_{11})] \wedge \{\neg P_{12} \vee [(P_8 \Delta P_9) \wedge (P_{10} \Delta P_{11})]\}$, by distributivity law |

<table>
<tr>
<td colspan="2">

**Drug association occurs** ($\exists \mathbf{AS}$): $P_{12} = 1$ (True)
$P_8 \Delta P_9 = P_8$, by exclusive disjunction
$\therefore [\neg P_8 \vee (P_{10} \Delta P_{11})] \wedge \{0 \vee [P_8 \wedge (P_{10} \Delta P_{11})]\}$
$[\neg P_8 \vee (P_{10} \Delta P_{11})] \wedge [P_8 \wedge (P_{10} \Delta P_{11})]$, by disjunctive identity
$\{[\neg P_8 \vee (P_{10} \Delta P_{11})] \wedge (P_{10} \Delta P_{11})\} \wedge P_8$, by association law
$\{(P_{10} \Delta P_{11}) \vee [P_8 \wedge 0]\} \wedge P_8$, by distributivity law
$P_8 \wedge (P_{10} \Delta P_{11})$, by disjunction and conjunction definition
</td>
<td colspan="2">

**Internalization occurs** ($\exists \mathbf{IN}$):
$(IN \geq AS) \wedge (IN \leq ECT+)$
**Externalization occurs** ($\exists \mathbf{EX}$):
*Contradiction*
**Partial equilibrium condition** ($\nexists \mathbf{EX} \wedge \nexists \mathbf{IN}$):
*Contradiction*
</td>
</tr>
<tr>
<td colspan="2">

**Drug dissociation occurs** ($\nexists \mathbf{AS}$): $P_{12} = 0$ (False)
$P_8 \Delta P_9 = P_9$, by exclusive diffusion
$\therefore [\neg P_9 \vee (P_{10} \Delta P_{11})] \wedge \{1 \vee [P_9 \wedge (P_{10} \Delta P_{11})]\}$
$\therefore \neg P_9 \vee (P_{10} \Delta P_{11})$, by disjunction and conjunction definition
</td>
<td colspan="2">

**Internalization occurs** ($\exists \mathbf{IN}$): $IN \leq ECT+$
**Externalization occurs** ($\exists \mathbf{EX}$):
$(EX \geq DIS) \vee (EX \geq ECT-)$
*If $C_1$ decreases, it leads to* $(EX \geq DIS)$
**Partial equilibrium condition** ($\nexists \mathbf{EX} \wedge \nexists \mathbf{IN}$):
*Contradiction*
</td>
</tr>
</table>

One-short, tri-exponential injection (TPK)

$$C_v(t)|_{z=0} = \begin{cases} \dfrac{D_o}{\tau'} \left[ \dfrac{A'}{A''}\left(1 - e^{-A''t}\right) + \dfrac{B'}{B''}\left(1 - e^{-B''t}\right) + \dfrac{C'}{C''}\left(1 - e^{-C''t}\right) \right], & for\ t \leq \tau' \\[2ex] \dfrac{D_o}{\tau'} \left[ \dfrac{A'}{A''}\left(e^{A''\tau'} - 1\right)e^{-A''t} + \dfrac{B'}{B''}\left(e^{B''\tau'} - 1\right)e^{-B''t} + \dfrac{C'}{C''}\left(e^{C''\tau'} - 1\right)e^{-C''t} \right], & for\ t > \tau' \end{cases} \tag{35}$$

where $D_o = 1.1983 x 10^2\ \mu mol$, $\tau' = 180\ s$, $A' = 7.46 x 10^{-2}\ l^{-1}$, $A'' = 2.69 x 10^{-3}\ s^{-1}$, $B' = 2.49 x 10^{-3}\ l^{-1}$, $B'' = 2.83 x 10^{-4}\ s^{-1}$, $C' = 5.52 x 10^{-4}\ l^{-1}$ and $C'' = 1.18 x 10^{-5}\ s^{-1}$.

*Mono-exponential injection (MPK).*

$$C_v(t)|_{z=0} = A'.e^{-A''t} \tag{36}$$

where $A' = 50\ \mu M$ and $A'' = 0.005\ s^{-1}$.

The time behavior of *TPK* and *MPK* profiles until $t = 1h$ is shown in Fig 6. Semilog-y plots are reported for both profiles to identify the change of the exponential rates. As can be observed, the bolus IV injection of *MPK* leads to the higher drug concentration, $C_v$, at the beginning of the treatment, with a subsequent exponential decay at a constant rate. For *TPK*, a slower IV infusion with duration of $\tau' = 180\ s$ is considered, leading to a significant exponential increase of $C_v$ at the first three minutes of the treatment, after which an exponential decay,

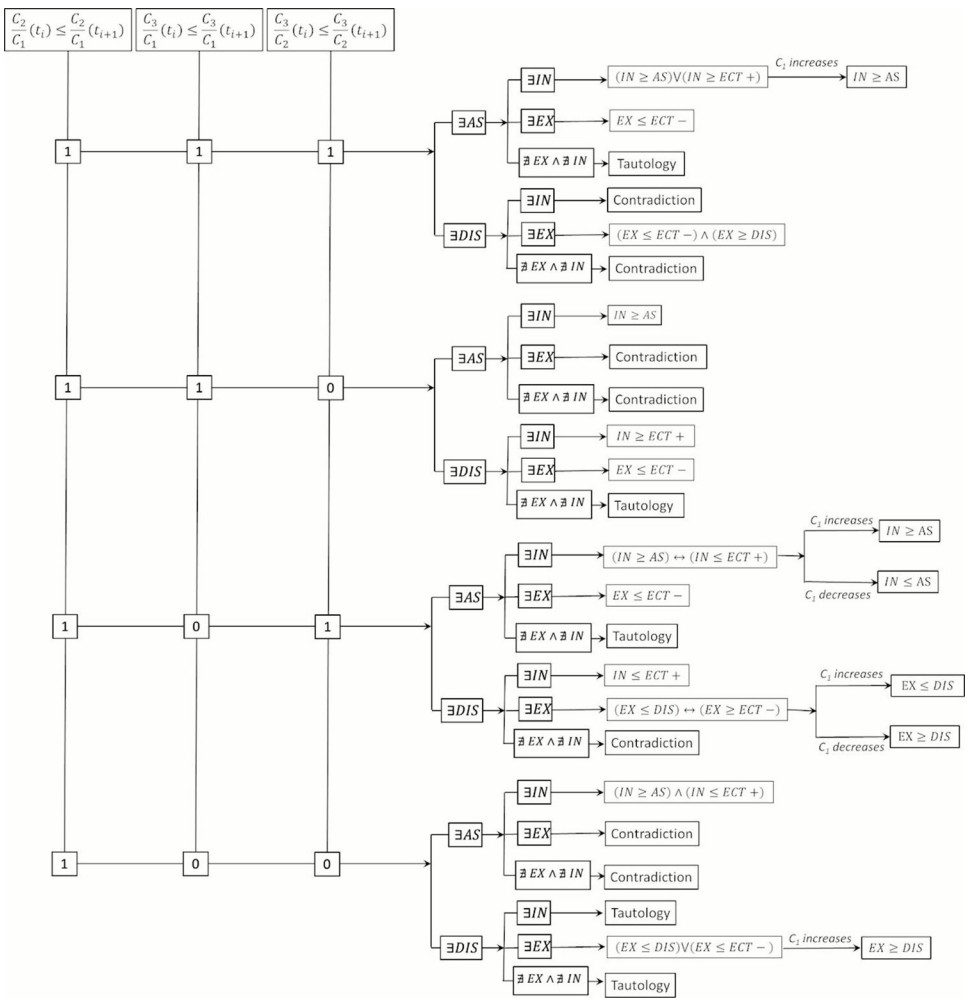

**Fig 4. Boolean schemes for the reaction and transport mechanisms (I).**

with varying rates much smaller than such of *MPK*, is obtained. It is important to emphasize that both profiles entail the same total drug exposure or Area Under the Curve, i.e,

$AUC = \int_0^\infty C_v.dt \approx 2.78 \ \mu Mh.$

For sake of clarity, a simple workflow summarizing the present numerical study is represented in Fig 7.

## 5. Results and discussion

In the present section, the influence of the electric field magnitude ($E$), type of pharmacokinetic profile ($PK$) and inlet blood velocity ($\lambda_{inl}$) on the contour plots of $RTMs$, $C_2/C_1$ and $C_3$ is analyzed. For sake of simplicity, boundaries of the tissues are named as shown in Fig 8, namely, *TVWI* represents the tissue-vessel wall interface where drug extravasation takes place, *FBV* stands for the farthest boundary from the vessel where a non-flux condition is assumed, *FIB* represents the front inlet boundary, and *ROB* is the rear outlet boundary. To limit the manuscript length, some of the contour plots corresponding to the results of the following subsections are reported as supplementary material.

The contour plots of the *RTMs* considers both equilibrium (equal rates of *RTMs*) and non-equilibrium (different rates of *RTMs*) conditions, but it is worth mentioning that the last

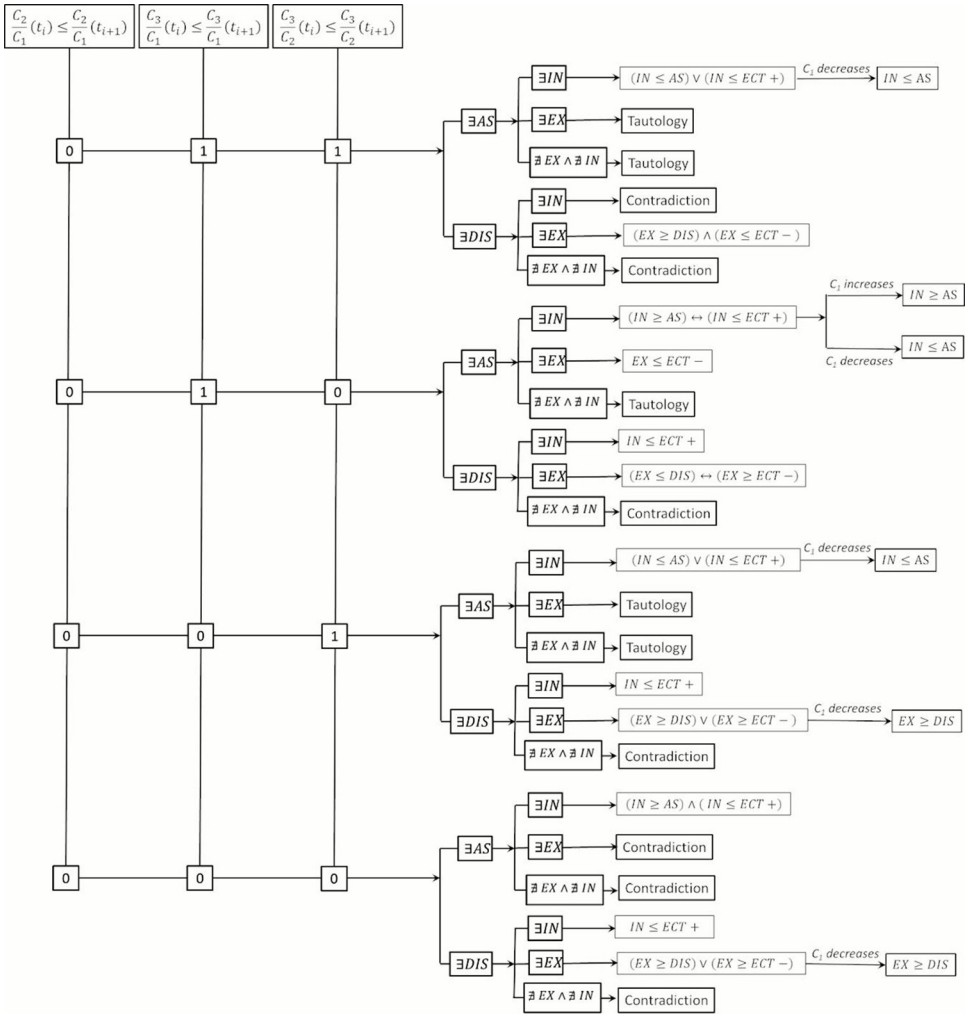

**Fig 5. Boolean schemes for the reaction and transport mechanisms (II).**

condition type is most common during the whole treatment since drug concentrations usually vary between two times instants, $t_i$ and $t_{i+1}$, according to numerical simulations. Additionally, it is important to take into account that the presence of black markers in any zone of the *RTMs* contour plot means that the extracellular concentrations, $C_1$, decreases between two time instants, $t_i$ and $t_{i+1}$.

## 5.1 Analysis of non-electroporated tissue

**5.1.1. One-short tri-exponential profile (*TPK*).** *Interaction between reaction and transport mechanisms (RTMs)*. In Figs 9 and 10, and S1 Fig of supplementary material the spatio-temporal change of the interaction between the different reaction and transport mechanisms (*RTMs*) is represented for the non-electroporated tissue and *TPK*. It is important to emphasize that both equilibrium (equal rates of *RTMs*) and non-equilibrium (different rates of *RTMs*) conditions can occur along the tissue, with the last condition type as the predominant since the drug concentration usually changes between two time instants, $t_i$ and $t_{i+1}$, as shown later. For $\lambda_{inl} = 0.0001 m/s$ (Fig 9) at the first time interval (0h to 0.5h), internalization rate imposes over or is equal to association rate (*IN*≥*AS*). For the next time interval (0.5h to 1h), *IN* can be

**Table 4. Variable and fixed parameters of the numerical simulations.**

*Electrical properties*

| | *Tissue* | | *Vessel wall* | |
|---|---|---|---|---|
| | *Value* | *Reference* | *Value* | *Reference* |
| $E_{rev}$ (kV/m) | 46 | [11] | 46 | [46] |
| $E_{irrev}$ (kV/m) | 70 | [11] | 175 | [46] |
| $\alpha^*$ | 10 | [11] | 10 | [11] |
| $\beta^*$ | 8 | [11] | 8 | [11] |
| $\sigma_{max}$ (S/m) | $3.141 \times 10^{-1}$ | [11] | $6.250 \times 10^{-1}$ | [47] |
| $\sigma_{min}$ (S/m) | $1.998 \times 10^{-2}$ | [11] | $0.630 \times 10^{-2}$ | [47] |
| $\tau_i$ (s) | 100 | [11] | 100 | [47] |

| *Transport and reaction properties* | | |
|---|---|---|
| | *Value* | *Reference* |
| $C_o$ (μM) | $2.6 \times 10^3$ | [9, 10] |
| $k_o$ (m/s) | $2.5 \times 10^{-6}$ | [10, 48] |
| $k_{1,min}$ (m/s) | $1.0 \times 10^{-9}$ | [9, 10] |
| $k_{1,max}$ (m/s) | $1.0 \times 10^{-6}$ | [9, 10] |
| $k_2$ ($\mu M^{-1}.s^{-1}$) | $0.9 \times 10^{-6}$ | [9, 10] |
| $k_{-2}$ ($s^{-1}$) | $14 \times 10^{-6}$ | [9, 10] |
| $k_{v,min}$ (m/s) | $2.8 \times 10^{-6}$ | [8, 10] |
| $k_{v,max}$ (m/s) | $4.4 \times 10^{-6}$ | [10] |
| $k_{vc,min}$ (m/Pa.s) | $2.7 \times 10^{-12}$ | [49, 50] |
| $k_{vc,max}$ (m/Pa.s) | $2.1 \times 10^{-11}$ | [49, 50] |
| $D_o$ ($m^2$/s) | $5.0 \times 10^{-11}$ | [10, 48, 51] |

| *Geometrical properties* | | |
|---|---|---|
| | *Value* | *Reference* |
| $r_{vo}$ (μm) | 16 | [9, 10, 52] |
| $R_{cord}$ (μm) | 200 | [9, 10, 51] |
| $L_{cord}$ (μm) | 400 | [9, 10, 53] |
| $\delta_{1,o}$ | $5.88 \times 10^{-2}$ | [9, 10, 48, 54] |
| $\delta_{2,o}$ | $9.41 \times 10^{-1}$ | [9, 10, 48, 54] |
| $\alpha_o$ ($m^{-1}$) | $1.94028 \times 10^5$ | [9, 10, 48, 54] |

| *Information of electroporation protocol* | | |
|---|---|---|
| | *Value* | *Reference* |
| $E$ (kV/m) | [0,46,70] | Authors |
| $d_{pulses}$ (s) | 10 | Authors |
| $T_{ep}$ (min) | 10 | Authors |
| $N_{pulses}$ | 1–5 | Authors |
| $t_{pulse}$ (μs) | 100–1000 | Authors |
| $f_{pulse}$ (Hz) | 1–10 | Authors |
| $N_{ep}$ | 6 | Authors |

| *Information about chemotherapeutic treatment* | | |
|---|---|---|
| | *Value* | *Reference* |
| Drug type | Doxorubicin | Authors |
| $M_{w,DOX}$ (g/mol) | 543.52 | [55] |
| Pharmacokinetic profiles | One-short tri-exponential, Mono-exponential | [10] |
| $T_{chemo}$ (hr) | 24 | Authors |

| *Information about vasoconstriction model* | | |
|---|---|---|
| | *Value* | *Reference* |

*(Continued)*

**Table 4.** (Continued)

*Electrical properties*

|  | Tissue | | Vessel wall | |
|---|---|---|---|---|
|  | Value | Reference | Value | Reference |
| $\tau_r$ (s) | 330 | | [7] | |
| $m_r$ $(V/m)^{-1}$ | 6.49x10$^{-5}$ | | [7] | |
| $\hat{r}_{min}$ | 0.25 | | [7] | |
| | *Blood characteristics* | | | |
| | Value | | Reference | |
| Inlet Velocity, $\lambda_{inl}$ (m/s) | [1x10$^{-4}$, 1x10$^{-3}$, 1x10$^{-2}$] | | Authors | |
| Inlet pressure, $p_{inl}$ (mm Hg) | 150 | | [56] | |
| Permeate pressure, $p_p$ (mm Hg) | 0 | | Authors | |
| Viscosity@37˚C, $\mu$ (cps) | 1.24 | | [57] | |

lower or equal than *AS* and/or than the positive extracellular transport rate (*ECT+*). Afterwards, it occurs that ($IN{\geq}AS$)∨($IN{\geq}ECT+$) up to the end of treatment, and the absence of black markers at these time instants means that the extracellular concentration ($C_1$) increases, indicating that it is not possible that $IN{\geq}ECT+$, signifying that ($IN{\geq}AS$) as for the first-time interval. For $\lambda_{inl}$ = 0.001$m/s$ (Fig 9), at the first-time interval, it is observed that $IN{\geq}AS$, like in the previous case ($\lambda_{inl}$ = 0.0001$m/s$). In the next interval (0.5h to 1h), *IN* can be lower or equal than *AS* and/or than *ECT+*, and the presence of black markers indicates a decrease in $C_1$ in all points of the tissue, concluding that $IN{\leq}ECT+$ is false and $IN{\leq}AS$ is true. Next, until the end of the treatment, it can only be inferred that there exist dissociation and internalization (∃*DIS*∧∃*IN*), and that the extracellular concentration ($C_1$) is continuously decreasing. For the greater velocity, $\lambda_{inl}$ = 0.01$m/s$ (S1 Fig of supplementary material), the behavior of the *RTMs* is similar to such observed with $\lambda_{inl}$ = 0.0001$m/s$, except for the presence of black markers from 0.5h until the end of treatment ($C_1$ continuously decreases from this time instant onwards), indicating that $IN{\leq}AS$ between $t$ = 0.5$h$ and $t$ = 1$h$, and that ($IN{\geq}AS$)∨($IN{\geq}ECT+$) from $t$ = 1$h$ until the end of the treatment.

*Space and time behavior of internalization and/or externalization rate.* According to Fig 11, for $\lambda_{inl}$ = 0.0001$m/s$, the values of $C_2/C_1$ are lower than 1 at all time instants, indicating the existence of the internalization mechanism. It is also noticed that the highest internalization rates (*IN*) occur at the first hour of evaluation ($t$ = 1$h$). Results shows that after one hour of treatment, internalization rate (*IN*) continuously decreases until the end of treatment ($C_2/C_1$ ratios increase after this time instant). Discriminating by zone of the tissue, it can be noticed that for $t$ = 0.5$h$ and $t$ = 1$h$, *IN* is higher in *TVWI* (where $C_2/C_1$ ratios are smaller), and after $t$ = 2$h$, this rate is greater in the opposed boundary (*FBV*). Moreover, the difference between $(C_2/C_1)_{max}$ and $(C_2/C_1)_{min}$ allows quantifying the radial uniformity of *IN*. For the present velocity ($\lambda_{inl}$ = 0.0001$m/s$) this difference is increasing over time, which means a reduction in the radial uniformity. For instance, for $t$ = 0.5$h$ such difference is 0.0016, for $t$ = 12$h$ it is 0.0037, and for $t$ = 24$h$ it is 0.013. For the intermediate velocity, $\lambda_{inl}$ = 0.001$m/s$ (S2 Fig of supplementary material), in a similar fashion as for the lower velocity ($\lambda_{inl}$ = 0.0001$m/s$), the values of $C_2/C_1$ are less than 1 at all time instants evaluated, i.e., internalization is also present, with the corresponding rates (*IN*) increasing until the first hour and then continuously decreasing during the whole treatment. For this velocity ($\lambda_{inl}$ = 0.001$m/s$), *IN* is also higher in TVWI at $t$ = 0.5$h$ and $t$ = 1$h$, and then higher in FBV. According to the difference between $(C_2/C_1)_{max}$ and $(C_2/C_1)_{min}$, the radial distribution of *IN* is more uniform at $t$ = 1$h$, time from which this uniformity reduces

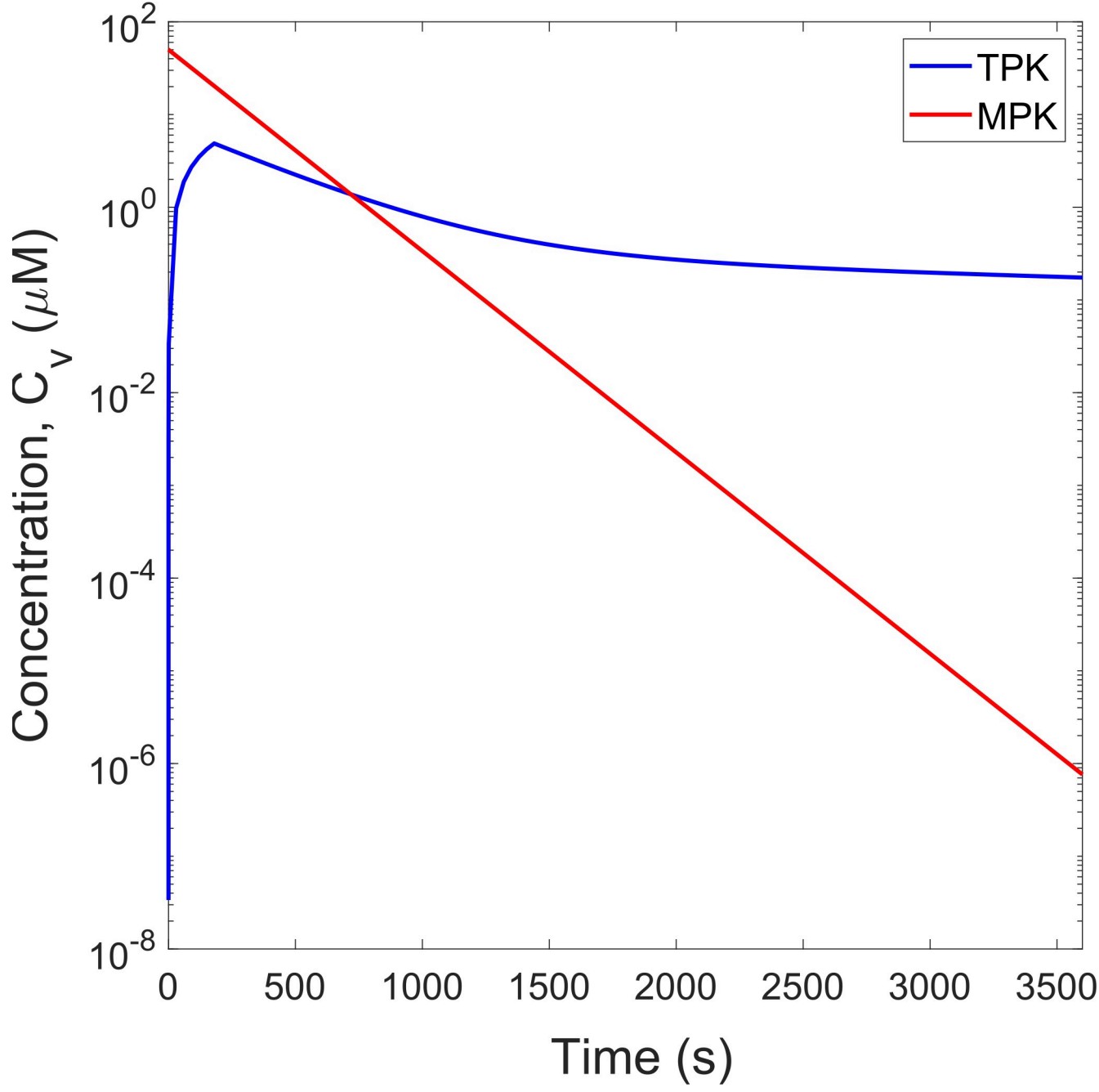

**Fig 6. Time behavior of *TPK* and *MPK* profiles until *t* = 1*h*.**

until the end of the treatment. For the larger velocity ($\lambda_{inl} = 0.01 m/s$), according to S3 Fig of supplementary material, the values and behavior of the ratio $C_2/C_1$ are very similar to the previous case with $\lambda_{inl} = 0.001 m/s$, which indicates that the increase in the blood speed has an irrelevant influence on the internalization rate (*IN*) for these two cases.

On the other hand, the straightness of the contour lines of $C_2/C_1$ accounts for the axial uniformity of drug internalization and/or externalization. In this regard, it is observed that for $\lambda_{inl}$ = 0.0001 $m/s$ the internalization rate (*IN*) is not uniform at all time instants, unlike the other

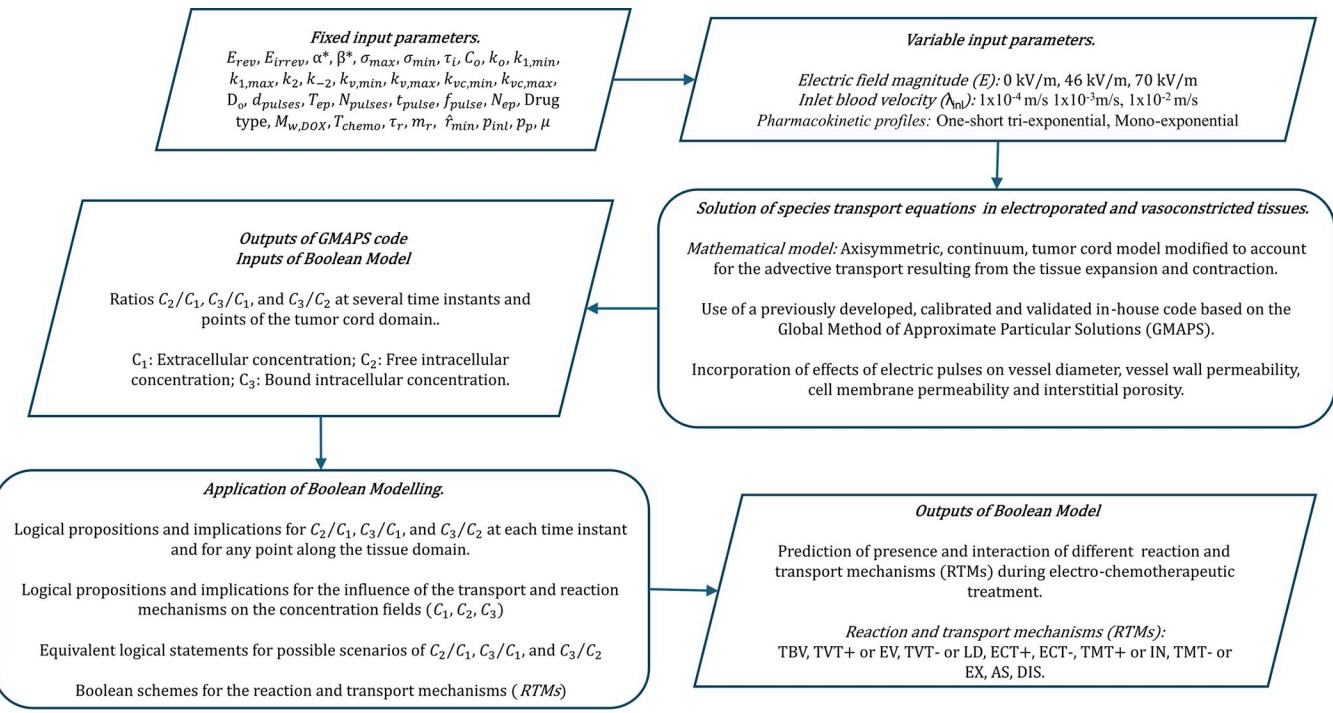

**Fig 7. Workflow of the present numerical study.**

blood speeds ($\lambda_{inl} = 0.001 m/s$ and $\lambda_{inl} = 0.01 m/s$) where this rate is evidently more uniform, indicating that the increase of $\lambda_{inl}$ favors this uniformity.

*Space and time behavior of association and/or dissociation rate.* As can be seen in the Fig 12, for $\lambda_{inl} = 0.0001 m/s$, the bound intracellular concentration ($C_3$) always increases in time, indicating the existence of association during the whole treatment. At all time instants, $C_3$ is maximum in the *TVWI* border and minimum in the *FBV*. The difference between $C_3)_{max}$ and $C_3)_{min}$ allows quantifying the radial uniformity of drug association and/or dissociation rate. This difference increases over time, in such a way that for $t = 0.5h$ this difference is $1.7176 x 10^{-3}\ \mu M$, for $t = 1h$ it is $0.00355\ \mu M$; for $t = 12h$ it is $0.0421\ \mu M$, and for $t = 24h$ it is $0.0803\ \mu M$. This shows that the association rate ($AS$) is higher in the *TVWI* border than in the *FBV* border during the whole treatment. For $\lambda_{inl} = 0.001 m/s$ (S4 Fig of supplementary material), as in the previous case ($\lambda_{inl} = 0.0001 m/s$), $C_3$ is maximum in *TVWI* and minimum in *FBV*, but in this case there is dissociation after $t = 2h$. The behavior of the difference between $(C_3)_{max}$ and $(C_3)_{min}$ is not monotonous in time, i.e., for $t = 0.5h$ this difference is $0.2374\ \mu M$, for $t = 1h$ it is $0.2505\ \mu M$, for $t = 12h$ it is $0.1988\ \mu M$, and for $t = 24h$ it is $0.2101\ \mu M$. This allows inferring that there is not predominance of the association or dissociation rate at any of the boundaries (*TVWI* and *FBV*) at all time instants. For $\lambda_{inl} = 0.01 m/s$ (S5 Fig of supplementary material), like in the first case ($\lambda_{inl} = 0.0001 m/s$), there is association during the whole treatment, $C_3$ is maximum in *TVWI* and minimum in *FBV*, and the association rate ($AS$) is higher in *TVWI* than in *FBV*. After 24 hours of electrochemotherapy, the highest intracellular bound concentration ($C_3$) is obtained for the intermediate blood velocity, $\lambda_{inl} = 0.001 m/s$.

On the other hand, the straightness of the contour lines of $C_3$ accounts for the axial uniformity of drug association and/or dissociation. The axial distribution of $C_3$ is not uniform for the three blood speeds, indicating that the association/dissociation rate is not the same for a set of points equidistant from the *TVWI* border.

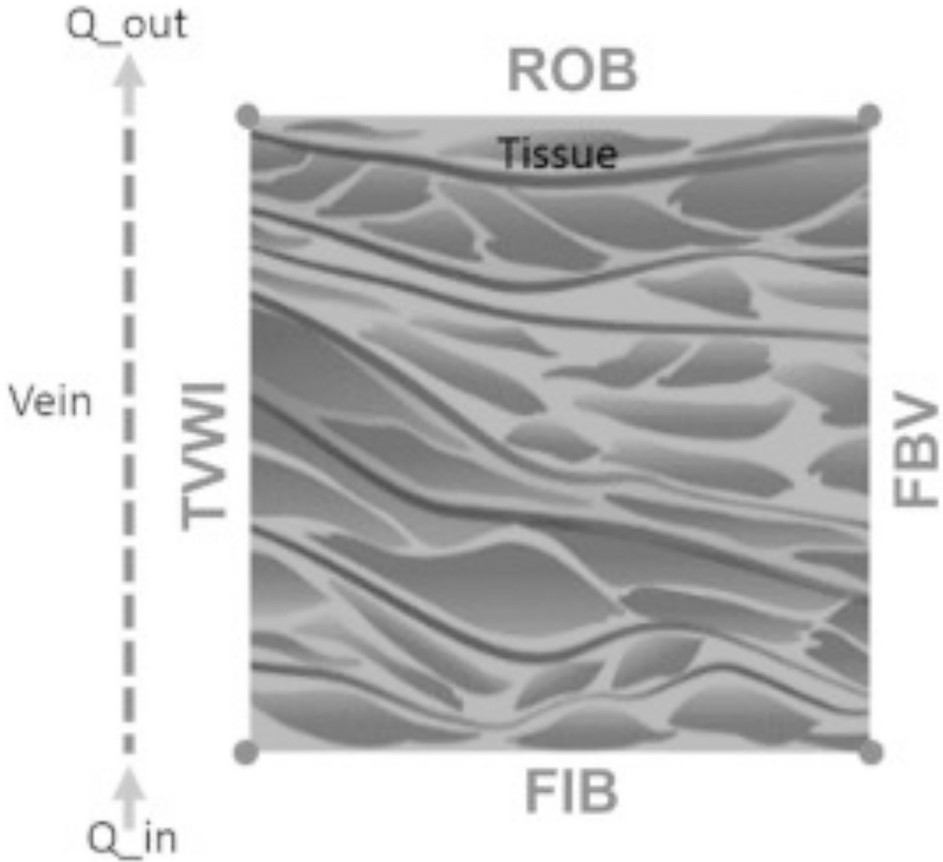

**Fig 8. Boundaries of the tumor cord domain:** *TVWI* **is the tissue–vessel wall interface,** *FBV* **is the farthest boundary from the vessel,** *FIB* **is the front inlet boundary, and** *ROB* **is the rear outlet boundary.**

**5.1.2 Mono-exponential profile (*MPK*).**   Interaction between reaction and transport mechanisms (*RTMs*). For the *MPK* profile and the three inlet blood velocities ($\lambda_{inl} = 0.0001m/s$, $\lambda_{inl} = 0.001m/s$ and $\lambda_{inl} = 0.01m/s$), S13 Fig, as well as S6 and S7 Figs of supplementary material, represent the evolution of the interaction between the *RTMs* for the non-electroporated tissue. For $\lambda_{inl} = 0.0001m/s$ (Fig 13) at the first-time interval (0h to 0.5h), $IN{\geq}AS$ is initially observed in all points in the tissue, however, from a certain time instant within this interval, externalization takes place in the lower part of *TVWI*. For the following two-time intervals (0.5h to 1h and 1h to 2h), externalization rate can be inferior or equal to negative extracellular transport rate ($EX{\leq}ECT-$) and $C_1$ increases, but from 2h until the end of treatment only the existence of association and externalization ($\exists AS{\wedge}\exists EX$) can be inferred in most of the tissue; only in some points of the *FBV* boundary, it can be asserted that $EX{\leq}ECT-$ remains. For the entire treatment $C_1$ increases continuously. For $\lambda_{inl} = 0.001m/s$ (S6 Fig of supplementary material), from $t = 0h$ to $t = 0.5h$, the behavior is the same as in the previous case. For the next time interval (0.5h to 1h), $EX{\leq}ECT-$ and $C_1$ decreases, whereas from 1h to 2h, it occurs that $(EX{\leq}ECT-){\wedge}(EX{\geq}DIS)$, but from 2h onwards, it appears a zone in the tissue where $(EX{\geq}DIS){\vee}(EX{\geq}ECT-)$ and $C_1$ continuously decreases, inferring that $EX{\geq}DIS$ is true. Therefore, for all points in the tissue, it can be concluded that externalization is present from $t = 1h$ until the end of the treatment, with rates imposing over or being equal to dissociation ones ($EX{\geq}DIS$). For $\lambda_{inl} = 0.01m/s$ (S7 Fig of supplementary material), the behavior of the *RTMs* is

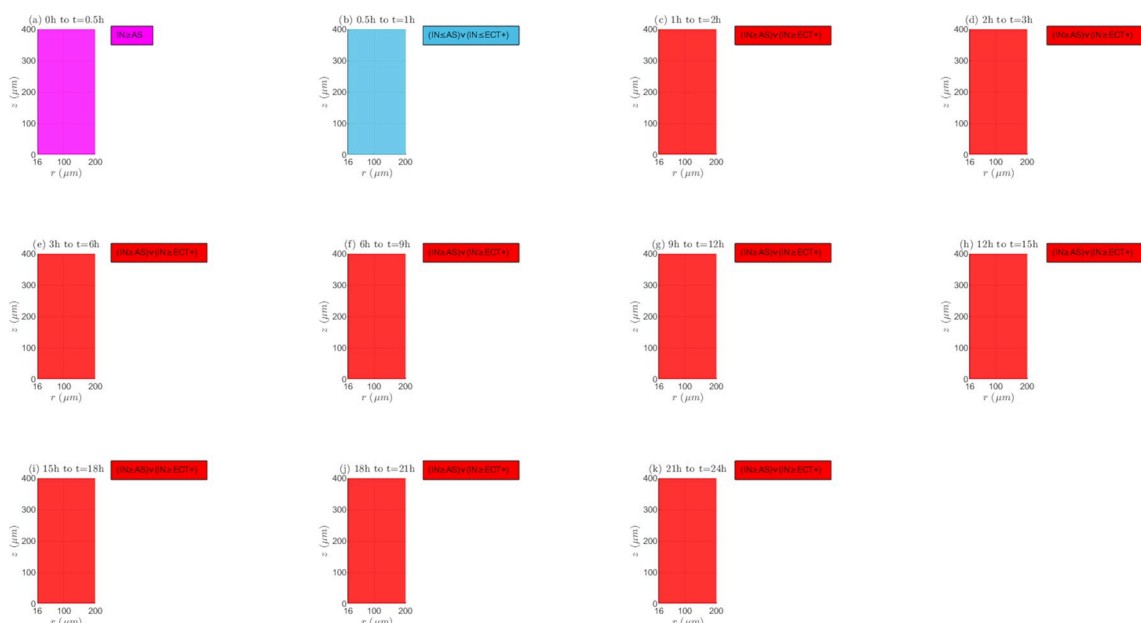

**Fig 9. Spatio-temporal evolution of the presence and interaction of reaction and transport mechanisms– $E = 0kV/m$, $\lambda_{inl} = 0.0001m/s$ and *TPK*.**

very similar to such of the lower velocity, $\lambda_{inl} = 0.0001m/s$, with the difference that $C_1$ decreases from $t = 0.5h$ until the end of the treatment.

*Space and time behavior of internalization and/or externalization rate*. In Fig 14, for $\lambda_{inl} = 0.0001m/s$, it is observed that, contrary to the results of the *TPK* with the same velocity (Fig 11), the values of $C_2/C_1$ are greater than 1 at all time instants above $t = 0.5h$, which confirms the presence of the externalization mechanism for the *MPK* profile. It is also worth-noticing that externalization rates (*EX*) considerably increase until the first hour, where the values of

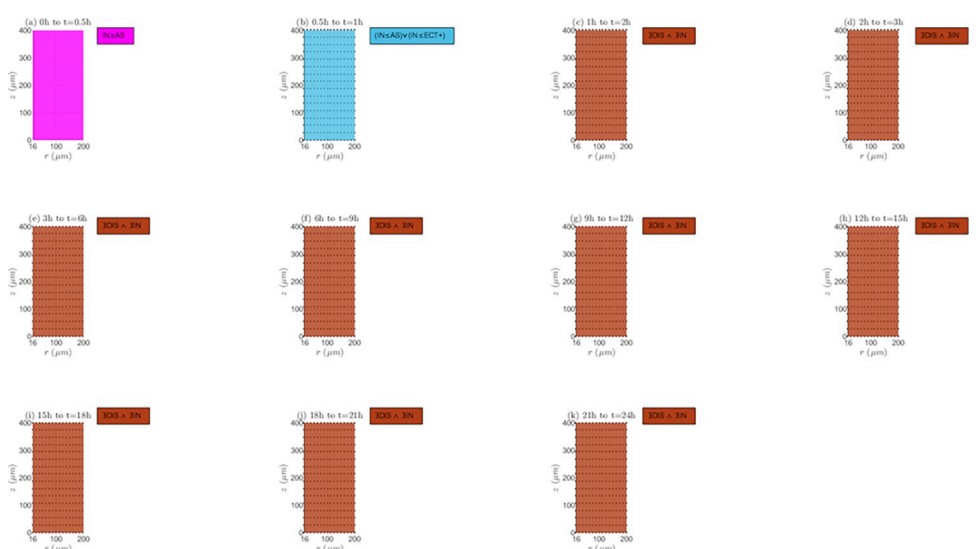

**Fig 10. Spatio-temporal evolution of the presence and interaction of reaction and transport mechanisms– $E = 0kV/m$, $\lambda_{inl} = 0.001m/s$ and *TPK*.**

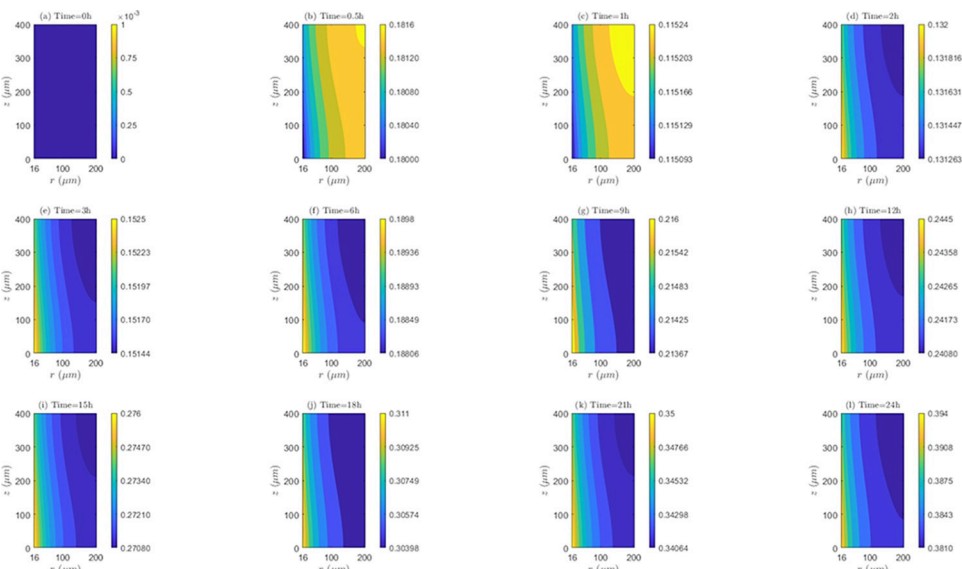

**Fig 11. Spatio-temporal evolution of $C_2/C_1$ ratios that account for internalization and externalization rates—$E = 0kV/m$, $\lambda_{inl} = 0.0001m/s$ and $TPK$.**

$C_2/C_1$ significantly augments. After $t = 1h$, $EX$ is almost stable until the end of the treatment. Discriminating by tissue areas, it can be concluded that $EX$ is always higher in the $TVWI$. With regards to the radial uniformity of $EX$, the difference between $(C_2/C_1)_{max}$ and $(C_2/C_1)_{min}$ is 2.0897 for $t = 0.5h$, 9.0139 for $t = 1h$, 8.9682 for $t = 12h$ and 8.9645 for $t = 24h$, pointing out a reduction of the radial uniformity of $EX$ until $t = 1h$, but then a stabilization of this rate until the end of treatment. For $\lambda_{inl} = 0.001m/s$, S8 Fig of supplementary material, the values of $C_2/C_1$ are greater than 1 at all time instants above $t = 0.5h$ (externalization is present too), and the externalization rate ($EX$) behaves the same in time and space as for $\lambda_{inl} = 0.0001m/s$, although

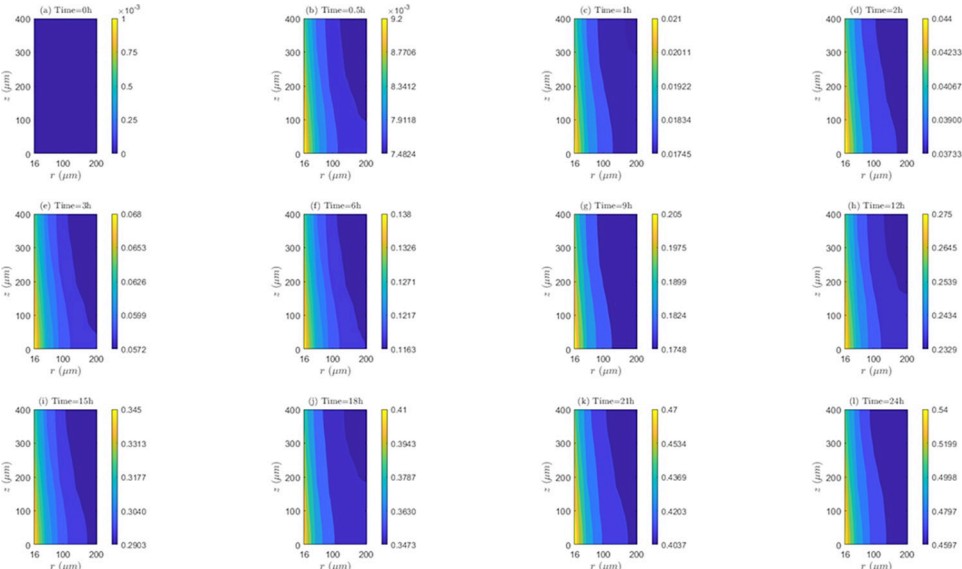

**Fig 12. Spatio-temporal evolution of bound intracellular concentration $C_3$ that account for association and dissociation $-E = 0kV/m$, $\lambda_{inl} = 0.0001m/s$ and $TPK$.**

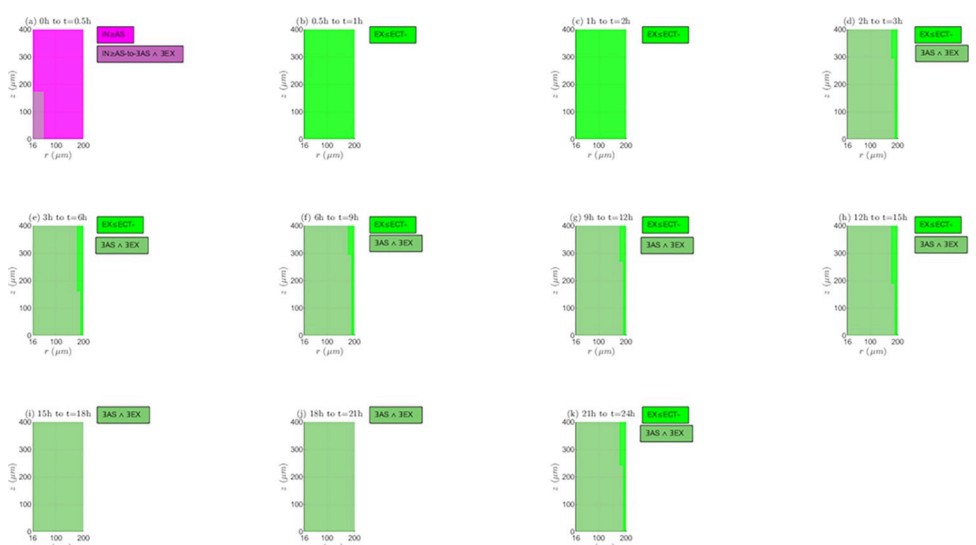

**Fig 13. Spatio-temporal evolution of the presence and interaction of reaction and transport mechanisms**
$-E = 0kV/m$, $\lambda_{inl} = 0.0001m/s$ and *MPK*.

in this case the values of *EX* are comparatively higher. The difference between $(C_2/C_1)_{max}$ and $(C_2/C_1)_{min}$ for $t = 0.5h$ is2.1017, for $t = 1h$ is 10.1706, for $t = 12h$ is 10.1266 and for $t = 24h$ is 10.1234, which illustrates that the time behavior of the radial uniformity of *EX* is the same as for $\lambda_{inl} = 0.0001m/s$, although in this case this uniformity is lower as shown by the greater differences between $(C_2/C_1)_{max}$ and $(C_2/C_1)_{min}$. For $\lambda_{inl} = 0.01m/s$, S9 Fig of supplementary material, the contours of $C_2/C_1$ are very similar to the previous case of $\lambda_{inl} = 0.001m/s$.

Regarding the axial distribution of the drug's internalization, it can be noticed that this behaves similarly to the *TPK* profile, i.e., for $\lambda_{inl} = 0.0001m/s$ the internalization rate (*IN*) is not uniform in the axial direction, but for the other two velocities ($\lambda_{inl} = 0.001m/s$ and $\lambda_{inl} =$

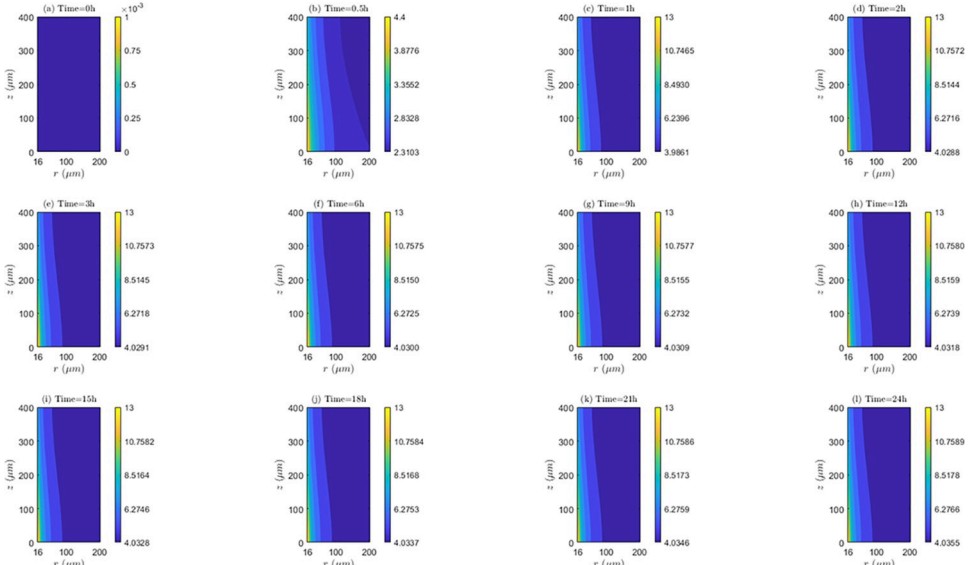

**Fig 14. Spatio-temporal evolution of $C_2/C_1$ ratios that account for internalization and externalization rates**
$-E = 0kV/m$, $\lambda_{inl} = 0.0001m/s$ and *MPK*.

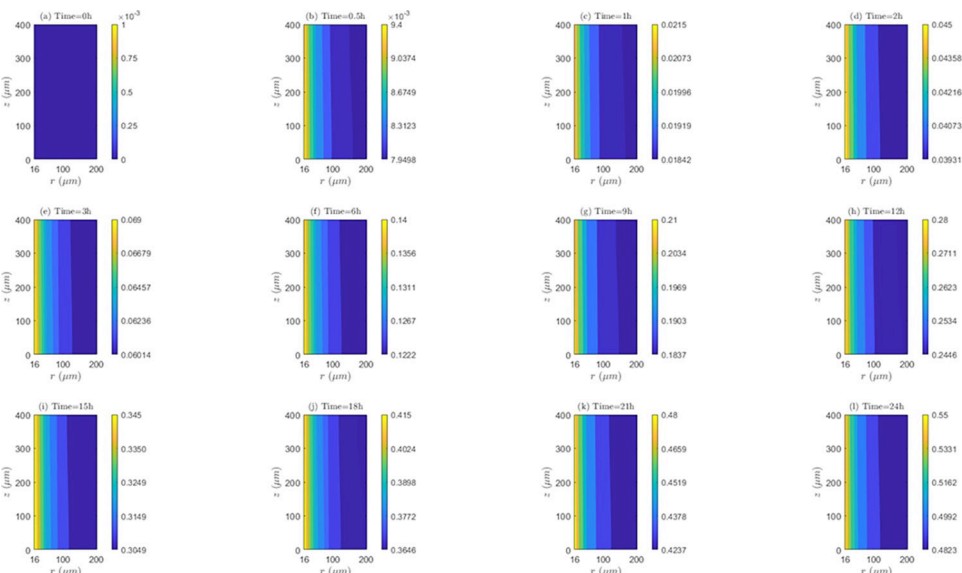

**Fig 15. Spatio-temporal evolution of bound intracellular concentration $C_3$ that account for association and dissociation $-E = 0 kV/m$, $\lambda_{inl} = 0.0001 m/s$ and MPK.**

$0.01 m/s$) an axial homogeneity is obtained, confirming that the increase in the blood velocity benefits this homogeneity.

*Space and time behavior of association and/or dissociation rate.* As can be observed in Fig 15 for $\lambda_{inl} = 0.0001 m/s$, in a similar fashion as for *TPK*, $C_3$ increases at all time instants (representing the existence of association) and $C_3$ is maximum in the *TVWI* border. The difference between $(C_3)_{max}$ and $(C_3)_{min}$ increases over time, in such a way that for $t = 0.5h$ it is $1.4502 x 10^{-3} \mu M$, for $t = 1h$ it is $0.00308\ \mu M$, for $t = 12h$ it is $0.0354\ \mu M$, and for $t = 24\ h$ it is $0.0977\ \mu M$, showing that the association rate (*AS*) is higher in the *TVWI*. For $\lambda_{inl} = 0.001 m/s$ (S10 Fig of supplementary material), differently to the previous case ($\lambda_{inl} = 0.0001 m/s$), $C_3$ increases only until $t = 1h$ and then decreases due to the presence of dissociation during most of the treatment. Similar to the previous case, $C_3$ is maximum in the *TVWI* and minimum in the *FBV*. The difference between $(C_3)_{max}$ and $(C_3)_{min}$ does not behave monotonously over time, indicating that *AS* or *DIS* might be higher or lower in the *TVWI* versus the *FBV*. For $\lambda_{inl}$ $= 0.01 m/s$ (S11 Fig of supplementary material), likewise the case of $\lambda_{inl} = 0.0001 m/s$, there is association during the treatment and $C_3$ is maximum in the *TVWI*. Moreover, the difference between $(C_3)_{max}$ and $(C_3)_{min}$ is increasing over time, showing a larger *AS* in the *TVWI*. As for the *TPK* profile, the highest intracellular bound concentration ($C_3$) is obtained for the intermediate blood velocity, $\lambda_{inl} = 0.001\ m/s$.

The straightness of the contour lines shows that the axial distribution of $C_3$ is uniform for the three blood velocities considered here, indicating that *AS* or *DIS* is almost the same for certain points equidistant from the *TVWI*.

**5.1.3 Comparison with recent literature of non-electroporated tissues.** The geometric, transport, reaction and pharmacokinetic properties used in this paper are very similar to those used in Groh et al [9], where both *TPK* and *MPK* were considered for drug administration in non-electroporated tissues ($E = 0\ kV/m$). Even though, contrary to the present work, the axial change of drug concentration in the vessel ($C_v$) was not considered there [9], some results are consistent with the present findings. For instance, in the present work, for all inlet blood velocities, $\lambda_{inl} = [1 x 10^{-4}\ m/s, 1 x 10^{-3} m/s, 1 x 10^{-2}\ m/s]$, the cellular internalization is present during

the whole treatment for *TPK* (Figs 9 and 10 of manuscript, and S1 Fig of supplementary material), whereas this mechanism is only present at the first instants for *MPK*, for which, in turns, the cellular externalization takes place during almost the entire treatment (Fig 13 of manuscript, and S6 and S7 Figs of supplementary material). Groh et al [9] found a significant increase of the extracellular concentration ($C_1$) at the first instants of treatment for both profiles (*TPK* and *MPK*), which was more pronounced as the points get closer to the vessel (*TVWI* boundary in this case). Because the initial intracellular concentration was zero, this sudden increment of $C_1$ generated cellular internalization at these first time instants, as founded here from $t = 0\,h$ to $t = 0.5\,h$. Due to the nature of both profiles, Groh et al [9] predicted that, once it reaches the maximum value, $C_1$ decreases exponentially until $t = 5h$ approximately for both profiles, but this reduction is more abrupt in *MPK*, boosting the drug externalization for this profile. In *TPK*, this reduction is smoother as the points get closer to the vessel, which could favor the internalization. Afterwards, Groh et al [9] predicted that the decrease of $C_1$ is kept along the whole tissue domain until $t = 24\,h$ for *MPK*, favoring the externalization, whereas $C_1$ increases at several points for *TPK*, which benefits the internalization.

## 5.2 Analysis for electroporated tissue *(E = 46 kV/m)*

**5.2.1 One-short tri-exponential profile (*TPK*).**   *Interaction between reaction and transport mechanisms (RTMs).* In the S12–S14 Figs of supplementary material, they are shown the interactions between the *RTMs* arising in the tissue electroporated with an electrical field magnitude of 46 *kV/m* (reversible threshold), for the three different blood speeds ($\lambda_{inl} = 0.0001 m/s$, $\lambda_{inl} = 0.001 m/s$ and $\lambda_{inl} = 0.01 m/s$). For $\lambda_{inl} = 0.0001 m/s$ (S12 Fig of supplementary material) and $\lambda_{inl} = 0.001 m/s$, (S13 Fig of supplementary material), the behavior of the *RTMs* is similar to such observed for the non-electroporated tissue ($E = 0\,kV/m$), with the only difference that in the interval from $t = 9h$ to $t = 12h$, $C_1$ decreases for all points of the domain. For $\lambda_{inl} = 0.01 m/s$ (S14 Fig of supplementary material), the behavior of the *RTMs* is the same as for the non-elecroporated tissue with this velocity.

*Space and time behavior of internalization and/or externalization rate.* According to the Fig 16 of manuscript, and S15 and S16 Figs of supplementary material, some aspects of the behavior of $C_2/C_1$ change with respect to the non-electroporated tissue when electrical pulses are applied, while others are not altered. For all blood velocities, it is possible to verify the presence of the internalization mechanism during the whole treatment. For the particular case of $\lambda_{inl} = 0.0001 m/s$, S16 Fig of manuscript, from the first hour to the end of the treatment, the internalization rate (*IN*) decreases, coinciding with the non-electroporated tissue. It is also observed that the internalization rate (*IN*) is smaller in the lower part of the *TVWI* at all time instants. As for the non-electroporated tissue, the radial uniformity of *IN* decreases with the time according to the evolution of the difference between $(C_2/C_1)_{max}$ and $(C_2/C_1)_{min}$. For $\lambda_{inl} = 0.001 m/s$ (S15 Fig of supplementary material) and $\lambda_{inl} = 0.01 m/s$ (S16 Fig of supplementary material), *IN* behaves the same as for $\lambda_{inl} = 0.0001\,m/s$, namely, it decreases over the time after the first hour, being lower along the whole *TVWI*. For both blood velocities ($\lambda_{inl} = 0.001 m/s$ and $\lambda_{inl} = 0.01 m/s$), the decrease in the internalization rate (*IN*) after the first hour coincides with the results of the non-electroporated tissue. Furthermore, the increment of the difference between $(C_2/C_1)_{max}$ and $(C_2/C_1)_{min}$ indicates that the radial uniformity of *IN* decreases after the first hour for these two velocities.

On the other hand, contour lines of $C_2/C_1$ depict that for $\lambda_{inl} = 0.0001 m/s$ and $\lambda_{inl} = 0.001 m/s$, internalization rate (*IN*) is not axially uniform at all times, as opposed to $\lambda_{inl} = 0.01 m/s$ where this rate is homogeneous, confirming the positive influence of increasing $\lambda_{inl}$ on the longitudinal uniformity of *IN*.

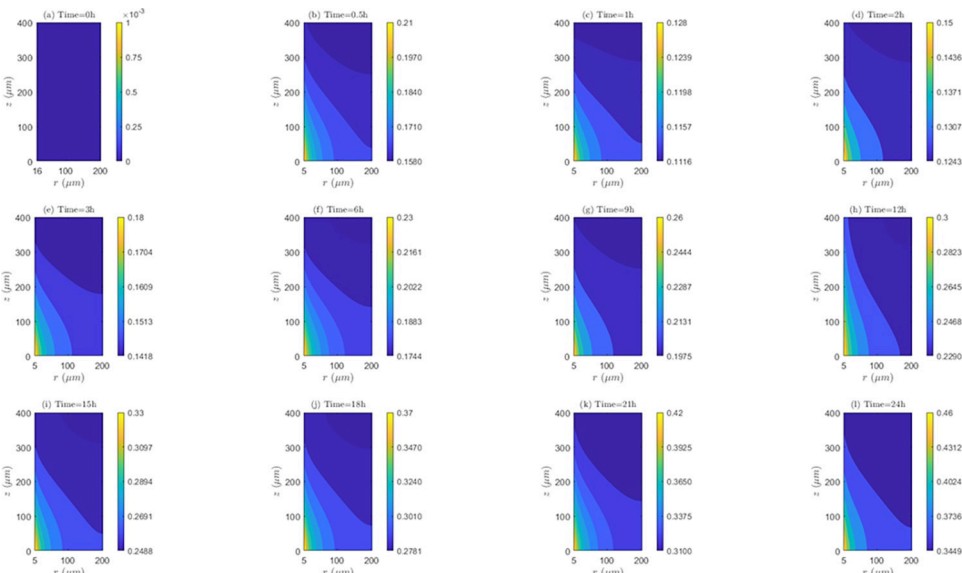

**Fig 16. Spatio-temporal evolution of $C_2/C_1$ ratios that account for internalization and externalization rates—$E = 46kV/m$, $\lambda_{inl} = 0.0001m/s$ and TPK.**

*Space and time behavior of association and/or dissociation rate.* As observed in Fig 17 of manuscript, for $\lambda_{inl} = 0.0001m/s$, $C_3$ increases during the treatment (there is association) and this concentration is maximum in *TVWI* and minimum in *FBV*, being this result similar to the one obtained for the non-electroporate tissue, although in this case ($E = 46\ kV/m$) the maximum values of $C_3$ are concentrated in the inferior zone of the *TVWI*. The difference between $(C_3)_{max}$ and $(C_3)_{min}$ increases over time, indicating that *AS* remains higher in the *TVWI*, also coinciding with the non-electroporate tissue. For the velocity of $\lambda_{inl} = 0.001m/s$ (S17 Fig of

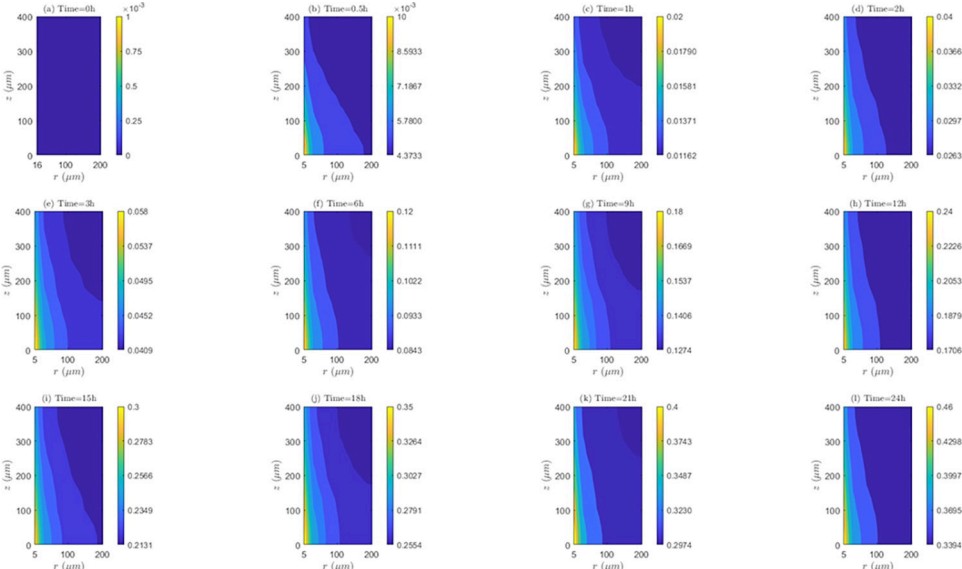

**Fig 17. Spatio-temporal evolution of bound intracellular concentration $C_3$ that account for association and dissociation—$E = 46kV/m$, $\lambda_{inl} = 0.0001m/s$ and TPK.**

supplementary material), the changes of $C_3$ show that there is association at the first hour, and afterwards, dissociation occurs, which also agrees with the non-electroporated tissue. On the contrary, for $\lambda_{inl} = 0.01 m/s$ (S18 Fig of supplementary material), association occurs during the whole treatment, also coinciding with the case of $E = 0\ kV/m$. For these two last speeds ($\lambda_{inl} = 0.001 m/s$, $\lambda_{inl} = 0.01 m/s$), $C_3$ is also maximum in the inferior part of the $TVWI$, but the difference between $(C_3)_{max}$ and $(C_3)_{min}$ is not monotonous, concluding that there is no predominance of the dissociation rate ($DIS$) for $\lambda_{inl} = 0.001 m/s$ or association rate ($AS$) for $\lambda_{inl} = 0.01 m/s$ in any of the radial boundaries ($TVWI$ or $FBV$).

In a similar fashion to the case with $E = 0\ kV/m$, the axial distribution of $C_3$ is not uniform for the three blood velocities, namely, $AS/DIS$ differs for points equidistant to the $TVWI$, and the highest values of $C_3$ are also achieved for the intermediate velocity, $\lambda_{inl} = 0.001 m/s$.

**5.2.2 Mono-exponential profile ($MPK$).** *Interaction between reaction and transport mechanisms ($RTMs$).* For $\lambda_{inl} = 0.0001 m/s$, S19 Fig of supplementary material, the behavior of the $RTMs$ is similar to the one observed for the non-electroporated tissue and this profile ($MPK$), with the differences that in the interval from $t = 9h$ to 12h, $C_1$ decreases for all points of the domain, and from $t = 2h$ onwards, the phenomenon $EX{\leq}ECT-$ does not necessarily occur in some points close to the $FBV$, where only the existence of association and externalization without any particular interaction ($\exists AS \wedge \exists EX$) can be inferred at this time. For $\lambda_{inl} = 0.001 m/s$, S20 Fig of supplementary material, at the first three-time intervals, the behavior of the $RTMs$ is the same than for non-electroporated tissue. For the following three intervals (2h to 3h, 3h to 6h, 6h to 9h), it occurs that $(EX{\geq}DIS) \vee (EX{\geq}ECT-)$ in some parts of the tissue, with the last inequality as false since $C_1$ decreases, leading to $EX{\geq}DIS$ as true. In some parts of the tissue closed to the $FBV$, it happens that $(EX{\leq}ECT-) \wedge (EX{\geq}DIS)$, namely, both conditions shall be complied. At the time interval from $t = 9h$ to $t = 12h$, $(EX{\geq}DIS) \vee (EX{\geq}ECT-)$ and $C_1$ increases, whereas from $t = 12h$ to $t = 15h$, $EX{\leq}ECT-$ and $C_1$ decreases. From $t = 15h$ up to the end of the treatment, the behavior of $RTMs$ is similar to such between 2h and 9h. For $\lambda_{inl} = 0.01 m/s$ (S21 Fig of supplementary material), the behavior of $RTMs$ is similar to such observed for the non-electroporated tissue with the same blood velocity.

*Space and time behavior of internalization and/or externalization rate.* In the Fig 18 of manuscript, for $\lambda_{inl} = 0.0001 m/s$, resembling the phenomenon observed for the non-electroporated tissue, externalization occurs, and an important increment of externalization rates ($EX$) takes place until the first hour, from which $EX$ remains almost constant in most of tissue domain until the end of treatment. There is a small difference with the non-electroporated tissue, since $EX$ is not higher along the whole $TVWI$ for the present electric field magnitude ($E = 46\ kV/m$), but only in the inferior part of this boundary. Regarding the radial uniformity of $EX$, which is accounted for by the difference between $(C_2/C_1)_{max}$ and $(C_2/C_1)_{min}$, the behavior is the same as for the protocol without electroporation ($E = 0\ kV/m$). For the other two velocities, $\lambda_{inl} = 0.001 m/s$ (S22 Fig of supplementary material) and $\lambda_{inl} = 0.01 m/s$, (S23 Fig of supplementary material), the behavior of contours $C_2/C_1$ is similar to such observed in the non-electroporated tissues with this profile and these same velocities, hence, the aforementioned observations can be extended for these cases. Regarding the axial uniformity of $EX$, similar conclusions to the ones of the non-electroporated tissue with the same profile ($MPK$) can be addressed.

*Space and time behavior of association and/or dissociation rate.* Observing the Fig 19 of manuscript, for $\lambda_{inl} = 0.0001 m/s$, $C_3$ increases during the treatment (there is association), it is maximum in $TVWI$, and the incremental difference between $(C_3)_{max}$ and $(C_3)_{min}$ points out that the association rate ($AS$) is higher in $TVWI$ during the whole treatment. This agrees with the corresponding case with $E = 0\ kV/m$. For $\lambda_{inl} = 0.001 m/s$ (S24 Fig of supplementary material), $C_3$ is also maximum in the $TVWI$ and minimum in the $FBV$. For this velocity, association is present in all points of the tissue until the second hour ($t = 2h$), from which dissociation

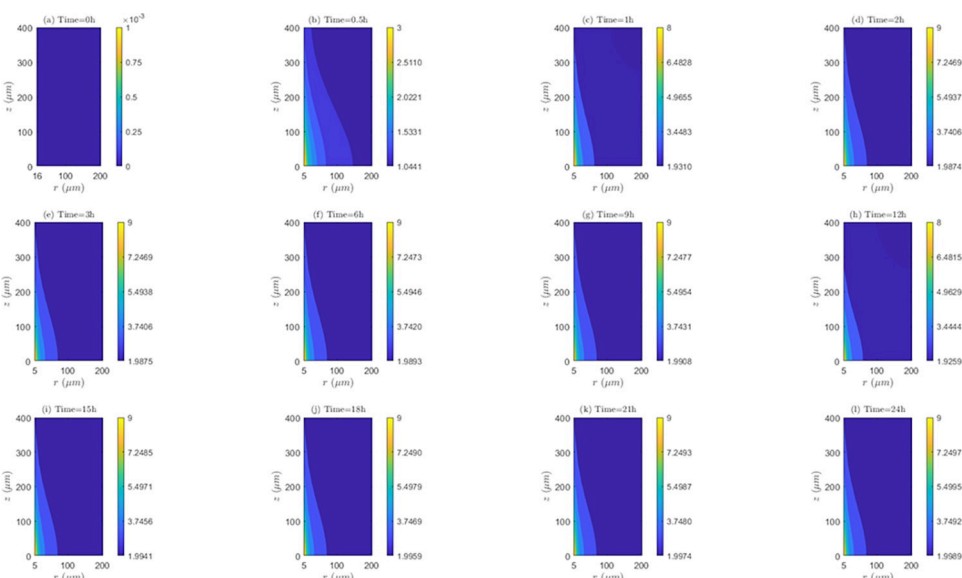

**Fig 18. Spatio-temporal evolution of $C_2/C_1$ ratios that account for internalization and externalization rates—$E = 46kV/m$, $\lambda_{inl} = 0.0001m/s$ and $MPK$.**

arises. The behavior of the difference between $(C_3)_{max}$ and $(C_3)_{min}$ is not monotonous over time, indicating that there is no predominance of the rate of association or dissociation in any of the radial boundaries (*TVWI* and *FBV*) at all time instants. For $\lambda_{inl} = 0.01m/s$ (S25 Fig of supplementary material), the behavior of $C_3$ is similar as for the lower velocity, $\lambda_{inl} = 0.0001$ $m/s$.

The axial distribution of $C_3$ is slightly uneven for the three blood speeds ($\lambda_{inl} = 0.0001m/s$, $\lambda_{inl} = 0.001m/s$ and $\lambda_{inl} = 0.01m/s$), indicating that *AS/DIS* can have minimum variations for certain points equidistant from the *TVWI*. As for the non-electroporated tissue, maximum values of $C_3$ are achieved for the intermediate velocity, $\lambda_{inl} = 0.001$ $m/s$.

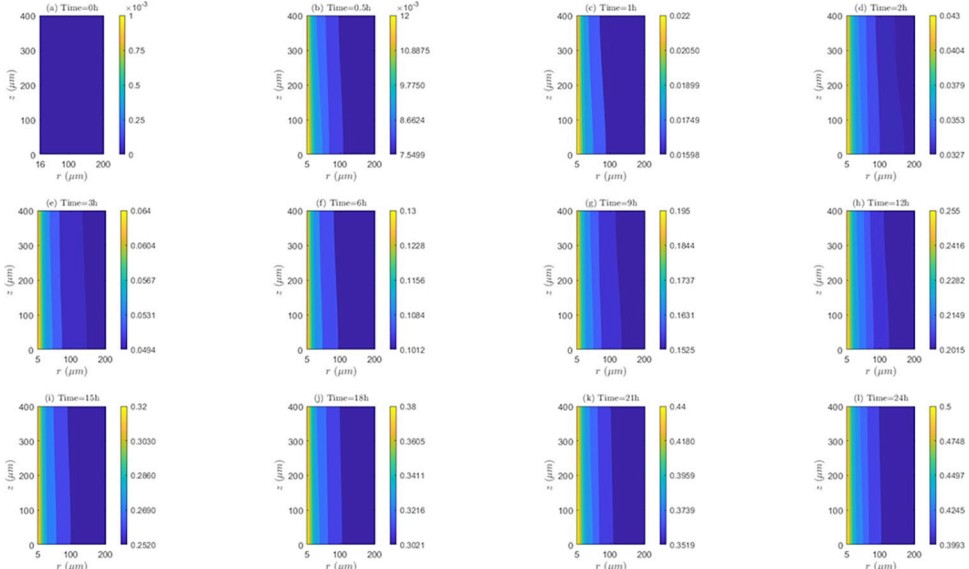

**Fig 19. Spatio-temporal evolution of bound intracellular concentration $C_3$ that account for association and dissociation—$E = 46kV/m$, $\lambda_{inl} = 0.0001m/s$ and $MPK$.**

**5.2.3 Comparison with recent literature of tissues electroporated with the reversible electric field.** As it was shown in the present section, when tissue is electroporated with the reversible electric field magnitude ($E = 46\ kV/m$), the $C_2/C_1$ ratio that accounts for the cellular internalization or externalization rates under the assumption of Fickian flow, decreases radially from the *TVWI* towards the *FBV* for both *TPK* (Fig of manuscript, and S15 and S16 Figs of supplementary material) and *MPK* (Fig 18 of manuscript, S22 and S23 Figs of supplementary material), being this more noticeable as $\lambda_{inl}$ is greater. In a recent work [59], it was studied the influence of the pulse numbers ($N_{ep}$) on the extracellular and intracellular concentrations of a tissue electroporated with a reversible electric field magnitude of $E = 28\ kV/m$, considering a uniform extracellular concentration of $C_1 = 1M$ in one of the borders of the tumor domain (corresponding to the *TVWI* boundary in the present work). Despite the difference in the reversible thresholds ($28\ kV/m$ vs. $46\ kV/m$), it was found that at a given time instant, the ratio between intracellular and extracellular concentrations ($C_2/C_1$) is greater in the drug inlet boundary (*TVWI* in this case), decreasing towards the outermost boundary (*FBV* in this case), and this behavior remains unchanged by increasing the number of pulses ($N_{ep}$).

## 5.3 Analysis for electroporated tissue *(E = 70 kV/m)*

All contour plots of the present subsection are reported as supplementary material since the workflow analysis of these plots has been illustrated enough in the previous subsections, and this considerably reduces the manuscript length.

**5.3.1 One-short tri-exponential profile (*TPK*).** *Interaction between reaction and transport mechanisms (RTMs).* For *TPK* and $\lambda_{inl} = 0.0001 m/s$ (S26 Fig), the behavior of the *RTMs* until $t = 9h$ is very similar as the one obtained for $E = 0\ kV/m$ and $E = 46\ kV/m$. In the interval from 9h to 12h in some points near the lower part of the *TVWI*, it is observed that $(IN{\geq}AS){\vee}(IN{\geq}ECT+)$ to $EX{\geq}ECT-$, which means that in those points, at a given time instants, a change from internalization to externalization, and positive to negative extracellular transport takes place. However, in most of the domain for this time interval (9h to 12h), it occurs that $(IN{\geq}AS){\vee}(IN{\geq}ECT+)$ and contrary to the previous intervals, $C_1$ decreases. In the interval from 12h to 15h, $(IN{\leq}AS){\vee}(IN{\leq}ECT+)$ and $C_1$ increases in most of the domain, but from a certain time instant within this same interval, in some points of the lower part of the *TVWI*, it happens that $(IN{\leq}AS){\vee}(IN{\leq}ECT+)$ to $\exists AS{\wedge}\exists EX$, indicating that internalization ceases to then occurs externalization, but the association remains present. From $t = 15h$ to the end of treatment, in most of the tissue, it happens that $(IN{\geq}AS){\vee}(IN{\geq}ECT+)$ and $C_1$ increases, signifying that $IN{\geq}ECT+$ is false and therefore $IN{\geq}AS$ is true, matching with the results of $E = 0kV/m$ and $E = 46kV/m$ for the *TPK* profile.

For $\lambda_{inl} = 0.001 m/s$ (S27 Fig), the behavior of *RTMs* until $t = 3h$ is equal to the one observed for $E = 46\ kV/m$. The following two intervals (3h to 6h and 6h to 9h) show that $\exists DIS{\wedge}\exists IN$ in the neighborhood of the *FIB*, and that $(IN{\geq}AS){\vee}(IN{\geq}ECT+)$ and $C_1$ decreases in most of the domain, i.e., internalization is present in all points, but not association. In the interval from 9h to 12h, it can appreciated that $(EX{\leq}ECT-){\wedge}(EX{\geq}DIS)$ to $\exists DIS{\wedge}\exists IN$ in some points along the *TVWI*, in other words, in these points externalization ceases to then occur internalization, but dissociation is always present. For the rest of the domain at this time interval (9h to 12h), it can be noticed that $IN{\geq}ECT+$. From $t = 12h$ to $t = 15h$, in all points of the domain, it is present that $IN{\leq}ECT+$ until a certain time instant within this interval, but some points on the *TVWI* shows that $(IN{\geq}AS){\wedge}(IN{\leq}ECT+)$ to $\exists AS{\wedge}\exists EX$, which means that internalization stops and externalization arises in these points. The next two-time intervals (15h to 18h and 18h to 21h) show that for most of the domain, it occurs that $(IN{\geq}AS){\vee}(IN{\geq}ECT+)$ and $C_1$ increases, implying that it is not possible that $IN{\geq}ECT+$, and therefore $IN{\geq}AS$ is true. From $t = 21h$ to

$t = 24h$, it occurs that $(EX{\leq}ECT-){\wedge}(EX{\geq}DIS)$ and $C_1$ decreases, that is, in the last three hours of treatment the externalization appears in almost all points of the tissue.

For $\lambda_{inl} = 0.01m/s$ (S28 Fig), the behavior of the $RTMs$ is the same as the one observed with $E = 0\ kV/m$ and $E = 46\ kV/m$ up to $t = 9h$. In the interval from 9h to 12h, it occurs that $(IN{\geq}AS){\vee}(IN{\geq}ECT+)$ in all points of the domain until a certain time instant, from which it happens that $(IN{\geq}AS){\vee}(IN{\geq}ECT+)$ *to* $EX{\leq}ECT-$ in some points near the $TVWI$, i.e., internalization and positive extracellular transport ceases, and the opposite mechanisms surge. In the interval from 12h to 15h, $IN{\leq}ECT+$ occurs in all areas of the tissue until a certain time instant, from which, it can be noticed that $(IN{\geq}AS){\wedge}(IN{\leq}ECT+)$ *to* $\exists AS{\wedge}\exists EX$ in some points along the $TVWI$, meaning that externalization takes place instead of internalization. During the whole-time interval (12h to 15 h), $C_1$ increases. From $t = 15h$ to $t = 21h$, $RTMs$ behaves the same as from $t = 9h$ to $t = 12h$, except for some points near the TVWI where the externalization occurs. In the last time interval (21h to 24h), it is present that $(IN{\geq}AS){\vee}(IN{\geq}ECT+)$ in most of the domain, but in some points near the $TVWI$ it occurs that $EX{\leq}ECT-$.

*Space and time behavior of internalization and/or externalization rate.* According to the S29 Fig, for $\lambda_{inl} = 0.0001m/s$, internalization prevails at all time instants and the highest internalization rates ($IN$) are present at the first hour, where the values of $C_2/C_1$ are the lowest, coinciding this with the results of $E = 0kV/m$ and $E = 46kV/m$. Moreover, as for $E = 46kV/m$, $IN$ is lower at the bottom of $TVWI$ for all time instants. The difference between $(C_2/C_1)_{max}$ and $(C_2/C_1)_{min}$ is increasing over time, which means a reduction in the radial uniformity of $IN$. For $\lambda_{inl} = 0.001m/s$, S30 Fig, internalization is present in almost the entire tissue, apart from some points near the $TVWI$ border at some time instants where externalization appears ($C_2/C_1{>}1$). As for $E = 0kV/m$ and $E = 46kV/m$, highest internalization rates ($IN$) still occur at the first hour. Additionally, $IN$ is lower along the $TVWI$ at all time instants. As for the lowest velocity ($\lambda_{inl} = 0.0001m/s$), the incremental difference between $(C_2/C_1)_{max}$ and $(C_2/C_1)_{min}$ indicates a reduction in the radial uniformity of $IN$. For $\lambda_{inl} = 0.01m/s$, S31 Fig, the values and behavior of the ratio $C_2/C_1$ are very similar to ones of the previous velocity ($\lambda_{inl} = 0.001m/s$), which denotes an irrelevant influence of the blood velocity on $IN$ for these two cases ($\lambda_{inl} = 0.001m/s$ and $\lambda_{inl} = 0.01m/s$).

The internalization rate ($IN$) is not axially uniform for $\lambda_{inl} = 0.0001m/s$, whereas for the other two velocities ($\lambda_{inl} = 0.001m/s$ and $\lambda_{inl} = 0.01m/s$), contour lines of $C_2/C_1$ show negligible horizontal deviations, which means that $IN$ is more axially uniform.

*Space and time behavior of association and/or dissociation rate.* According to S32 Fig, the behavior of $C_3$ for $\lambda_{inl} = 0.0001m/s$ is similar to such with $E = 46kV/m$, that is, there is association during the treatment, $C_3$ is maximum in $TVWI$ and minimum in $FBV$, and association rate ($AS$) is higher in the $TVWI$; however, after 24h of treatment, $C_3$ is higher for $E = 70kV/m$ than for $E = 46\ kV/m$ in almost all points of the tissue domain. For $\lambda_{inl} = 0.001m/s$, S33 Fig, the variation of $C_3$ over the time in all points of the domain is not monotonous, indicating alternating periods of association and dissociation. As for $E = 46\ kV/m$, $C_3$ is maximum in the inferior part of the $TVWI$ and minimum along the $FBV$. In the association periods, the difference between $(C_3)_{max}$ and $(C_3)_{min}$ decreases over the time, which indicates that the association rate ($AS$) is lower in the $TVWI$ than in $FBV$. For $\lambda_{inl} = 0.01m/s$, S34 Fig, $C_3$ increases the over time, indicating association, and remains maximum in the inferior part of the $TVWI$, whereas the reduction of the difference between $(C_3)_{max}$ and $(C_3)_{min}$ after $t = 0.5h$ figures out that $AS$ is also lower in $TVWI$ during most of the treatment.

For all blood speeds ($\lambda_{inl} = 0.0001m/s$, $\lambda_{inl} = 0.001m/s$ and $\lambda_{inl} = 0.01m/s$), the axial distribution of $C_3$ is clearly affected along the treatment, i.e., the association and/or dissociation rates are not uniform along the bloodstream direction. As in all previous cases, maximum values of $C_3$ are obtained for the intermediate velocity, $\lambda_{inl} = 0.001m/s$.

**5.3.2 Mono-exponential profile (*MPK*).** *Interaction between reaction and transport mechanisms (RTMs).* For $\lambda_{inl} = 0.0001m/s$ (S35 Fig), the behavior of the *RTMs* is the same as the one observed for $E = 46kV/m$. For $\lambda_{inl} = 0.001m/s$ (S36 Fig), behavior is also similar as for $E = 46kV/m$, with the exception that in the fourth interval (2h to 3h), it occurs that $(EX \leq DIS) \lor (EX \leq ECT-)$ in some points along the *FBV*. For $\lambda_{inl} = 0.01m/s$ (S37 Fig), the behavior of RTMs also resembles that of $E = 46kV/m$, with the difference that in the interval from 12h to 15h, $C_1$ increases for all points of the tissue.

*Space and time behavior of internalization and/or externalization rate.* For $\lambda_{inl} = 0.0001m/s$ (S38 Fig), just as for $E = 0kV/m$ and $E = 46kV/m$, externalization is the predominant mechanism for this *MPK* profile. The highest externalization rates (*EX*) occur after the first hour of treatment, except for $t = 12h$ where a significant decrease of these rates is appreciated. As in the case of $E = 46kV/m$, the highest *EX* occurs at the bottom of the *TVWI*; the radial uniformity of *EX* behaves in the same manner as for $E = 46kV/m$. For the other two blood velocities, i.e., $\lambda_{inl} = 0.001m/s$ (S39 Fig) and $\lambda_{inl} = 0.01m/s$ (S40 Fig), the behavior of internalization/externalization rates is kept regarding the case with $E = 46kV/m$, with the exception of $t = 12h$ where a noticeable drop of $C_2/C_1$ occurs.

The influence of $\lambda_{inl}$ on the axial uniformity of externalization/internalization rates (*EX/IN*) is the same as for the non-electroporate tissue, i.e., for $\lambda_{inl} = 0.0001m/s$, *EX/IN* are not axially homogenous, while for the other two speeds, they are.

*Space and time behavior of association and/or dissociation rate.* For $\lambda_{inl} = 0.0001m/s$ (S41 Fig), $C_3$ behaves in the same way as for $E = 46kV/m$, that is, increases continuously, is maximum in *TVWI* and minimum in *FBV*, and the *AS* is higher in the *TVWI*. The maximum value of $C_3$ for the present electrical field ($E = 70 \ kV/m$) is greater than for the other two ($E = 0 \ kV/m$, $E = 46 \ kV/m$). For the other two blood speeds ($\lambda_{inl} = 0.001m/s$ and $\lambda_{inl} = 0.01m/s$), S42 and S43 Figs, respectively, $C_3$ behaves similarly to the corresponding speeds with $E = 46kV/m$. As for $E = 0kV/m$ and $E = 46kV/m$, $C_3$ is maximum for the intermediate velocity, $\lambda_{inl} = 0.001m/s$, when the electric field magnitude is $E = 70kV/m$. In regard to the axial uniformity of $C_3$, it can be seen that this is similar as with $E = 46kV/m$ for the three values of $\lambda_{inl}$, that is, the influence of $\lambda_{inl}$ on such uniformity is negligible.

**5.3.3 Comparison with recent literature of tissues electroporated above the reversible electric field.** The influence of the application of electric pulses, with magnitude above the reversible electric field, on the intracellular and extracellular concentrations have been recently studied [60, 61], achieving results qualitatively consistent with the present study. According to [60], the increase of the electric field magnitude (*E*) above the reversible threshold (10 $kV/m$ in that case) leads to the equilibrium of concentrations ($C_1 = C_2$) in a shorter time. In such work, for a time window of 200s, the concentration balance between $C_1$ and $C_2$ was reached for 13 $kV/m$, 14 $kV/m$ and 15 $kV/m$ at approximate times of 190s, 75s and 20s, respectively. Under the simulation conditions of the present work, it was not possible to obtain a continuous concentration balance probably due to the presence of protein association, which was disregarded in [60]; however, for $E = 70 \ kV/m$, partial equilibrium conditions are achieved ($C_2/C_1 = 1$) for *TPK* profile from 18 hours until the end of the treatment in some points near the *TVWI* (S29–S31 Figs). On the hand, for $E = 46 \ kV/m$ (Fig 16 of manuscript, and S15 and S16 Figs), partial equilibrium conditions are not achieved, confirming that the increase of E above the reversible threshold ($E = 46 \ kV/m$ in the present case) can promote this concentration balance. The influence of the pulses' strengths above the reversible threshold on extracellular and intracellular concentrations was also studied in [61] using a single cell-electroporation model. Three levels were considered for the electrical field magnitude (15 $kV/m$, 25 $kV/m$ and 40 $kV/m$) and two for the pulse spacing (50 s and 100 s), in a time window of 1000s. In such work [61], for a constant pulse spacing ($d_{pulses}$), when E increases 1.67 times (from 15 $kV/m$ to 25 $kV/m$) and 1.60

times (from 25 $kV/m$ to 40 $kV/m$), the average intracellular concentration augmented 17 and 2 times, respectively; on the other hand, for a constant electrical field magnitude ($E$), by reducing twice the pulse spacing (from 100 s to 50 s), the increase of the average intracellular concentration was only 1.2 times approximately, implying that the electric field magnitude ($E$) has a greater influence on the internalization rate ($IN$) than the pulse spacing ($d_{pulses}$). This could be corroborated in further research with the model developed in the present work.

## 5.4 Transvascular transport at the first hour of treatment

Since both pharmacokinetic profiles ($TPK$ and $MPK$) are exponential, the largest changes of the ratio $C_v/C_{1,TVWI}$, with $C_v$ and $C_{1,TVWI}$ as the drug concentration along the bloodstream and the extracellular concentration at the tissue-vessel wall interface, respectively, are expected at the first instants of the electro-chemotherapeutic treatment. The $C_v/C_{1,TVWI}$ ratio is useful to determine the time instants where reverse diffusion takes place, namely, drug transport from the tissue towards the free plasma. This emulates a drug clearance back to the plasma by reverse concentration gradients, but it is not exactly a lymphatic drainage since lymphatic vessels are usually more permeable than the capillary vessel included in the tumor cord domain (Fig 2). The time evolution of the $C_v/C_{1,TVWI}$ ratio calculated at the axial coordinate $z = 0$ until the first hour of treatment ($t = 1h$) is shown in Fig 20A and 20C for the three blood velocities ($1x10^{-4} m/s$, $1x10^{-3} m/s$, $1x10^{-2} m/s$) and electric field magnitudes (0 $kV/m$, 46 $kV/m$, 70 $kV/m$) considered here. In these figures, the green line corresponds to $C_v/C_{1,TVWI,z=0} = 1$, above which extravasation occurs and below which a reverse drug diffusion takes place. For visualization purposes, semilog-y plots are reported.

For all values of $\lambda_{inl}$ and $E$, the $C_v/C_{1,TVWI,z=0}$ ratios are larger in $MPK$ by one order of magnitude at the beginning of the treatment, indicating greater rates of extravasation for this pharmacokinetic profile, which is associated to the IV bolus injection in a short period of time. However, the time decrease of $C_v/C_{1,TVWI,z=0}$ is more pronounced for $MPK$ as well, suggesting

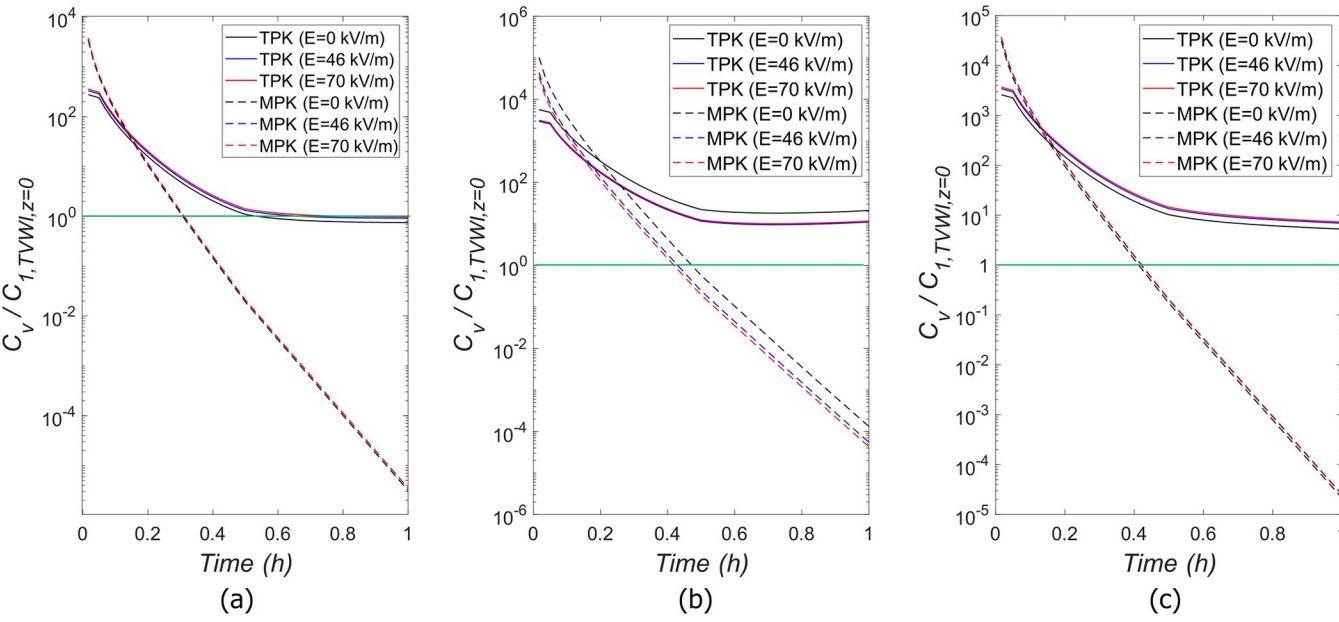

**Fig 20. Time behavior of the ratio of drug concentration at the bloodstream to the extracellular concentration at the tissue-vessel wall interface ($TVWI$) at z =0, $C_v/C_{1,TVWI,z=0}$ vs. t.** (a) $\lambda_{inl} = 1x10^{-4}$ $m/s$, (b) $\lambda_{inl} = 1x10^{-3}$ $m/s$, (c) $\lambda_{inl} = 1x10^{-2}$ $m/s$. The green line corresponds to $C_v/C_{1,TVWI,z=0} = 1$, above which extravasation occurs and below which a reverse drug diffusion takes place (from tissue towards the bloodstream).

a faster increase of the extracellular concentration at the *TVWI* regarding the plasma concentration, which leads to drug diffusion back towards the bloodstream at a time instant ranging between 0.31h to 0.48h depending on the combination between $\lambda_{inl}$ and *E*. The exponential decay rate of $C_v/C_{1,TVWI,z=0}$ is maintained for *MPK*. On the other hand, for the *TPK* profile, $C_v/C_{1,TVWI,z=0}$ considerably decreases until the first half hour ($t = 0.5h$) approximately, from which the time change of $C_v/C_{1,TVWI,z=0}$ is significantly smaller, remaining almost constant for $\lambda_{inl} = 1x10^{-4}$ *m/s* where it is predicted drug transport back to the vessel at very low rates when compared with *MPK*; for $\lambda_{inl} = 1x10^{-3}$ *m/s* and $\lambda_{inl} = 1x10^{-2}$ *m/s*, $C_v/C_{1,TVWI,z=0}$ barely increases and decreases, respectively, and drug extravasation is kept for both velocities and all values of *E*.

## 6. Conclusions and clinical implications

In the present study, the influence of two parameters of the systemic administration of chemotherapeutic drugs on the presence, relative importance and rates of the transport and reaction mechanisms (*RTMs*) involved in electroporated and vasoconstricted tissues is investigated, namely, the bloodstream inlet velocity ($\lambda_{inl}$) and the pharmacokinetic profile (*PK*), considering several electric field magnitudes (*E*). Results were obtained by simulating several treatment scenarios with an in-silico tool previously developed, calibrated and validated, which is based on the Global Method of Approximate Particular Solutions (*GMAPS*) [7]. According to the numerical results, the *RTMs* for the One-short Tri-exponential (*TPK*) and Mono-exponential profile (*MPK*) can differ each other for a given combination of *E* and $\lambda_{inl}$. For *TPK*, the reversible threshold (*E* = 46 *kV/m*) does not affect the interaction of the *RTMs* with respect to the non-electroporated tissue (*E* = 0 *kV/m*) for the three blood speeds considered here, $\lambda_{inl}$ = [$1x10^{-4}$ *m/s*, $1x10^{-3}$ *m/s*, $1x10^{-2}$ *m/s*], but this interaction is clearly influenced by $\lambda_{inl}$. Accordingly, in non-equilibrium conditions, the drug internalization rate (*IN*) imposes over the drug association rate (*AS*) at the first half hour, and then, from $t = 0.5h$ to $t = 1h$ a reverse behavior can occur for all values of $\lambda_{inl}$ and both electric field magnitudes (*E* = 0 *kV/m* and *E* = 46 *kV/m*). However, from $t = 1h$ to the end of the treatment, the existence and relative importance of *RTMs* depends on the blood velocity, in such a way that for $\lambda_{inl} = 1x10^{-4}$ *m/s*, internalization rate (*IN*) is greater than association one (*AS*); for $\lambda_{inl} = 1x10^{-3}$ *m/s*, there exist dissociation and internalization but nothing can be inferred about how they interact with each other; and for $\lambda_{inl} = 1x10^{-2}$ *m/s*, it can be asserted that internalization rate (*IN*) can impose over association rate (*AS*) and/or over positive extracellular transport rate (*ECT+*). On the other hand, the irreversible threshold (*E* = 70 *kV/m*) evidently affects the *RTMs* regarding the non-electroporated tissue (*E* = 0 *kV/m*), mainly after $t = 1h$, and this influence depends on $\lambda_{inl}$. Accordingly, in non-equilibrium conditions, for the lower velocity ($\lambda_{inl} = 1x10^{-4}$ *m/s*), *IN* is greater than *AS* during most of the treatment after $t = 1h$ and in the majority of points of the domain, but from $t = 9h$ to $t = 12h$, *IN* imposes over *AS* and/or over *ECT+*, and externalization occurs in some points close to the *TVWI*. For the intermediate velocity ($\lambda_{inl} = 1x10^{-3}$ *m/s*), internalization is present until $t = 21h$ except in some points closed to *TWVI* where externalization occurs between $t = 9h$ to $t = 12h$; additionally, periods of association and dissociation can be present from $t = 1h$ to $t = 21h$, whereas positive extracellular transport is predominant from $t = 3h$ to $t = 21h$. In the last three hours of the chemotherapeutic treatment ($t = 21h$ to $t = 24h$), externalization, dissociation and negative extracellular transport take place, with the externalization rate (*EX*) lower than the negative extracellular transport rate (*ECT−*), but larger than the dissociation one (*DIS*). For the larger velocity considered here ($\lambda_{inl} = 1x10^{-2}$ *m/s*), internalization, association and positive extracellular transport are the predominant *RTMs* during the whole treatment, whereas externalization and negative extracellular transport only occurs in some

points close to the *TVWI* at some time intervals. In most of the points of the domain, internalization rate (*IN*) surpasses the association rate (*AS*) but can be smaller or larger than the positive extracellular transport rate (*ECT+*) depending on the time interval.

For the *MPK* profile, the reversible threshold ($E = 46\ kV/m$) and blood velocity ($\lambda_{inl}$) can affect the interaction between the *RTMs*. For both electric field magnitudes ($E = 0\ kV/m$ and $E = 46\ kV/m$) and the three inlet velocities, $\lambda_{inl} = [1x10^{-4}\ m/s, 1x10^{-3}\ m/s, 1x10^{-2}\ m/s]$, in a different manner as for *TPK*, internalization is only present up to the first half hour, but then externalization imposes in the whole domain until the end of the treatment. Additionally, for both values of *E* and all values of $\lambda_{inl}$, in non-equilibrium conditions, externalization rate (*EX*) is inferior to the negative extracellular transport rate (*ECT−*) until $t = 2h$. The main difference appears after $t = 2h$, where association exists until the end of the treatment in the non-electroporated tissue ($E = 0\ kV/m$) for $\lambda_{inl} = 1x10^{-4}\ m/s$ and $\lambda_{inl} = 1x10^{-2}\ m/s$, whereas dissociation is predominant for the intermediate velocity ($\lambda_{inl} = 1x10^{-3}\ m/s$). For $E = 46\ kV/m$, association is also present until the end of treatment for $\lambda_{inl} = 1x10^{-4}\ m/s$ and $\lambda_{inl} = 1x10^{-2}\ m/s$, whereas both dissociation and negative extracellular transport can be present for $\lambda_{inl} = 1x10^{-3}\ m/s$, with the externalization rate (*EX*) imposing over dissociation rate (*DIS*), and being smaller or larger than the negative extracellular transport rate (*ECT−*) depending on the time instant. On the other hand, contrarily to the *TPK* profile, the application of the electric field magnitude of $E = 70\ kV/m$ does not considerably alter the interaction between the *RTMs* involved in the electro-chemotherapeutic treatment with respect to the reversible threshold ($E = 46\ kV/m$).

In general, numerical results allow concluding that, for both pharmacokinetic profiles (*TPK* and *MPK*), the radial uniformity of internalization/externalization rates (*IN/EX*) for the non-electroporated tissues is negatively affected when the blood velocity increases, specially from $\lambda_{inl} = 1x10^{-4}\ m/s$ to $\lambda_{inl} = 1x10^{-3}\ m/s$, whereas an opposite effect is reached for the axial homogeneity of *IN/EX*, i.e., the increase of $\lambda_{inl}$ benefits this homogeneity. On the other hand, the effect of the blood velocity ($\lambda_{inl}$) on the radial and axial uniformity of association/dissociation rates (*AS/DIS*) is not necessarily monotonic for the conditions considered here. These conclusions can be extended for the electroporated tissues ($E = 46\ kV/m$ and $E = 70\ kV/m$), but it is worth mentioning that, in general, the increase of *E* reduces the radial uniformity of *IN/EX* and *AS/DIS* regarding the non-electroporated tissue, whereas the axial uniformity is affected in a lower extent.

The Boolean model presented in this study could have practical applications in the development of electro-chemotherapy treatments since it could be useful for establishing appropriate drug dosages and pharmacokinetic profiles, as well as for determining the necessary electroporation parameters. Prior knowledge of the interaction between the reaction and transport mechanisms (*RTMs*) of the drug can aid in determining these parameters before the treatment implementation. For instance, when comparing two treatments with the same drug type and total drug exposure, one in which cellular internalization (*IN*) of drug occurs continuously over time in most of the tissue, and another in which internalization (*IN*) only happens at the beginning and is followed by externalization (*EX*), as *TPK* and *MPK* considered here, the former may be more advantageous because it leads to a higher possibility of increased the final cell death since drug resides inside the cells during more time. The above findings also suggest that the selection of the appropriate electric field magnitude (*E*) should consider the type and parameters of the *PK* profile, along with the expected blood perfusion rates in the target tissues, because the spatio-temporal behavior of the concentration fields and exposures of the drug, as well as the presence, interaction and rates of the *RTMs* can significantly change for a given value of *E* when the other parameters are modified. According to the numerical results, this can be particularly decisive as the electric field magnitude (*E*) is closer to the irreversible threshold (70 kV/m in the present case). Accordingly, the characterization, monitoring,

control and/or correction of the *PK* parameters (specially when AUC-based dosing is employed) and the blood perfusion rate (vascular normalization, selective depletion of *TME* constituents, among others) could be relevant for the prescription of the electric field strength in electro-chemotherapeutic treatments.

As formerly mentioned, the total drug exposure or Area Under the Curve (*AUC*) is fixed for both profiles (*MPK* and *TPK*) in the present work, having a value of $AUC = 2.78 \ \mu M. \ h$. However, some parameters of the *PK* profiles can be fitted to reach a target *AUC* or target exposures to extracellular drug ($C_{1exp} = \int_0^{T'} C_1.dt$), free intracellular drug ($C_{2exp} = \int_0^{T'} C_2.dt$) and/or bound intracellular drug ($C_{3exp} = \int_0^{T'} C_3.dt$) after a given time (T'). For instance, by modifying the total drug dose ($D_0$) and time of infusion ($\tau'$) in the Eq (35), which predicts the time evolution of $C_v$ for *TPK*, different target values of *AUC*, $C_{1exp}$, $C_{2exp}$ and $C_{3exp}$, which are more common parameters in the clinical practice related with the cell-kill capacity of the therapy, can be obtained using the present model. Despite the *PK* parameters of each profile are not modified here, it is worth mentioning that the present model could be used in future works for evaluating the effects of modifying these parameters on the drug concentration fields, drug exposures (which are more useful at clinical level), and the presence and interaction of the reaction and transport mechanisms (*RTMs*) of chemotherapeutic drugs.

The mechanisms of protein association (*AS*) or dissociation (*DIS*) are highly significant in clinical contexts. Accordingly, the high drug association in the blood vessels and the interstitial matrix slows down the rates of extravasation (*EV*) and cellular internalization (*IN*), respectively, which may imply higher drug doses or treatment duration to reach the target cell death. Nevertheless, association (*AS*) also delays the lymphatic drainage (*LD*), resulting in a longer drug residence time inside the tumor, which can favor the therapy efficiency. Within the intracellular environment, a high association (*AS*) can be advantageous for enhancing the therapy effectiveness by inhibiting the cellular externalization (*EX*). As reasonable, dissociation (*DIS*) produces reverse impacts. The protein content of cancer patients has a significant role in the side effects of the treatment. Patients with low protein levels, known as hypoproteinemia, are more susceptible to side effects and require lower drug doses. Additionally, these patients are less prone to drug association, with the implications that this entails for the treatment efficacy [62–65].

The extracellular transport (*ECT*) is also highly significant in a clinical context. Facilitating the interstitial transportation of drug (*ECT+*) to cancer cell receptors is advantageous for Antibody-Drug Conjugate (*ADC*) therapies, as the drug is only released upon internalization. On the other hand, the positive extracellular transport (*ECT+*) to healthy peripheral cells can promote the bystander effect. This occurs when the death of tumor cells is intensified due to the absence of feed and protection from the neighboring healthy cells that have been injured by the medicament. In clinical practice, the process of negative extracellular transport (*ETC–*) can be beneficial in some areas of the tissue (for example the tissue-vessel wall interface) since it could help prevent the formation of reverse concentration gradients or allows for the release of the interstitial pressure.

In general, any combination of parameters that promotes the radial and axial uniformity of the rates associated to the *RTMs* considered here brings about more homogenous cell-kill effects on the target tissue, which is very beneficial at clinical level, particularly in low-vascularized tumors.

It is worth-mentioning that the present model is not intended to replace the Heuristic methods commonly used for dosing of anticancer drugs (weight-based dosing, body surface area dosing, fixed-dose prescribing, AUC-based dosing, among others), but to complement these methods. Heuristic methods have been defined based on in vitro and in vivo studies, and

clinical trials considering several aspects such as: type of drug, type and subtype of cancer, stage of cancer, previous health treatments, current patient's medications and mainly, the patient's characteristics (age, weight, height, health problems, among others). One of the main drawbacks of these methods is their poor relationship with the pharmacokinetic parameters of chemotherapeutic drugs (loading and maintenance dose, injection time, decay rates, among others), affecting the accuracy in the prediction of the drug exposure and drug clearance, and consequently, the individualization capacity of the therapy. Moreover, since some heuristic models are not able to correctly predict the drug clearance for some cases, the definition of the chemotherapy schedule can also carry some inaccuracies when defined by these methods. For instance, the Body Surface Area (*BSA*), which is still an extensively used variable for definition of the drug dose, has not shown an acceptable correlation with the *AUC* and clearance of doxorubicin [66] or cisplatin [67], which are common drugs employed in the cancer treatment. As a complement, the present model is able to predict the spatio-temporal evolution of drug concentration in the vessels, extracellular and intracellular space, as well as the transport and reaction mechanisms of the chemotherapeutic drug in electroporated and vasoconstricted tissues. Accordingly, the present model could assist the initial evaluation of the effects of the chemotherapy doses and schedule on the drug distribution, concentration and exposure along the vessel and tissues during the whole treatment prior to the application of the therapy.

Generally speaking, most anticancer agents have a very narrow therapeutic window, which means that a small change in dosage can lead to unexpected side effects in the patient. This is one of the main risks of some heuristic methods widely used in clinical practice, such as BSA, which, not being closely correlated with AUC, may generate different cytotoxic processes in patients with similar characteristics (weight, age, height, health problems, etc.), such as systemic DNA degeneration and inflammation of healthy tissues, in turns leading to less severe side effects (nausea, vomiting, dysgeusia, hair loss), life-threatening effects (myelosuppression, febrile neutropenia, severe oral mucositis, sepsis, among others) or late effects at cardiac, pulmonary, neurological, renal or hepatic level. The model for the concentration fields developed and validated in previous works [7, 17, 30], which was briefly presented in Section 2 of the manuscript and was used to generate the inputs of the Boolean model, allows estimating the exposure to the chemotherapeutic drug considering the parameters of the electroporation protocol, the reaction and transport properties of the drug in the tissue, the geometrical and electrical properties of the tissue, information of the chemotherapeutic treatment (type of drug, injected dose, pharmacokinetic profile, treatment time, among others), fitting parameters of the chemotherapeutic cell death models, and characteristics related to the blood perfusion rate (vasoconstriction, blood velocity and pressure, viscosity, etc.). Accordingly, under the premise of the high correlation between the drug exposure and side effects of the therapy, this model could be very useful to complement the design of chemotherapeutic protocols by heuristic methods, focused on reducing the side effects without considerably compromising the efficacy of the treatment. This could be addressed in detail in future works.

The scaling to macroscopic domains with complex vascular and lymphatic networks, the consideration of more specific reaction and transport mechanisms arising in several subdomains (like the endothelial wall), the inclusion of cell death models by electroporation and cytotoxic effects, among others, are challenges that represent opportunities of improvement of the present model. Additionally, it could be necessary to enhance the model by including the mechanisms of drug resistance of tumor cells, such as: (1) tumor heterogeneity due to mutations, deletions, chromosomal rearrangements, among others [68, 69], (2) inactivation of the medication by alteration of its molecular characteristics when binding to proteins [68, 69], (3) uncontrolled release of certain medications (Doxorubicin, Vinblastine, and Taxol) into the extracellular space due to certain substances like chloride transported by ATP-dependent

carriers [68, 70], (4) changes in drug metabolism by enzymes [68, 71], (5) secondary mutations in cancer cells such as topoisomerase II [68, 72], (6) processes of DNA damage and repair, especially in platinum compounds and doxorubicin [68, 73].

## Supporting information

**S1 Fig. Spatio-temporal evolution of the presence and interaction of reaction and transport mechanisms—$E = 0kV/m$, $\lambda_{inl} = 0.01m/s$ and *TPK*.**
(TIF)

**S2 Fig. Spatio-temporal evolution of $C_2/C_1$ ratios that account for internalization and externalization rates—$E = 0kV/m$, $\lambda_{inl} = 0.001m/s$ and *TPK*.**
(TIF)

**S3 Fig. Spatio-temporal evolution of $C_2/C_1$ ratios that account for internalization and externalization rates—$E = 0kV/m$, $\lambda_{inl} = 0.01m/s$ and *TPK*.**
(TIF)

**S4 Fig. Spatio-temporal evolution of bound intracellular concentration $C_3$ that account for association and dissociation—$E = 0kV/m$, $\lambda_{inl} = 0.001m/s$ and *TPK*.**
(TIF)

**S5 Fig. Spatio-temporal evolution of bound intracellular concentration $C_3$ that account for association and dissociation—$E = 0kV/m$, $\lambda_{inl} = 0.01m/s$ and *TPK*.**
(TIF)

**S6 Fig. Spatio-temporal evolution of the presence and interaction of reaction and transport mechanisms–$E = 0kV/m$, $\lambda_{inl} = 0.001m/s$ and *MPK*.**
(TIF)

**S7 Fig. Spatio-temporal evolution of the presence and interaction of reaction and transport mechanisms–$E = 0kV/m$, $\lambda_{inl} = 0.01m/s$ and *MPK*.**
(TIF)

**S8 Fig. Spatio-temporal evolution of $C_2/C_1$ ratios that account for internalization and externalization rates—$E = 0kV/m$, $\lambda_{inl} = 0.001m/s$ and *MPK*.**
(TIF)

**S9 Fig. Spatio-temporal evolution of $C_2/C_1$ ratios that account for internalization and externalization rates—$E = 0kV/m$, $\lambda_{inl} = 0.01m/s$ and *MPK*.**
(TIF)

**S10 Fig. Spatio-temporal evolution of bound intracellular concentration $C_3$ that account for association and dissociation—$E = 0kV/m$, $\lambda_{inl} = 0.001m/s$ and *MPK*.**
(TIF)

**S11 Fig. Spatio-temporal evolution of bound intracellular concentration $C_3$ that account for association and dissociation—$E = 0kV/m$, $\lambda_{inl} = 0.01m/s$ and *MPK*.**
(TIF)

**S12 Fig. Spatio-temporal evolution of the presence and interaction of reaction and transport mechanisms—$E = 46kV/m$, $\lambda_{inl} = 0.0001m/s$ and *TPK*.**
(TIF)

**S13 Fig. Spatio-temporal evolution of the presence and interaction of reaction and transport mechanisms—$E = 46kV/m$, $\lambda_{inl} = 0.001m/s$ and $TPK$.**
(TIF)

**S14 Fig. Spatio-temporal evolution of the presence and interaction of reaction and transport mechanisms—$E = 46kV/m$, $\lambda_{inl} = 0.01m/s$ and $TPK$.**
(TIF)

**S15 Fig. Spatio-temporal evolution of $C_2/C_1$ ratios that account for internalization and externalization rates—$E = 46kV/m$, $\lambda_{inl} = 0.001m/s$ and $TPK$.**
(TIF)

**S16 Fig. Spatio-temporal evolution of $C_2/C_1$ ratios that account for internalization and externalization rates—$E = 46kV/m$, $\lambda_{inl} = 0.01m/s$ and $TPK$.**
(TIF)

**S17 Fig. Spatio-temporal evolution of bound intracellular concentration $C_3$ that account for association and dissociation—$E = 46kV/m$, $\lambda_{inl} = 0.001m/s$ and $TPK$.**
(TIF)

**S18 Fig. Spatio-temporal evolution of bound intracellular concentration $C_3$ that account for association and dissociation—$E = 46kV/m$, $\lambda_{inl} = 0.01m/s$ and $TPK$.**
(TIF)

**S19 Fig. Spatio-temporal evolution of the presence and interaction of reaction and transport mechanisms—$E = 46kV/m$, $\lambda_{inl} = 0.0001m/s$ and $MPK$.**
(TIF)

**S20 Fig. Spatio-temporal evolution of the presence and interaction of reaction and transport mechanisms—$E = 46kV/m$, $\lambda_{inl} = 0.001m/s$ and $MPK$.**
(TIF)

**S21 Fig. Spatio-temporal evolution of the presence and interaction of reaction and transport mechanisms—$E = 46kV/m$, $\lambda_{inl} = 0.01m/s$ and $MPK$.**
(TIF)

**S22 Fig. Spatio-temporal evolution of $C_2/C_1$ ratios that account for internalization and externalization rates—$E = 46kV/m$, $\lambda_{inl} = 0.001m/s$ and $MPK$.**
(TIF)

**S23 Fig. Spatio-temporal evolution of $C_2/C_1$ ratios that account for internalization and externalization rates—$E = 46kV/m$, $\lambda_{inl} = 0.01m/s$ and $MPK$.**
(TIF)

**S24 Fig. Spatio-temporal evolution of bound intracellular concentration $C_3$ that account for association and dissociation—$E = 46kV/m$, $\lambda_{inl} = 0.001m/s$ and $MPK$.**
(TIF)

**S25 Fig. Spatio-temporal evolution of bound intracellular concentration $C_3$ that account for association and dissociation—$E = 46kV/m$, $\lambda_{inl} = 0.01m/s$ and $MPK$.**
(TIF)

**S26 Fig. Spatio-temporal evolution of the presence and interaction of reaction and transport mechanisms—$E = 70kV/m$, $\lambda_{inl} = 0.0001m/s$ and $TPK$.**
(TIF)

**S27 Fig. Spatio-temporal evolution of the presence and interaction of reaction and transport mechanisms—$E = 70kV/m$, $\lambda_{inl} = 0.001m/s$ and *TPK*.**
(TIF)

**S28 Fig. Spatio-temporal evolution of the presence and interaction of reaction and transport mechanisms—$E = 70kV/m$, $\lambda_{inl} = 0.01m/s$ and *TPK*.**
(TIF)

**S29 Fig. Spatio-temporal evolution of $C_2/C_1$ ratios that account for internalization and externalization rates—$E = 70kV/m$, $\lambda_{inl} = 0.0001m/s$ and *TPK*.**
(TIF)

**S30 Fig. Spatio-temporal evolution of $C_2/C_1$ ratios that account for internalization and externalization rates—$E = 70kV/m$, $\lambda_{inl} = 0.001m/s$ and *TPK*.**
(TIF)

**S31 Fig. Spatio-temporal evolution of $C_2/C_1$ ratios that account for internalization and externalization rates—$E = 70kV/m$, $\lambda_{inl} = 0.01m/s$ and *TPK*.**
(TIF)

**S32 Fig. Spatio-temporal evolution of bound intracellular concentration $C_3$ that account for association and dissociation—$E = 70kV/m$, $\lambda_{inl} = 0.0001m/s$ and *TPK*.**
(TIF)

**S33 Fig. Spatio-temporal evolution of bound intracellular concentration $C_3$ that account for association and dissociation—$E = 70kV/m$, $\lambda_{inl} = 0.001m/s$ and *TPK*.**
(TIF)

**S34 Fig. Spatio-temporal evolution of bound intracellular concentration $C_3$ that account for association and dissociation—$E = 70kV/m$, $\lambda_{inl} = 0.01m/s$ and *TPK*.**
(TIF)

**S35 Fig. Spatio-temporal evolution of the presence and interaction of reaction and transport mechanisms—$E = 70kV/m$, $\lambda_{inl} = 0.0001m/s$ and *MPK*.**
(TIF)

**S36 Fig. Spatio-temporal evolution of the presence and interaction of reaction and transport mechanisms—$E = 70kV/m$, $\lambda_{inl} = 0.001m/s$ and *MPK*.**
(TIF)

**S37 Fig. Spatio-temporal evolution of the presence and interaction of reaction and transport mechanisms—$E = 70kV/m$, $\lambda_{inl} = 0.01m/s$ and *MPK*.**
(TIF)

**S38 Fig. Spatio-temporal evolution of $C_2/C_1$ ratios that account for internalization and externalization rates—$E = 70kV/m$, $\lambda_{inl} = 0.0001m/s$ and *MPK*.**
(TIF)

**S39 Fig. Spatio-temporal evolution of $C_2/C_1$ ratios that account for internalization and externalization rates—$E = 70kV/m$, $\lambda_{inl} = 0.001m/s$ and *MPK*.**
(TIF)

**S40 Fig. Spatio-temporal evolution of $C_2/C_1$ ratios that account for internalization and externalization rates—$E = 70kV/m$, $\lambda_{inl} = 0.01m/s$ and *MPK*.**
(TIF)

**S41 Fig. Spatio-temporal evolution of bound intracellular concentration $C_3$ that account for association and dissociation—$E = 70kV/m$, $\lambda_{inl} = 0.0001m/s$ and $MPK$.**
(TIF)

**S42 Fig. Spatio-temporal evolution of bound intracellular concentration $C_3$ that account for association and dissociation—$E = 70kV/m$, $\lambda_{inl} = 0.001m/s$ and $MPK$.**
(TIF)

**S43 Fig. Spatio-temporal evolution of bound intracellular concentration $C_3$ that account for association and dissociation—$E = 70kV/m$, $\lambda_{inl} = 0.01m/s$ and $MPK$.**
(TIF)

**S1 File.**
(ZIP)

## Author Contributions

**Conceptualization:** Fabián Mauricio Vélez Salazar, Iván David Patiño.

**Formal analysis:** Fabián Mauricio Vélez Salazar, Iván David Patiño.

**Investigation:** Fabián Mauricio Vélez Salazar, Iván David Patiño.

**Methodology:** Fabián Mauricio Vélez Salazar, Iván David Patiño.

**Validation:** Fabián Mauricio Vélez Salazar, Iván David Patiño.

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
