## [Decision Letter · Decision Letter 0]

6 Aug 2024

PONE-D-24-19637In-silico tool based on Boolean Networks and meshless simulations for prediction of reaction and transport mechanisms in the systemic administration of chemotherapeutic drugsPLOS ONE

Dear Dr. Vélez Salazar,

Thank you for submitting your manuscript to PLOS ONE. After careful consideration, we feel that it has merit but does not fully meet PLOS ONE’s publication criteria as it currently stands. Therefore, we invite you to submit a revised version of the manuscript that addresses the points raised during the review process.

We look forward to receiving your revised manuscript.

Kind regards,

Baeckkyoung Sung, Ph.D.

Academic Editor

PLOS ONE

Journal Requirements:

"NO authors have competing interests"

Additional Editor Comments:

This manuscript tries to deal with an interesting subject in the field of pharmaceutical computational modelling, but it seems that the work is only acceptable when there is a significant improvement in the manuscript quality based on the reviewers' comments.

Reviewers' comments:

Reviewer's Responses to Questions

**Comments to the Author**

1. Is the manuscript technically sound, and do the data support the conclusions?

Reviewer #1: No

Reviewer #2: Yes

Reviewer #3: Yes

Reviewer #4: No

2. Has the statistical analysis been performed appropriately and rigorously? 

Reviewer #1: N/A

Reviewer #2: Yes

Reviewer #3: I Don't Know

Reviewer #4: N/A

3. Have the authors made all data underlying the findings in their manuscript fully available?

Reviewer #1: No

Reviewer #2: Yes

Reviewer #3: Yes

Reviewer #4: Yes

4. Is the manuscript presented in an intelligible fashion and written in standard English?

Reviewer #1: No

Reviewer #2: Yes

Reviewer #3: Yes

Reviewer #4: No

5. Review Comments to the Author

Reviewer #1: The authors present a work that models three variables (bloodstream velocity, pharmacokinetic dynamics and electroporation intensity) that affect the drug delivery to a human tissue. This work continues a previous one in which they build a fairly similar model. The manuscript is hard to read, unclear in its motivations, unclear in its most relevant result, unclear in its usefulness as a clinical tool and lacks proper methods, discussion and connections to state of the art results. The text and the figures can be much improved to increase the readability and relevance of the work. I cannot accept this work in its present form.

Major changes:

1. The authors should better motivate their work.

1.1. It is not clear if this work is exploratory or if it aims at presenting clinical interventions. The authors simulate different values of three variables: bloodstream velocity, pharmacokinetic dynamics and electroporation intensity. Only the electroporation is a variable that could be used in a clinical intervention. The blood stream is hardly useful for clinicians and the pharmacokinetic is a property of the drugs.

1.2. The authors seem to completely ignore Fick’s second law in their analysis. This causes sentences like “It is also noticed that the highest internalization rates () occur at the first hour of evaluation ( = 1ℎ), where the 2⁄1 values are the lowest in the whole time window.” That should be self-evident to anyone who knows about Fick’s second law.

1.3. Two pharmacokinetic models are presented without any motivation and without presenting real cases of real drugs that follow any of these dynamics

1.4. What is the motivation of the many Boolean rules? Their only use seems to be to analyse the numerous results of the simulation in colours as in a lookup table, but explaining their details takes many pages and non-useful charts and tables.

1.5. In the Discussion section, page 41, the authors claim that the Boolean rules “could be useful for establishing appropriate drug dosages and pharmacokinetic profiles”.

1.5.1. Where are the drug dosages results?

1.5.2. How do the authors propose that clinicians establish appropriate pharmacokinetic profiles?

1.6. In the Discussion section the authors talk about protein content, while this is not included in their model. What they include is protein association to drug, which is not the same.

2. The text needs a complete rewrite. In its current state the work is just a long description of a model and a pure description of the resulting figures:

2.1. The work is difficult to read. It has too many details, too many acronyms, too many repetitions of the variables and too many uninteresting results. The authors should consider to select 2-3 combinations of their variables, explain and discuss them and leaving the rest for the SuppMat.

2.2. The introduction is a paragraph of true introduction followed by a long-winded description of Figure 1. This description looks like a review and summarises methods used in other works, but it’s unclear how the present work uses them. This description should go to SuppMat and leave here one single summarised paragraph (same for Figure 2 and 3). Also, a lot of information is given that is completely unnecessary for the understanding of the paper (DNA ability to go through electroporated holes?)

2.3. The paper lacks a methods section. The section “Data and Simulation campaign” is just a description of the values and PK models used. There is no way that anyone can reproduce this work with only that information. The authors have not explained how they have simulated their equations (what solver or tool), what they have used to plot the results, how does the Boolean rules connect to the Concentration equations.

2.4. The description of the results does not provide insights into the problem without any talk about synergies among variables under study and with many self-evident phrases such as:

2.4.1. “As can be observed, the axial distribution of 3 is not uniform for the three blood speeds, indicating that the association/dissociation rate is not the same for a set of points equidistant from the border” when the bloodstream has an inlet and outlet

2.4.2. “According to the numerical results, the for the One-short Tri-exponential () and Mono-exponential profile () can differ each other for a given combination of and .” Of course different PK dynamics have different results

2.4.3. MPK model “leads to a higher possibility of increased the final cell death since drug resides inside the cells during more time.” This is evident when drugs act killing cells. In fact, authors should read about the most common drug resistance mechanisms.

2.5. The paper lacks a proper discussion section. The present “Results and discussion” section is a dry section of pure description of the figures. There is no discussion, there is no connection of these results with state of the art.

2.6. The paper lacks a proper conclusions section. The present “Conclusions” section is a summary of the results and handwaved connections to clinical usefulness.

3. Relationship of this work with the state of the art:

3.1. The text is not clear in what is the added value of this work in respect to the state of the art. What is the gap in the state of the art that this work fills?

3.2. The text in unclear in respect to what are the precise contributions of this work and what has been inherited from the authors previous work (ref 7). The present work state at the end of the introduction “The principal purpose of the present research is to investigate the influence of the combined input parameters (, , ) on the interaction between the aforementioned reaction and transport mechanisms in the context of electro-chemotherapeutic treatments, aiming to propose potential clinical implications resulting from this combination.” How is this different from ref 7?

4. References

4.1. The authors do not need to cite 3 or 4 papers each time they present a methods. This could be cut down to 1 or 2. An example of this is page 14 when refs 70-74 are used to cite the approach used.

4.2. Even many times these references are not even using the method they are being cited for. An example of this is page 16 when refs 76-79 are cited but none of them are using Boolean rules as the authors are using (to discretise a continuous behaviour). Another one is refs 80-83, which of the three talks about the “design potential electro-chemotherapeutics protocols”.

4.3. References are needed for these sentences:

4.3.1. “The protein content of cancer patients has a significant role in the side effects of the treatment.”

4.3.2. “Patients with low protein levels, known as hypoproteinemia, are more susceptible to side effects and require lower drug doses.”

5. The figures need a complete rework:

5.1. The figure footers need to be explanatory. The authors barely describe the figures. For instance in Figure 6 the acronyms should be explained there, not in the text.

5.2. In Figures 7 forward, the panels are the same patch of tissue at different times. The authors should remove the space in-between panels and present them as facets of a single figure

5.3. In Figures 7 forward, the authors should use a common legend for all the facets. This would help understand better the colours across time points (as in Fig 9).

Minor:

1. Use page number and line numbers. It helps reviewing papers.

Reviewer #2: The article describes a numerical simulation to predict the concentration gradients of a chemotherapeutical drug in a tumor cord model as well as a Boolean model, which highlights the main mechanism of drug movement within the tumor cord. The publication also explores the effect of electroporation on the drug distribution within the represented fluid compartments and drug transport mechanisms. Both the differential equations and especially the Boolean tables are very well detailed. The figures generated by the simulation and their explanation are also very descriptive. However, I have some comments:

1. In the first section “when it occurs from the vessel to the interstitial space (extravasation), and as − or when it occurs in the opposite direction (lymphatic drainage”) lymphatic drainage was mentioned but based on the mathematical formulation it is not completely clear how was this implemented during the simulation. Also, Figure 1. Implies a localized drainage compared to the description of the model (Figure 2.), and the boundaries of the tumor cord (Figure 6.)

2. The diffusion back to the blood vessel, once the free plasma concentration decreases below the tumor tissue’s free drug concentration is not represented in either the Figures or the text. If it is present would it be possible to highlight it? If not present, adding this calculation to the Boolean model ( Cv/C1 – increasing or decreasing ) would be highly relevant (especially regarding the optimization of the timing of the electroporating magnetic field in relation to drug accumulation in the cancer tissue).

3. Is the Tchemo referring simply to the length of the simulation evaluated or does it have further meaning? It is not described in the text.

4. In section 1 it says: “The appropriate choice of parameters such as the voltage level, pulse form, spacing, length, and frequency, as well as number and duration of electroporation treatments, and electrodes location is crucial for achieving the primary objective of reversible electroporation (), namely, to augment the cell membrane permeability and facilitate the drug translocation towards the intracellular space while maintaining the cell viability [2], [3], [4].” But in Figure 6. The location magnetic field/poles are not represented.

5. In section 6 :“Facilitating the interstitial transportation of drug ( +) to cancer cell receptors is advantageous for Antibody-Drug Conjugate () therapies, as the drug is only released upon internalization”. It is not clear how can you optimize the model for drugs where the drug entry to the cell is ensured by single-use proteins (e.g.: one surface protein allows one ADC molecule to enter the cell and then its degraded)

6. It would be intriguing to see the doxorubicin content of the tumor cord in the different fluid compartments ( intravasal / Extracellular / Intracellular free /intracellular protein bound ) since the color scales – rightfully so – are cannot be synchronised. IT would help with the contextualization of the drug distribution within the model in the different time points. Figures 9-10. ; Figures 12-17.

7. One typo was noticed in the abstract

a. “Using in-house computational tools, this works investigates” - » “Using in-house computational tools, this work investigates

Reviewer #3: This study investigates the effects of electroporation (both its absence and varying intensity), different bloodstream velocities, and two pharmacokinetic profiles on the reaction and transport mechanisms in electro-chemotherapeutic treatment. These effects are analyzed through numerical simulations of the tumor cord model using the GMAPS scheme. Subsequently, a Boolean model is employed to analyze the interactions between the reaction and transport mechanisms by examining the evolution of ratios among different concentrations: extracellular drug concentration, free intracellular drug concentration, and bound intracellular drug concentration. Some points should be considered before further consideration of the manuscript, as follows;

1) The abstract should not include background information. The authors should move the initial sentences of the abstract to the introduction. The abstract should concisely describe the purpose of the research, the novelty of the work, the key results, and the conclusion.

2) The authors need to clearly articulate their study's contributions in comparison to previous research in both the abstract and the final paragraph of the introduction. They should emphasize any distinctions from various perspectives, including differences in the mathematical model, the incorporation of additional biological factors, the increased complexity of the study's physics, and the investigation of new subjects.

3) The quality of the contours shown in Figures 7-17 and in the supplementary file needs improvement. Currently, the readability of these contours is suboptimal.

4) After introducing abbreviations at their first mention, it is recommended to consistently use them thereafter and avoid reintroducing the full terms. The authors should use the abbreviations for the terms once they have been introduced, if they intend to abbreviate.

5) It is recommended that the authors incorporate the latest research into the introduction. For example, the study found at https://www.mdpi.com/2072-6694/15/22/5464 explores multi-scale modeling of drug delivery into solid tumors, addressing transport along blood vessels, transvascular transport, and extravascular transport. This research also examines the effects of the microvascular network and the changes in transport properties induced by vascular normalization from anti-angiogenic therapy on the convective and diffusive transport mechanisms of drug delivery.

6) It is recommended to include a flowchart of the current numerical study, which summarizes the application of the numerical method, the Boolean model, and the various scenarios examined (excluding electroporation, with two different electroporation values, different bloodstream velocities, and pharmacokinetic profiles). Providing such a flowchart will enhance the clarity and comprehensibility of the study for readers.

Reviewer #4: The paper "In-silico tool based on Boolean Networks and meshless simulations for prediction of

reaction and transport mechanisms in the systemic administration of chemotherapeutic

drugs." reports on systemic administration of chemotherapeutic drugs comprises some reaction and transport mechanisms ().

Comments:

1) The software / computational tools have already been previously reported:

F. M. Vélez Salazar and I. D. Patiño Arcila, “Influence of electric field, blood velocity, and

pharmacokinetics on electrochemotherapy efficiency,” Biophys J, vol. 122, no. 16, 2023, doi:

10.1016/j.bpj.2023.07.004.

The novelty in this manuscript is

"Subsequently, a Boolean model is proposed here to analyze the interaction between 

based on the comparison of the spatio-temporal evolution of concentration ratios."

So is then the Boolean model available for the interested reader?

2) The model is presented and impressively complex. Now, chemotherapy is routinely administered without such a complex model, can you explain the reader, why he/she should then use the

complex model of yours instead of simpler heuristics (e.g. dosage per weight, per surface, combinations of such parameters etc. etc.)?

3) For the non-specialist reader there are no experimental data visible. Also the supplement seems to have only even more simulations. Can you please explain is your model validated by experimental data or this is not yet the case?

4) Can you educate the reader of an actual, convincing use case of your model in the clinic?

5) Or is the usage thought more for fundamental biophysics research (in this case, please explain

for which biophysics question the model is useful).

6. PLOS authors have the option to publish the peer review history of their article (what does this mean?). If published, this will include your full peer review and any attached files.

Reviewer #1: No

Reviewer #2: No

Reviewer #3: No

Reviewer #4: No

---

## [Author Response · Author response to Decision Letter 0]

19 Sep 2024

Thank you very much for reviewing our manuscript. Each and every reviewer's comments were carefully addressed and highlighted in the response to reviewers document.

---

## [Decision Letter · Decision Letter 1]

23 Oct 2024

PONE-D-24-19637R1In-silico tool based on Boolean Networks and meshless simulations for prediction of reaction and transport mechanisms in the systemic administration of chemotherapeutic drugsPLOS ONE

Dear Dr. Vélez Salazar,

Thank you for submitting your manuscript to PLOS ONE. After careful consideration, we feel that it has merit but does not fully meet PLOS ONE’s publication criteria as it currently stands. Therefore, we invite you to submit a revised version of the manuscript that addresses the points raised during the review process.

We look forward to receiving your revised manuscript.

Kind regards,

Baeckkyoung Sung, Ph.D.

Academic Editor

PLOS ONE

Journal Requirements:

**Additional Editor Comments:**

The authors are invited to further revise the manuscript based on the comments by Reviewer 4.

Reviewers' comments:

Reviewer's Responses to Questions

**Comments to the Author**

1. If the authors have adequately addressed your comments raised in a previous round of review and you feel that this manuscript is now acceptable for publication, you may indicate that here to bypass the “Comments to the Author” section, enter your conflict of interest statement in the “Confidential to Editor” section, and submit your "Accept" recommendation.

Reviewer #2: All comments have been addressed

Reviewer #3: All comments have been addressed

Reviewer #4: (No Response)

2. Is the manuscript technically sound, and do the data support the conclusions?

Reviewer #2: Yes

Reviewer #3: Yes

Reviewer #4: Yes

3. Has the statistical analysis been performed appropriately and rigorously? 

Reviewer #2: Yes

Reviewer #3: N/A

Reviewer #4: Yes

4. Have the authors made all data underlying the findings in their manuscript fully available?

Reviewer #2: Yes

Reviewer #3: Yes

Reviewer #4: No

5. Is the manuscript presented in an intelligible fashion and written in standard English?

Reviewer #2: Yes

Reviewer #3: Yes

Reviewer #4: Yes

6. Review Comments to the Author

Reviewer #2: (No Response)

Reviewer #3: From my perspective, this version of the paper can be accepted for publication.

Reviewer #4: Regarding the author responses, I have the following comments:

Model availability, your response is "If manuscript is accepted, algorithm

will be available on demand."

 This is not sufficient for progress and FAIR data policy in science, the model must become available for download as a condition for acceptance.

Comment 2: Fine, complement to existing heuristics, nice pharmacological example given and in the

manuscript revisions.

Comment 3: Experimental data are lacking -- still the case, but data from literature were collected,

so not optimal but ok.

Comment 4: Not well answered, if the practical use case was published before (ref 17) and is lacking here. Rather add your nice pharmacological model of chemotherapeutic concentrations and explain how you can protect the patient better from side-effects with your model

Comment 5: Fine.

Summary: Comment 1 and Comment 4 still need to be adressed.

7. PLOS authors have the option to publish the peer review history of their article (what does this mean?). If published, this will include your full peer review and any attached files.

Reviewer #2: **Yes: **Daniel Veres

Reviewer #3: No

Reviewer #4: No

---

## [Author Response · Author response to Decision Letter 1]

6 Nov 2024

All reviewer comments were addressed both in the response letter to the reviewers and in the manuscript.

---

## [Decision Letter · Decision Letter 2]

22 Nov 2024

In-silico tool based on Boolean Networks and meshless simulations for prediction of reaction and transport mechanisms in the systemic administration of chemotherapeutic drugs

PONE-D-24-19637R2

Dear Dr. Vélez Salazar,

We’re pleased to inform you that your manuscript has been judged scientifically suitable for publication and will be formally accepted for publication once it meets all outstanding technical requirements.

Kind regards,

Baeckkyoung Sung, Ph.D.

Academic Editor

PLOS ONE

Additional Editor Comments (optional):

The authors have well responded to the reviewers' comments and the revised manuscript can be considered for acceptance.

Reviewers' comments:

Reviewer's Responses to Questions

**Comments to the Author**

1. If the authors have adequately addressed your comments raised in a previous round of review and you feel that this manuscript is now acceptable for publication, you may indicate that here to bypass the “Comments to the Author” section, enter your conflict of interest statement in the “Confidential to Editor” section, and submit your "Accept" recommendation.

Reviewer #3: All comments have been addressed

Reviewer #4: All comments have been addressed

2. Is the manuscript technically sound, and do the data support the conclusions?

Reviewer #3: Yes

Reviewer #4: Yes

3. Has the statistical analysis been performed appropriately and rigorously? 

Reviewer #3: N/A

Reviewer #4: N/A

4. Have the authors made all data underlying the findings in their manuscript fully available?

Reviewer #3: Yes

Reviewer #4: Yes

5. Is the manuscript presented in an intelligible fashion and written in standard English?

Reviewer #3: Yes

Reviewer #4: Yes

6. Review Comments to the Author

Reviewer #3: (No Response)

Reviewer #4: (No Response)

7. PLOS authors have the option to publish the peer review history of their article (what does this mean?). If published, this will include your full peer review and any attached files.

Reviewer #3: No

Reviewer #4: No

---

## [Editor Report · Acceptance letter]

29 Nov 2024

PONE-D-24-19637R2 

PLOS ONE

Dear Dr. Vélez Salazar, 

I'm pleased to inform you that your manuscript has been deemed suitable for publication in PLOS ONE. Congratulations! Your manuscript is now being handed over to our production team.

Kind regards, 

on behalf of

Dr. Baeckkyoung Sung 

Academic Editor

PLOS ONE